# Global satellite survey reveals uncertainty in landfill methane emissions

Matthieu Dogniaux[1✉], Joannes D. Maasakkers[1], Marianne Girard[2], Dylan Jervis[2], Jason McKeever[2], Berend J. Schuit[1,2], Shubham Sharma[1], Ana Lopez-Noreña[1], Daniel J. Varon[3,4] & Ilse Aben[1,5]

Methane is a potent but short-lived greenhouse gas and rapid reductions of its anthropogenic emissions could help decrease near-term warming[1]. Solid waste emits methane through the decay of organic material, which amounts to about 10% of total anthropogenic methane emissions[2]. Satellite instruments[3] enable monitoring of strong methane hotspots[4], including many strongly emitting urban areas that include solid waste disposal sites as most prominent sources[5]. Here we present a survey of methane emissions from 151 individual waste disposal sites across six continents using high-resolution satellite observations that can detect localized methane emissions above 100 kg h$^{-1}$. Within this dataset, we find that our satellite-based estimates generally show no correlation with reported or modelled emission estimates at facility scale. This reveals major uncertainties in the current understanding of methane emissions from waste disposal sites, warranting further investigations to reconcile bottom-up and top-down approaches. We also observe that managed landfills show lower emission per area than dumping sites, and that detected emission sources often align with the open non-covered parts of the facility where waste is added. Our results highlight the potential of high-resolution satellite observations to detect and monitor methane emissions from the waste sector globally, providing actionable insights to help improve emission estimates and focus mitigation efforts.

Global waste production has nearly tripled since 1965, reaching 2 billion tonnes per year in 2016 and, with a growing population[6], is expected to further increase by 70% by 2050[7]. Close to 70% of waste currently ends up in landfills or dumping sites[7], in which anaerobic decomposition of organic material produces methane. Methane is a short-lived (with about a nine-year lifetime[8]) but potent greenhouse gas and its anthropogenic emissions make it the second most important contributor to human-induced climate change after anthropogenic carbon dioxide emissions, accounting for ~30% of current positive warming relative to pre-industrial temperatures (1850–1900 average)[9]. Deep and rapid reductions in global anthropogenic methane emissions are essential to keep net warming below 1.5 °C by 2100[1,10]. Methane emissions from solid waste currently amount to 38 million tonnes per year, roughly 10% of total anthropogenic methane emissions[2], and could reach 60 million tonnes annually by 2050[11]. However, some mitigation options are available, for example, banning organic waste in landfills, source separation, reuse, recycling or treatment with an anaerobic digester[11]. If these are implemented to their fullest potential, 2050 methane emissions from solid waste could be as low as 11 million tonnes per year[11].

Solid waste emission estimates are based on widely used first-order decay models[12] that are used in country-level reporting of methane emissions[13] as well as at facility scale. Different variants of such models exist and can yield very different results for similar facilities[14].

The parameters (for example, methane generation potential of the waste) that drive these models are also uncertain and specific to each facility[15,16]. Finally, waste disposal management practices can greatly impact methane emissions, from unmanaged dumping sites to managed sanitary landfills that include linings, covers and gas capture systems of variable efficiency[17]. Considering all of these uncertainties, independent observations of methane emitted from waste disposal sites are critical and can be obtained through various on-ground and/or aerial-measurement methods[18] that are deployable at the site level and that can provide emission estimates at high continuous temporal resolution. Complementarily to these site-specific approaches, satellites offer extensive global coverage, providing consistent observation sets across a large number of sites. We present here a global-scale survey of methane emissions from waste disposal sites using 1,447 high-resolution satellite observations.

Satellite remote sensing of atmospheric methane can have an active role in methane emission mitigation by locating emission hotspots and identifying the super-emitting sources they contain[19]. Over the past decade, a range of spaceborne instruments have been transformative for methane imaging from space[3,20–24]. They provide spatial images of atmospheric methane concentrations that enable the detection of anthropogenic emission plumes. These consist of strong enhancements in methane concentration that extend downwind

[1]SRON Space Research Organisation Netherlands, Leiden, The Netherlands. [2]GHGSat Inc., Montreal, Canada. [3]Department of Aeronautics and Astronautics, Massachusetts Institute of Technology, Cambridge, MA, USA. [4]Institute for Data, Systems, and Society, Massachusetts Institute of Technology, Cambridge, MA, USA. [5]Department of Earth Sciences, Vrije Universiteit Amsterdam, Amsterdam, The Netherlands. ✉e-mail: M.Dogniaux@sron.nl

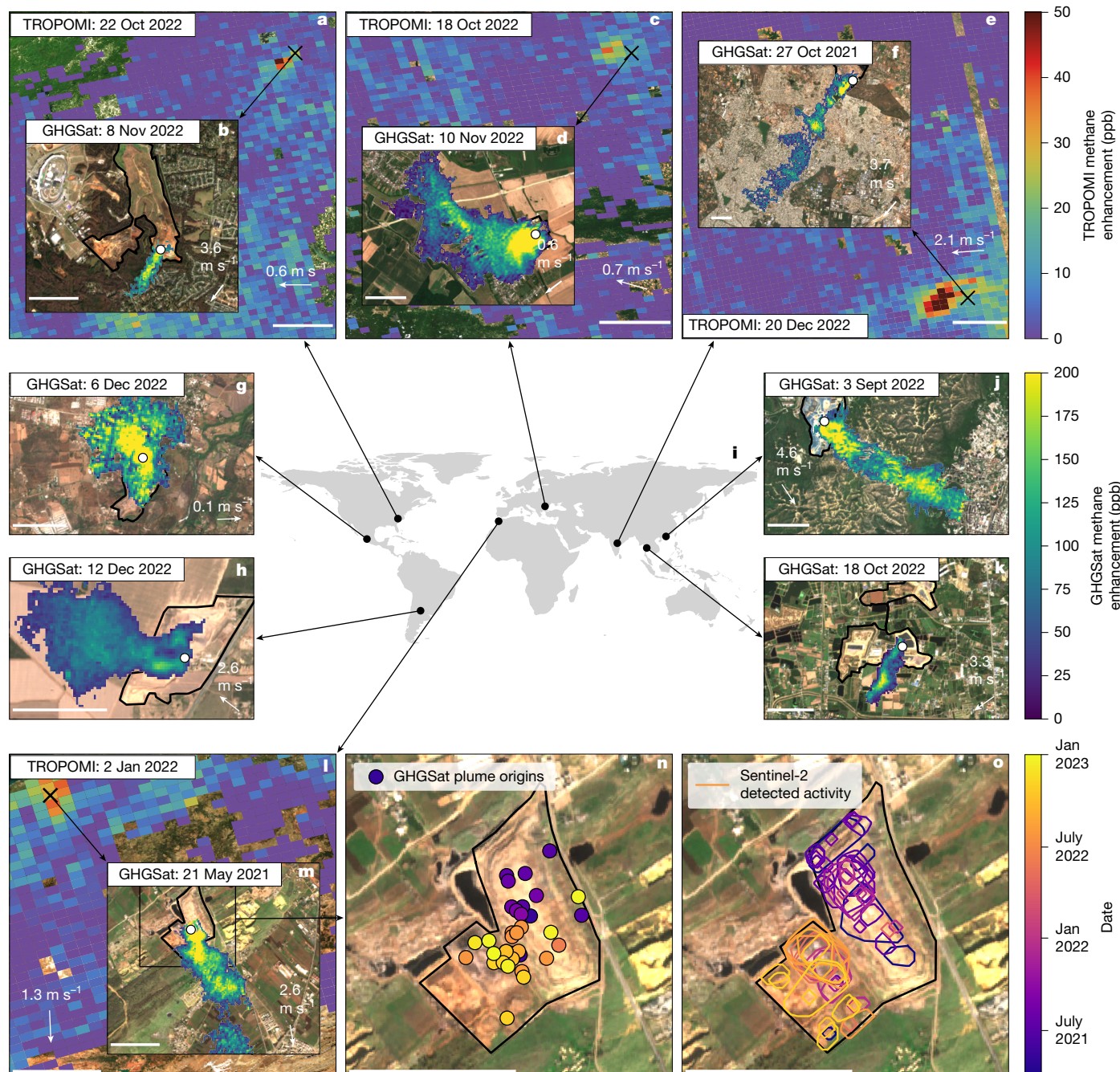

**Fig. 1 | Examples of urban- and facility-scale satellite observations of methane plumes. a–o**, Examples of GHGSat facility-scale (**b**,**d**,**f**,**g**,**h**,**j**,**k**,**m**) and TROPOMI-detected urban-area (**a**,**c**,**e**,**l**) methane emission plumes for urban areas in Charlotte (USA, **a**,**b**), Bucharest (Romania, **c**,**d**), Hyderabad (India, **e**,**f**), Guadalajara (Mexico, **g**), Córdoba (Argentina, **h**), Hong Kong (China, **j**), Bangkok (Thailand, **k**) and Casablanca (Morocco, **l**,**m**). The spatio-temporal distributions of all GHGSat plume origins and Sentinel-2-detected surface activity (structural changes between visual Sentinel-2 images; Methods) for the Casablanca landfill are shown in **n** and **o**, respectively. Black crosses mark site locations, whereas white dots represent the GHGSat plume origins and thick black contours demarcate landfill site boundaries. White arrows (**a**–**h**, **j**–**m**) illustrate the wind direction sampled from the ERA5 reanalysis[48], with the associated labels indicating the wind speed. Plumes overall follow the reanalysis wind direction, with some exceptions at low wind speeds. Background imagery relies on non-concurrent Sentinel-2 data (2022). Copernicus Sentinel-2 data in parts **a**–**h**,**j**–**o** are adapted from Google Earth Engine[49,50]. Scale bars, 50 km (**a**,**c**,**e**,**l**), 1 km (**b**,**d**,**f**,**g**,**h**,**j**,**k**,**m**,**n**,**o**).

from localized emission sources, as illustrated in Fig. 1. Calibrated mass-balance approaches are employed to translate these instantaneous snapshots into emission rates (Methods), validated by single-blind controlled release[25]. Our study focuses on measurements from GHGSat's high-resolution (~25 × 25 m²) methane imaging satellites, which capture targeted 12 × 15 km² scenes and detect facility-scale plumes arising from localized sources at emission rates as low as

100 kg h⁻¹. They can be attributed to individual sources across oil and gas facilities (onshore and offshore), coal mines and waste disposal sites[5,26–28]. Many of these individual sources were first coarsely spatially identified[5,29–33] with the Tropospheric Monitoring Instrument (TROPOMI) on board the Sentinel-5 Precursor satellite[34]. It maps the atmospheric methane concentration with daily global coverage and a resolution of up to 7 × 5.5 km², enabling the detection[4] of methane

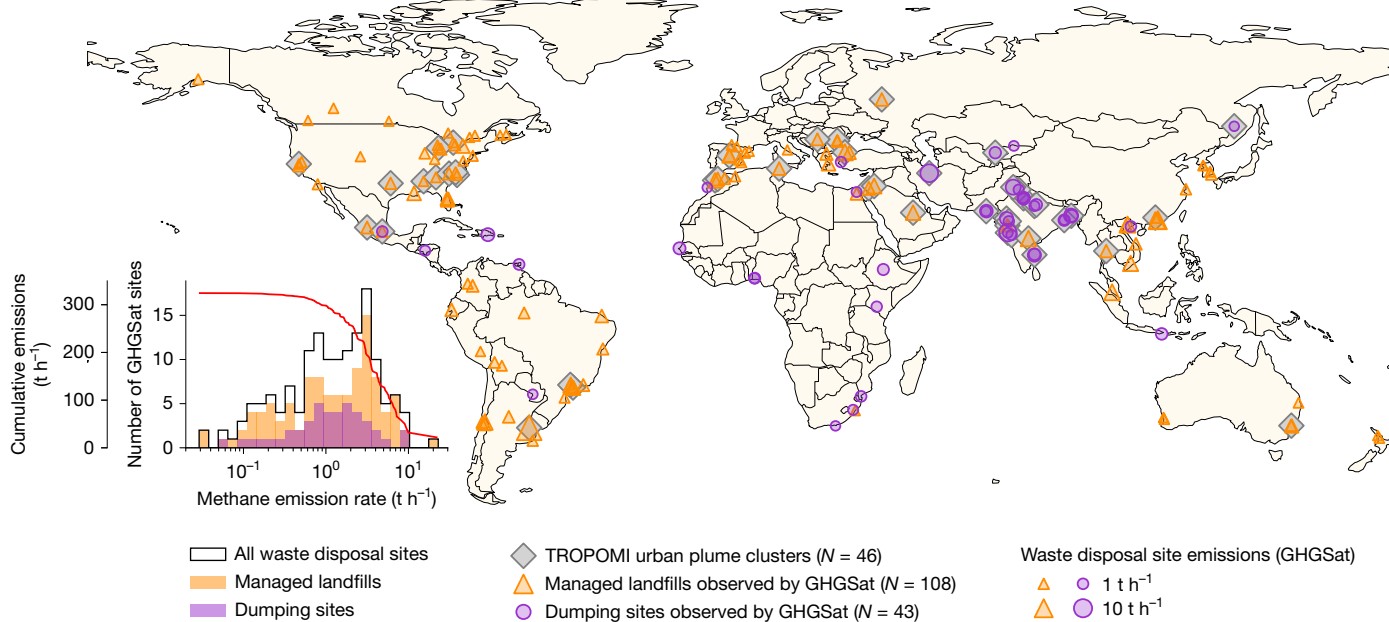

**Fig. 2 | Distribution of GHGSat-observed waste disposal sites and their urban areas.** Location of the 151 waste disposal sites observed by GHGSat satellites, and the 46 out of 130 corresponding urban areas for which methane emission plumes have been detected in TROPOMI data (grey). GHGSat methane emission rate distributions over logarithmically spaced bins are given for all sites (black line), and separately for managed landfills (orange) and dumping sites (purple). This site classification has been manually determined using satellite and aerial imagery from Google Earth (see main text). The red line shows cumulative emissions. The site-level and urban-area-level data supporting this figure are provided in the Supplementary Information.

plumes that can be followed-up—in a so-called tip-and-cue strategy—by targeted GHGSat observations to pinpoint their exact sources. This approach has been demonstrated for four urban areas with strongly emitting waste disposal sites[5]. Here we present a global GHGSat-based survey of methane emissions from waste disposal sites across 130 urban areas in 47 countries during 2021–2022.

One-third of methane emission plumes detected in TROPOMI data from the year 2021 is related to urban areas[4]. In 2021 and 2022, we detect 897 plumes with TROPOMI across 46 urban areas among the 130 covered by GHGSat (Extended Data Fig. 1 and Supplementary Notes 1 and 2). These detections—which depend on observational coverage and the magnitude of emissions (Extended Data Fig. 2 and Supplementary Note 3)—are located on six different continents, with the majority coming from Asia (Fig. 2). TROPOMI plumes illustrate the mitigation potential concentrated in urban areas, which harbour a range of sources such as wastewater treatment, natural gas distribution and incomplete combustion[35]. Waste disposal sites, however, are some of the most concentrated and mitigable sources in urban areas and are therefore the facilities that we focus on in our GHGSat analysis.

## A global facility-scale satellite survey

We use 1,447 clear-sky observations acquired by GHGSat's C1–C5 satellites in 2021 and 2022. These were targeted at 151 different waste disposal sites located in 130 urban areas scattered over six continents, as shown in Fig. 2. Only sites for which at least one methane emission plume has been detected by GHGSat are included, meaning that our sample is on the upper end of the global waste disposal site emission rate distribution. The median number of GHGSat observations per site is 5, with 23 sites that have been observed at least 20 times (Extended Data Fig. 3). These are opportunistic observations that could be made in parallel to regular GHGSat activities—a substantial fraction (51%) of which intersect with TROPOMI-detected urban methane hotspots.

Out of the 1,447 observations, 1,013 show at least one emission plume above GHGSat's detection threshold (1,085 plumes in total, Fig. 1; quantified as described in Methods). We conservatively consider the emission rate of the 434 site-level null detections to be zero even though we may miss (possibly diffuse) emissions that are lower than the GHGSat detection threshold. The positive plume detection rate per site ranges from 7% (two plumes among 30 observations at Icheon, South Korea) to 100% (which we find for 74 sites). The median of the plumes' detected methane emission rates is 2.4 t h$^{-1}$, with 5th and 95th percentiles of 0.5 t h$^{-1}$ and 15.1 t h$^{-1}$, respectively. The median relative uncertainty of these emission rates is ~45% (Supplementary Note 4).

Recurrent observations allow us to investigate the potential drivers of the detected emission variability. We compared site-wise emission variability against several meteorological variables (10 m wind speed, 2 m temperature, surface pressure, surface pressure change and accumulated precipitation over two weeks) as well as the hemisphere-corrected day in the year, but we did not find any significant link between them (Extended Data Fig. 4 and Supplementary Note 5); however, past on-site studies have indicated that surface pressure change drives landfill methane emissions[36–38]. Our findings based on satellite observations of high-emitting active sites are consistent with recent airborne-based results[39]. This finding could be explained by meteorological driving producing too small emission changes compared to single observation uncertainty for the sites included in our dataset.

The median site-wise averaged emission rate is 1.2 t h$^{-1}$ (including null detections), with 5th and 95th percentiles of 0.1 t h$^{-1}$ and 6.8 t h$^{-1}$, respectively (Methods, Extended Data Fig. 5 and Supplementary Note 4). The lowest three site-averaged detected emission rates are found at a Canadian landfill in British Columbia (0.03 ± 0.04 t h$^{-1}$), an Italian landfill near Rome (0.04 ± 0.03 t h$^{-1}$) and at a South-African landfill near Gqeberha (0.06 ± 0.04 t h$^{-1}$). The highest three site-averaged detected emission rates are found at the Norte III landfill in Buenos Aires, Argentina (22.0 ± 1.9 t h$^{-1}$), at a landfill near Hong Kong, China (10.0 ± 2.7 t h$^{-1}$) and at a landfill near Tehran, Iran (9.4 ± 4.9 t h$^{-1}$). Averaged emission rates have a median relative uncertainty of 45%, which accounts for both single observation and sampling uncertainties, calculated consistently across all sites (Methods). Using satellite and aerial imagery from Google Earth, we manually classify the 151 waste disposal sites into two categories: 108 managed landfills (sites with

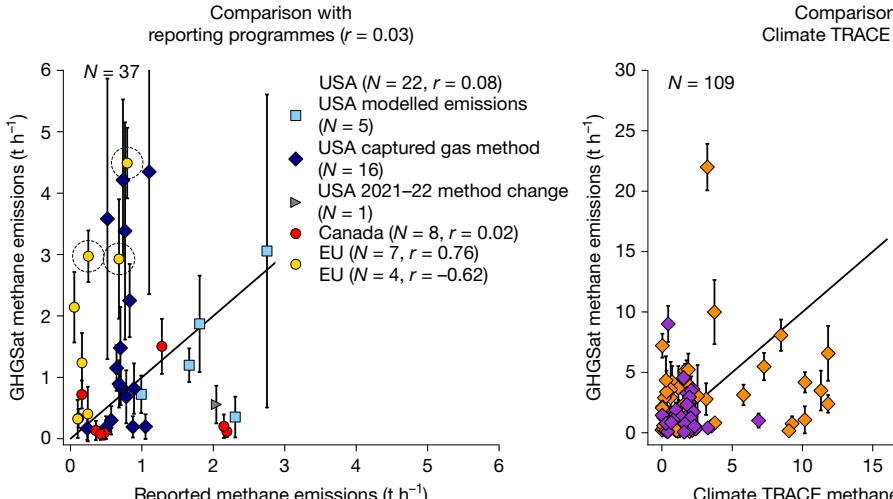

**Fig. 3 | Discrepancy between satellite-based and bottom-up emission estimates.** Comparison of site-wise methane emission rates observed by GHGSat against data included in reporting programmes (left) and emissions calculated by the Climate TRACE non-profit (right), both averaged over the corresponding GHGSat observation years. Reported and Climate TRACE data are provided as annual totals and have been converted to hourly rates assuming constant emission. Error bars show the site-wise-averaged GHGSat emission uncertainty. The one-to-one line is shown in black. The sites marked by dashed-circles in both panels drive the high correlations for EU site reports ($r = 0.76$) and Climate TRACE dumping sites ($r = 0.54$). If removed, these correlations drop to $r = -0.62$ and $r = -0.15$, respectively.

organized structures for burying waste, for example, featuring covers) and 43 dumping sites (with informal gathering of waste). Within this dataset, managed landfills and dumping sites do not show a statistically significant differennce in the total detected emission rate distributions. However, when normalized by the total site area, managed landfills show significantly lower area-normalized emission rates compared with dumping sites, thus demonstrating the expected effects of emissions mitigation by closing and covering modules of the landfill (Extended Data Fig. 6 and Supplementary Note 6). Overall, the distribution of site-wise averaged detected emissions is heavy-tailed, with the 60 (40%) strongest-emitting sites (47 managed landfills and 13 dumping sites) accounting for 80% of total emissions (Supplementary Note 4). This estimated skewness is probably conservative as the 100 kg h$^{-1}$ detection threshold and selective targeting of GHGSat would limit the inclusion of low-emitting sites. This detection threshold enables to cover 54% of the sites (assuming sufficiently localized emissions sources) included in the facility-scale waste disposal site emission database compiled by Climate Tracking Real-Time Atmospheric Carbon Emissions (Climate TRACE)[40], but these 54% of sites amount to 96% of total emissions (Extended Data Fig. 7 and Supplementary Note 4). Overall, the 151 waste disposal sites observed here represent a small fraction of the global total number of landfills (over 10,000 are included in the Climate TRACE datasets[40]) but, assuming constant emissions, their collective instantaneous emission rate scales up to a yearly total of 2.8 million tonnes. This corresponds to 7.4% of 2022 global solid waste emissions in version 8 of the Emissions Database for Global Atmospheric Research (EDGAR) inventory[2].

## Modelled and GHGSat-based rates disagree

Figure 3 compares facility-level GHGSat-detected methane emission rates against national site-level reporting programs[41–43] (which are based on process-based models or gas capture efficiency assumptions) and emissions obtained from data-driven models developed by the non-profit coalition, Climate TRACE[40]. National reporting data exclusively cover managed landfills in the United States, Canada and some EU countries, whereas Climate TRACE has more global coverage and includes dumping sites (Extended Data Fig. 8 and Supplementary Note 7). Overall, we find no correlation between satellite-based and reported or modelled estimates ($r = 0.03$ for reported emissions, and $r = 0.18$ for Climate TRACE), with differences showing an insignificant

bias and a large scatter, exceeding the averaged emission rates (Supplementary Note 8). Analysing managed landfills and dumping sites separately does not change this conclusion. Although no overall bias is found between reported and GHGSat-based estimates, emissions from 14 (out of 37) landfills are at least twice as large as what is reported to national programmes. As the US Greenhouse Gas Reporting Program includes reports based on two different methodologies—one based on gas capture efficiency and the other based on waste decay modelling—we can compare our results for the US to both (Fig. 3 separates US sites depending on which reporting method was chosen by the facilities). Comparing both estimates for all US landfills to GHGSat-detected emissions (Extended Data Fig. 9 and Supplementary Note 9), we observe that the approach based on gas capture efficiency tends to underestimate emissions (by a factor two), whereas the approach based on waste decay modelling tends to overestimate them (by a factor 1.5). These US results are consistent with a recent investigation of landfill emission models used for reporting[44] and with aerial-based observations[39]. Neither method shows a strong correlation with our results. These findings highlight the critical importance of coordinating bottom-up modelling efforts with independent observations of landfill emissions to improve the understanding of facility-scale waste emissions.

## Plume sources relate to site activity

Figure 1 shows that high-resolution observations also allow us to pinpoint where detected emissions originate from within a solid waste disposal facility. To understand these origins, we compare manually verified GHGSat emission plume origins with surface activity detected from clear-sky Sentinel-2 10-m resolution RGB observations (Methods and Supplementary Note 10). A landfill near Casablanca (Morocco; Fig. 1l–o) is a clear example, as both GHGSat plume sources and landfill surface activity show north-to-south migration as time progresses and a new section of the landfill is developed in the southwest. Across 107 facilities that have a sufficient number of clear-sky Sentinel-2 images and high-quality surface activity detection results, we find that 44 (41%) show a statistically significant proximity (P-value < 0.05) between surface landfill activity and GHGSat plume source location. When considering only the 21 sites with at least 16 identifiable plume origins in GHGSat observations, we find statistically significant proximity for 18 (86%) of them (Extended Data Fig. 10). Upon revisiting our

dataset showing that total site area-normalized emission rates are significantly lower for managed landfills compared with dumping sites (Extended Data Fig. 6 and above), we conclude that the small fraction of open active areas in managed landfill accounts for almost all of the emissions detected by GHGSat. This highlights the predominant role open modules have in managed landfill emissions. This result is consistent with reports of methane emissions being observed originating from landfill work faces in on-ground and satellite-based studies for a limited number of sites[5,45], and with an extensive aerial survey covering the United States that showed the prevalence of work faces in total landfill emissions[39,46]. Our dataset shows the active surface is the dominant emission source across management and economic development levels. This emphasizes the need to quantify emissions from the active surface, underscoring the importance of repeated observations to both reliably estimate mean emissions and to narrow down on (potentially migrating) source locations within a landfill. This spatial information can for example help site operators focus mitigation efforts more effectively. Our dataset also includes two example plumes originating from adjacent facilities: a biogas plant near the Las Dehesas landfill near Madrid and from a wastewater treatment plant near Shanghai (both filtered from the analysis; Supplementary Note 11). They illustrate the mitigation potential that satellite observation can detect in facilities related to and neighbouring waste disposal sites.

Our survey has extensive spatial coverage that brings top-down observation-based estimates of methane emissions for 151 waste disposal sites across six continents. It sheds new light on the mitigation potential of urban methane emissions and on the ability of high-resolution satellites to monitor methane emissions from waste disposal sites and support mitigation activities by pinpointing emission sources within the facility, highlighting the importance of the active surface. The availability of such high-resolution methane-sensitive satellite observations is currently increasing, with the expanding GHGSat constellation and new initiatives such as Carbon Mapper's Tanager-1 satellite, as well as public hyperspectral satellite missions[3]. Across the 151 surveyed sites—assuming constant emissions to scale up the snapshot averages provided by satellites—we find that bottom-up and top-down satellite-based solid waste emission estimates cannot currently be reconciled at facility scale. This disagreement is consistent with past facility-scale studies using aerial measurements[39] and country-scale studies using TROPOMI data[47]. These discrepancies highlight the importance of site-level data and practices, and call for further efforts that focus on both managed landfills and dumping sites, aiming to close the gap between current bottom-up and top-down understandings of methane emissions from solid waste. Ideally, such studies would involve partners operating waste disposal sites, bottom-up modellers, aerial and satellite-based methane observations augmented by complementary on-ground observations that can provide continuous measurements, including at night. An improved understanding of site-level solid waste methane emissions can support more effective emission mitigation strategies contributing to the worldwide efforts against climate change.

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

## Methods

### Automatic methane plume detection in TROPOMI data

TROPOMI[51] on board the European satellite Sentinel-5 Precursor was launched in 2017. It observes backscattered sunlight in the near- and shortwave infrared around the 0.76 μm $O_2$ and 2.3 μm methane bands, at approximately 1:30 pm local time. Total columns (vertically integrated concentrations) of methane with near vertically uniform sensitivity down to the surface are retrieved from these observations using a full-physics approach that accounts for the interfering impact of surface reflectance, aerosol and other geophysical variables on the shortwave infrared signal (v.2.6.0)[34]. TROPOMI is a methane flux mapper that offers daily global coverage with a 7 × 5.5 km² spatial resolution at nadir. In addition to being used in long-term inverse analyses, its imaging capabilities enable the detection of anthropogenic methane emission plumes that arise from the world's largest emitters[29]. We employ a two-step machine learning approach to explore TROPOMI data for methane emission plumes automatically[4]. We analyse and manually verify all plumes detected in 2021 and 2022 with estimated sources within 50 km from any of the landfills targeted by GHGSat. We apply the Integrated Mass Enhancement (IME) method[20], calibrated specifically for TROPOMI using atmospheric transport simulations, to quantify the methane emission rate and its uncertainty for each TROPOMI-detected plume[4].

Given TROPOMI's spatial resolution (7 × 5.5 km²) compared with GHGSat's (25 × 25 m²), we cluster the 151 landfills observed by GHGSat into 130 TROPOMI-relevant urban areas. For each urban area, we first apply a 2σ filter to remove outlier estimates that can be hampered by an unrepresentative plume mask due to variable meteorology or surface effects (for example, plume masks truncated by clouds for lower estimates). Then, relying on the remaining TROPOMI plume detections, we report their mean detected urban-scale methane emissions and their standard deviation. These averages only cover emissions detected as strong plumes and are not representative of mean urban emissions but do provide an indication of urban mitigation potential. Not detecting a plume does not imply that there are no emissions: it means that concentrated emissions are lower than the ~8 t h⁻¹ TROPOMI plume detection threshold, or that the observational or geographical conditions did not allow for TROPOMI detection[4]. The discrepancy with mean emissions is verified for four different (above-average emitting and often detected) cities (Buenos Aires, Delhi, Mumbai and Lahore) where IME-based rates show a 7–47% overestimation (while agreeing within uncertainties) compared with urban-level methane emission estimates based on atmospheric inversions and TROPOMI data[5] (Supplementary Note 1).

### GHGSat observations and emission quantification

GHGSat-C1 to -C5 instruments were launched between 2020 and 2022. Satellites C1-2-3 perform measurements in the morning around 10 am (local time), while satellites C4-5 perform measurements in the afternoon around 2 pm (local time). These instruments estimate the total column (vertically integrated content) of methane at ~25 × 25 m² resolution over targeted 12 ×15 km² domains[52] from backscattered sunlight measurements in the shortwave infrared near 1.65 μm, that provide near-surface sensitivity. The GHGSat instruments have an empirically measured methane column precision range of 1.4–2.9%[53], which allows them to observe emission plumes from point (for example, a gas pipeline leak) or very localized sources (for example, active faces of landfills) emitting more than ~100 kg h⁻¹ (this detection threshold increases with wind speed)[28]. Pixels exhibiting local spatially correlated methane column enhancements above background are clustered together and considered to belong to a plume[26]. We apply the IME method[20] to estimate an emission rate $Q$ based on a delineated plume and the local wind speed sampled from a meteorological model. We have:

$$Q = \frac{U_{\text{eff}}}{L} \sum_i \Delta X_{\text{CH}_4,i} a_i$$

where $U_{\text{eff}}$ is the effective wind speed, calibrated against the 10-m wind speed based on a set of large Eddy simulations (LES)[5]; $L = \sqrt{\sum_i a_i}$ is the plume length computed as the square-root of the plume total area, where $a_i$ is the area of the $i$th pixel included in the plume; and $\Delta X_{\text{CH}_4,i}$ is the local enhancement above the background of the methane total column for this $i$th pixel. Here we use an effective wind speed calibration specific to landfills, based on LES of area sources: $U_{\text{eff}} = 0.34 \times U_{10\,\text{m}} + 0.66$ (ref. 5), where $U_{10\,\text{m}}$ is the 10-m wind speed sampled from the GEOS-FP meteorological reanalysis[54]. The emission rate uncertainty calculation includes contributions from (1) wind speed error; (2) methane column retrieval error; and (3) IME calibration error[26].

The calibration of this mass-balance approach against LES of known synthetic emission rates ensures that the estimated rates correctly account for the different advective transport conditions explored within the set of LES. Beyond this calibration on simulations, numerous real-life validation efforts have been organized, including controlled-release experiments, which are the validation gold standard. Notably, GHGSat participated and showed excellent agreement with metered emission rates in internal controlled releases, as well as in two single-blind controlled-release campaigns, where the true emission rates (and wind speeds) are not known to the satellite data providers and the comparisons are done by a third party (in this case a research group from Stanford University)[25,55]. Beyond controlled releases, landfill emission rates obtained through aerial methane imagery with an instrument that can detect plumes down to 10 kg h⁻¹ have been validated against traditional aerial mass-balance results[18,39]. Besides, an in-depth study of two landfills near Madrid that included both similar airborne observations and GHGSat satellite observations showed that GHGSat satellite-based estimates match the total of airborne-detected plumes for same day observations within uncertainties[56]. Combined, these results show that GHGSat satellite-based observations can provide accurate estimates of methane emissions from waste disposal sites.

### Estimating site-level GHGSat averages

Three outcomes are possible for any individual waste disposal site observation during a single overpass by GHGSat: (1) no plume is detected; (2) only one plume is detected; and (3) several plumes (arising from the same site) are detected. In the first case, we conservatively consider the emission rate to be equal to zero, with no uncertainty. In the second and third cases, we apply the IME method to each plume separately to quantify its emission rate and uncertainty. In the third case, we sum together all of the detected plume emission rates (and sum their respective uncertainties quadratically) to obtain an emission rate for the whole site.

Given a set of observations for a waste disposal site, we employ a two-step random sampling approach to evaluate the site-level averaged emission rate and its uncertainty, accounting for both single-observation and sampling uncertainties. First, in a bootstrapping approach, we randomly ($N = 100{,}000$) resample our set of observations by randomly picking single observations with replacement. This enables us to generate an ensemble of averaged emission rates for which we also compute corresponding uncertainties assuming that single observations are independent Gaussian variables. We then sample a Gaussian distribution ($N = 1{,}000$) for all these ensemble elements relying on their respective rates and uncertainties. Finally, we report the mean and standard deviation across this two-step random sampling approach as averaged emission rate and its uncertainty. This method accounts for the single-observation uncertainties and is especially useful to handle bi- or multi-modal site-wise emission rate distributions that can have a peak at zero (all of the observations without any

detection) and one or several peaks for positive emission rate values (all of the observations with detected plumes).

## Comparison of GHGSat and reported or calculated emissions
For site-wise GHGSat-based methane emission rate comparison against site-wise reported values included within national reporting programs, we manually match sites based on addresses (no distance threshold is used). To compare site-wise GHGSat-based methane emission rates against values modelled by Climate TRACE, we only select sites for which we find matches within a 2 km distance of GHGSat targets, and then only consider the facilities within these 2 km that show the minimum distance from GHGSat targets (Supplementary Note 7 details the other data selection criteria specific to each dataset we compare with). Reported and Climate TRACE data are provided as annual totals and have been converted to hourly rates assuming constant emissions.

## Landfill surface activity detection from Sentinel-2 imagery
Managed landfills and dumping sites are active and constantly evolving as they accept new waste: they expand and their active surface(s) move(s) to accommodate the incoming waste. High-resolution visual imagery can be used to track the surface activities at waste disposal sites. To compare the spatio-temporal distributions of GHGSat-detected methane plumes origins and landfill activities, we devise an image analysis scheme to automatically detect surface activity from time series of clear-sky 10-m resolution Sentinel-2 satellite images of waste disposal sites.

For each of the 151 waste disposal sites observed by GHGSat, we convert the time series of Sentinel-2 clear-sky visual RGB images to greyscale by using the National Television Standard Committee's formula[57]:

$$Greyscale = 0.299 \times R + 0.587 \times G + 0.114 \times B$$

We then apply a three-image moving filter (over time) based on local structural analysis[58,59], which determines surface activity in a given image by detecting overlapping structural changes that occur between this image and the previous one, and between this image and the next one. Using manually outlined landfill masks based on the latest Google Earth imagery, we only consider surface activity that is detected within landfill boundaries, and use filters to ignore pixels associated with water, clouds or cloud shadows. We smooth the raw activity map with a median filter to remove spatially inconsistent noise and only keep spatially consistent activity clusters. Individual activity clusters are then identified, outlined with convex hulls and stored as surface activity results. For each landfill, surface activity results are manually verified before being included in the analysis (Supplementary Methods 1; illustrations are provided in Supplementary Note 10).

## Comparison of landfill surface activity results and GHGSat methane plume origins
The wind direction allows to estimate the plume origin as the most upwind highly enhanced pixel included in the plume mask. We also manually verify this result and pinpoint the approximate source(s) of all GHGSat plumes, allowing one to select multiple sources for overlapping plumes originating from the disposal site where appropriate. We use these source locations to compare to the Sentinel-2 based surface activity analysis.

For a given plume, we use the minimum distance between the manually determined plume origin and the nearest outline of a surface activity cluster detected in the closest-in-time Sentinel-2 image as the proximity metric. We set the metric to zero if a plume origin falls inside a detected activity cluster. Consequently, the lower the metric value, the closer the source is to a detected surface activity cluster. We also compute the same metric for $N = 10,000$ points randomly drawn within the landfill boundaries. This comparison is conservative because it is possible that GHGSat plumes show sources outside of landfill boundaries (their metric values have no upper boundary) whereas these random points can only be located inside (their metric values have an upper boundary).

For each site, we compute the averaged metric across all GHGSat-detected methane emission plumes and compare this result with the distribution of averaged metric values obtained for the $N = 10,000$ randomly drawn points. We then evaluate the $P$-value probability of randomly obtaining averaged metric values that are lower than the GHGSat-based result. We consider that GHGSat plume origins show a statistically significant proximity with detected landfill surface activity if we obtain a $P$-value lower than 0.05 (Supplementary Methods 2).

Supplementary Note 10 showcases examples from different landfills and present an overview of $P$-value results for all landfills where surface activity could be detected.

## Reporting summary
Further information on research design is available in the Nature Portfolio Reporting Summary linked to this article.

## Data availability
The Sentinel-5P TROPOMI data and Sentinel-2 data are available at the Copernicus Data Hub via https://dataspace.copernicus.eu. GEOS-FP wind data can be downloaded from https://gmao.gsfc.nasa.gov/GMAO_products/. ERA5 and GEOS-CF meteorological data were sampled using Google Earth Engine. The GHGSat-detected methane plumes are available on Zenodo via https://doi.org/10.5281/zenodo.16641834 (ref. 60). Tables summarizing site-level results for GHGSat, and urban-area-level results for TROPOMI, are available in Supplementary Notes 1–11.

## Code availability
The codes that have been developed to perform the analysis of detected plume statistics and landfill surface activity are available via Code Ocean at https://codeocean.com/capsule/2078268/tree. They enable the production of Figs. 1, 2 and 3 and the data instrumental to these figures.

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

**Acknowledgements** We thank the Global Methane Hub for funding the 'Targeting Waste Emissions Observed from Space—Phase 1' project. M.D. acknowledges funding from the GALES project (grant no. 15597) of the Dutch Technology Foundation STW-NWO. M.D. and A.L.N. acknowledge the NSO TROPOMI national programme. S.S. acknowledges funding from

the IMEO Studies programme contract DTIE22-EN5036. We thank the team that realized the TROPOMI instrument and its data products, consisting of the partnership between Airbus Defence and Space Netherlands, KNMI, SRON and TNO and commissioned by NSO and ESA. The Sentinel-5 Precursor and Sentinel-2 are part of the EU Copernicus programme, and Copernicus (modified) Sentinel-5P data (2021–2022) and Sentinel-2 data (2020–2023) have been used.

**Author contributions** M.D., J.D.M. and I.A. designed the study. M.D. performed the analysis and interpretation of TROPOMI and GHGSat methane plumes, as well as the Sentinel-2 activity detection analysis, under the supervision of J.D.M. and I.A. M.G., D.J. and J.M.K. selected the GHGSat methane plumes and contributed to their analysis and interpretation. D.J.V. performed and provided the effective wind speed calibration for landfills, and contributed to the result interpretation. B.J.S. performed the plume detection in TROPOMI data and contributed to their analysis. S.S. and A.L.N. contributed to the analysis of TROPOMI data. M.D. wrote this article with feedback from all co-authors.

**Competing interests** The authors declare no competing interests.

**Additional information**
**Correspondence and requests for materials** should be addressed to Matthieu Dogniaux.

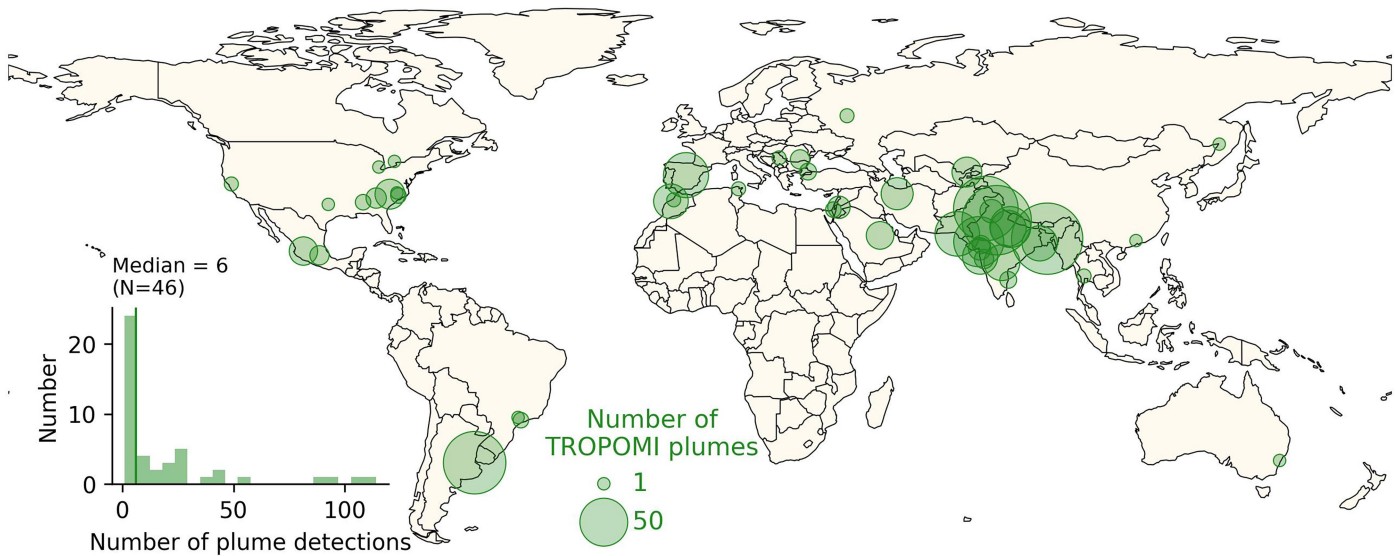

**Extended Data Fig. 1 | Map of the number of urban-scale methane emission plumes detected in TROPOMI data.** Spatial distribution of urban-scale methane emission plume numbers detected in TROPOMI data for the 46 urban areas targeted by GHGSat that show methane emission plumes in 2021–2022 TROPOMI data.

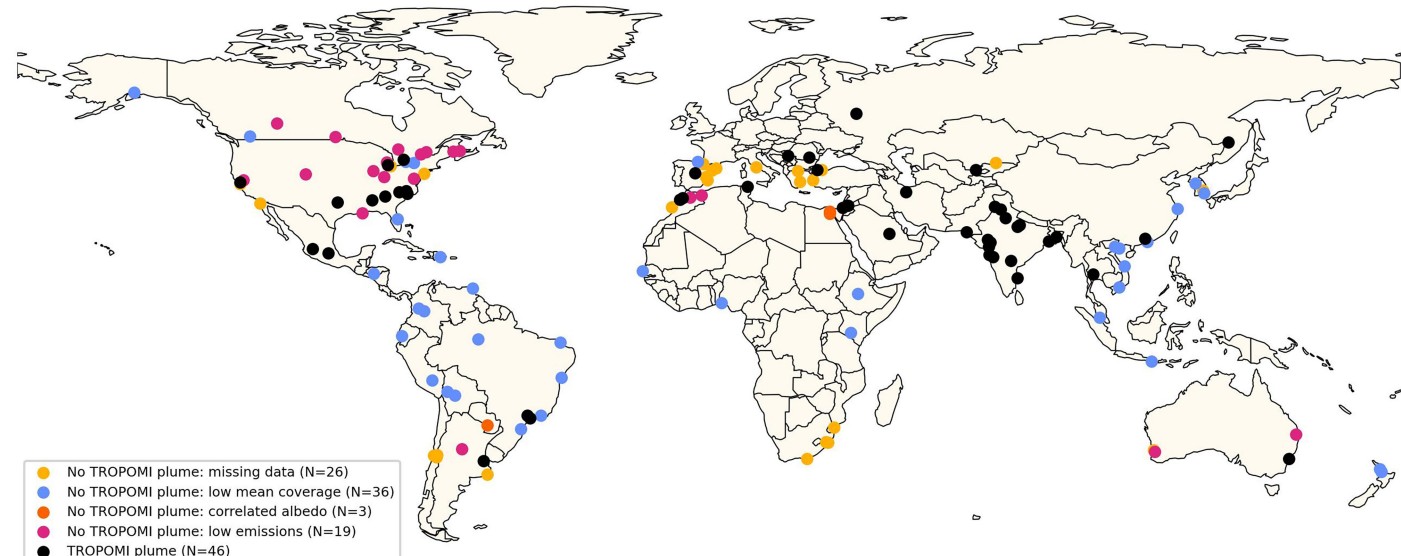

**Extended Data Fig. 2 | Prevailing explanation for the absence of plume detections in TROPOMI data.** Spatial distribution of all 130 urban areas observed by GHGSat. Urban areas that show methane emission plumes detected in TROPOMI data are depicted in black. Colored urban areas do not show methane emission plumes detected in TROPOMI data, and their color depicts the prevailing explanation of why that is the case. Overall, we find that the absence of a detected plume can be explained by observational-coverage-related issues for 62 urban areas. Expected urban-scale methane emissions lower than the TROPOMI plume detection threshold explain the absence of plume detections for 19 urban areas and, finally, strong TROPOMI methane column correlation with surface reflectance features hampering plume detection explains the absence of plume detections for 3 urban areas.

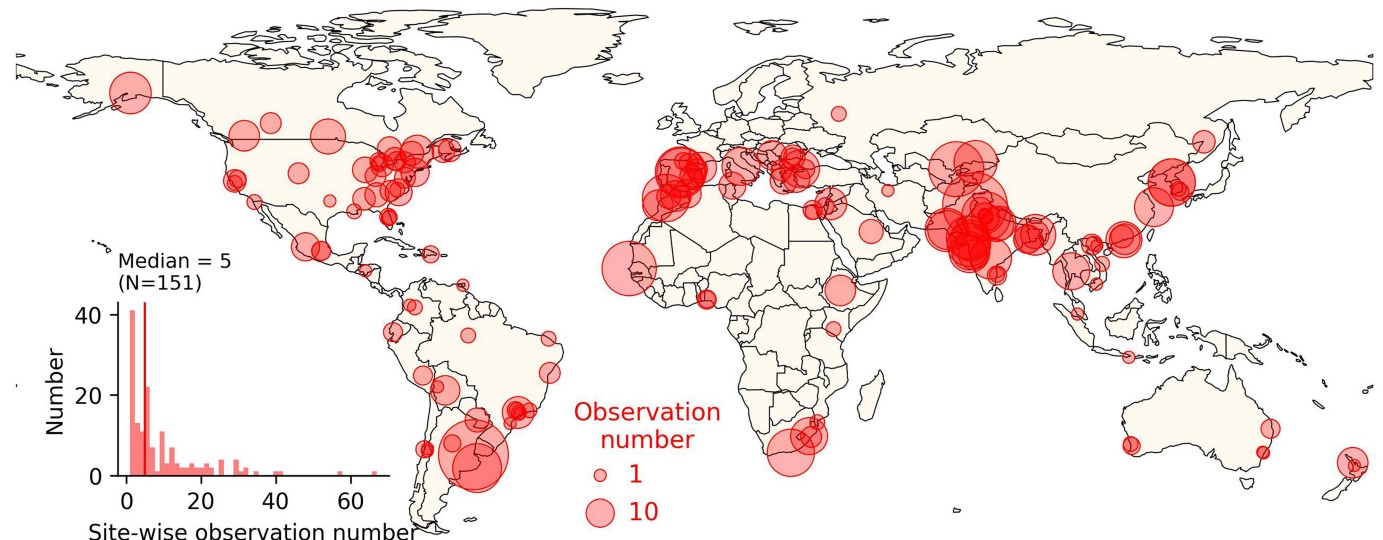

**Extended Data Fig. 3 | Map of the number of GHGSat observations per site.** Spatial and site-wise observation count distributions of the 151 waste disposal sites observed by GHGSat satellites. All sites have at least one plume detection.

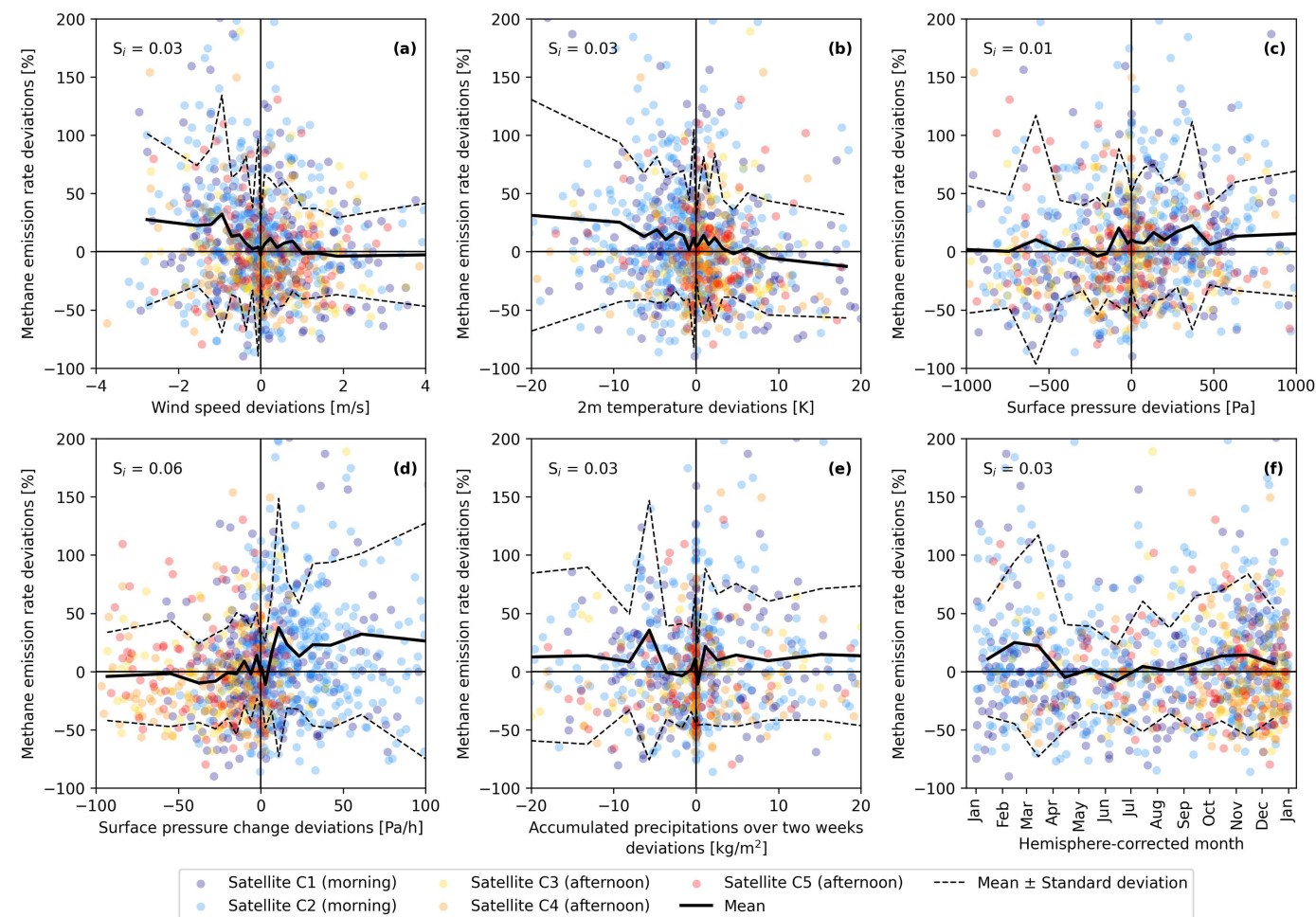

**Extended Data Fig. 4 | Comparison of methane emission rate variations against meteorological variables.** Methane emission rate deviations from site-wise medians (computed excluding null detections) against deviations from site-wise medians (computed excluding null detections) for wind speed (a), 2 m air temperature (b), surface pressure (c), change in surface pressure over 1 h (d), accumulated precipitations over two weeks (e), and month of the year (corrected for hemisphere). All meteorological data are sampled from ERA5. Smoothed mean curves are shown (thick black lines) and are used to compute first-order sensitivity indices $S_i$, providing the dataset variance fraction explained by the considered meteorological variable. Smoothed mean curves ± local standard deviations are also shown (thin dashed lines). Overall, no meteorological variable significantly explains site-wise temporal emission variability in our dataset.

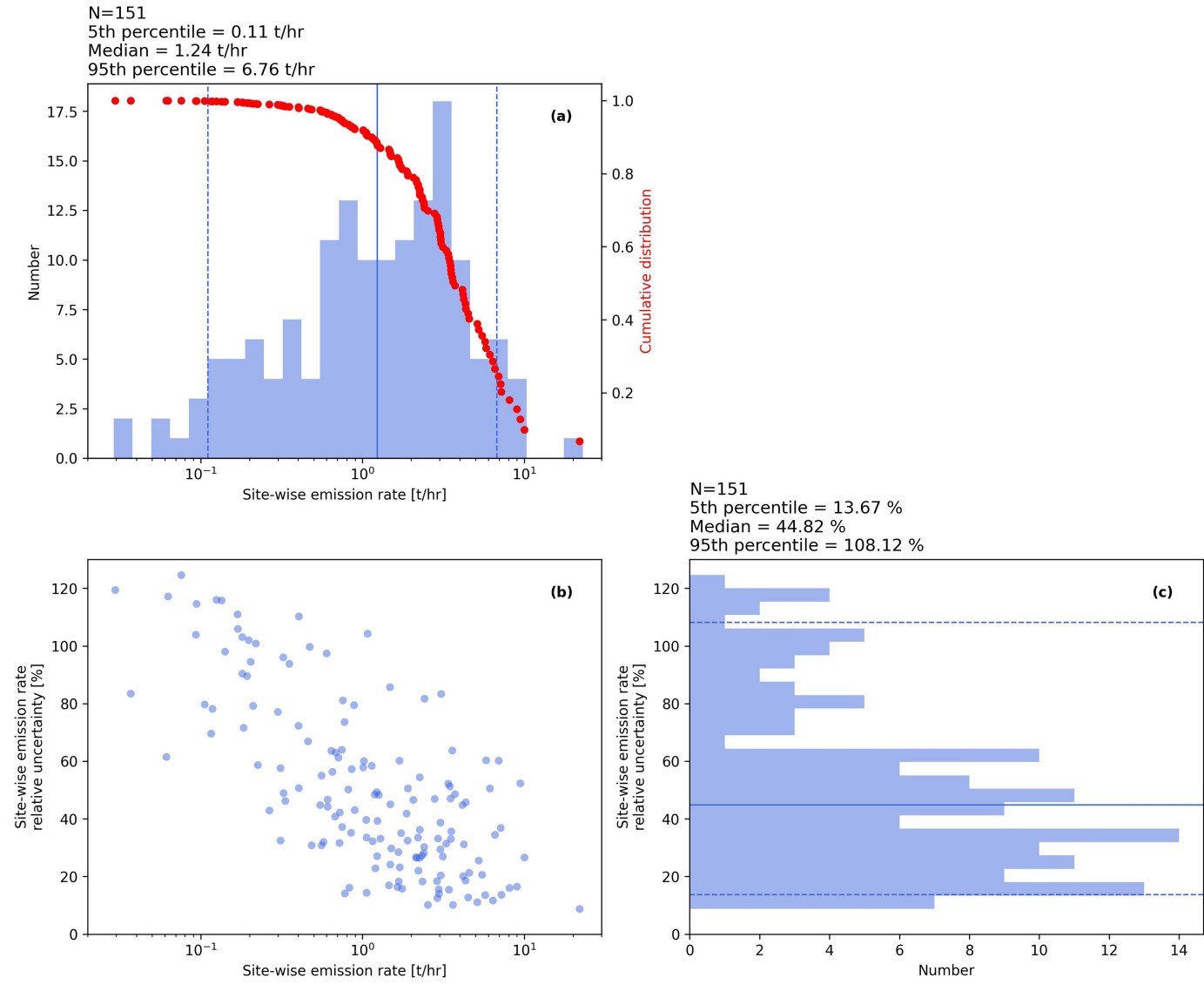

**Extended Data Fig. 5 | GHGSat site-wise emission rate and uncertainty distributions.** GHGSat site-wise averaged methane emission rate (a) and relative uncertainty (c) distributions, with their relationship (b). Relative uncertainties above 100% are related to sites showing one positive detection and a larger number of negative detections.

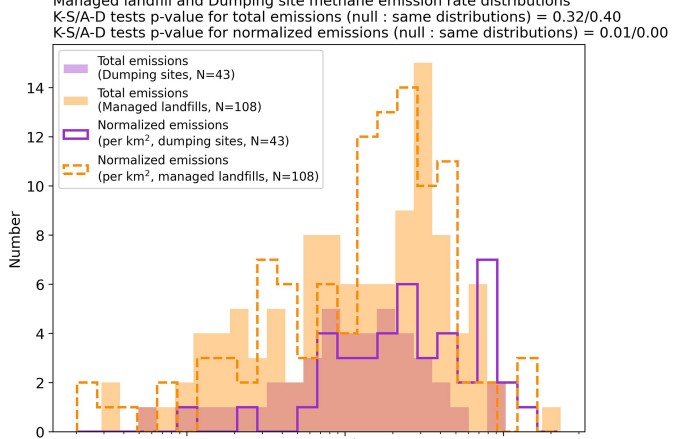

Managed landfill and Dumping site methane emission rate distributions
K-S/A-D tests p-value for total emissions (null : same distributions) = 0.32/0.40
K-S/A-D tests p-value for normalized emissions (null : same distributions) = 0.01/0.00

**Extended Data Fig. 6 | Comparison of managed landfill and dumping site total and area-normalized emission distributions.** Distributions of total (full colors) and area-normalized (lines) methane emission rates for managed landfills (orange) and dumping sites (purple). Two-sided two-sample Kolmogorov-Smirnov (K-S) and a two-sample Anderson-Darling (A-D) tests are performed to test whether distributions are significantly different between managed landfills and dumping sites. Total site-wise emission distributions are not significantly different, but area-normalized emission distributions are, with lower (on average) area-normalized emissions in managed landfills compared to dumping sites. This may be explained by managed landfills including closed inactive modules that generally show no emissions above the GHGSat detection threshold but still add to the total site area whereas, by definition, dumping sites do not show these closed inactive modules. This reflects the expected effect of definitively covering and closing some parts of managed landfills, thus confirming the efficiency of this mitigation strategy.

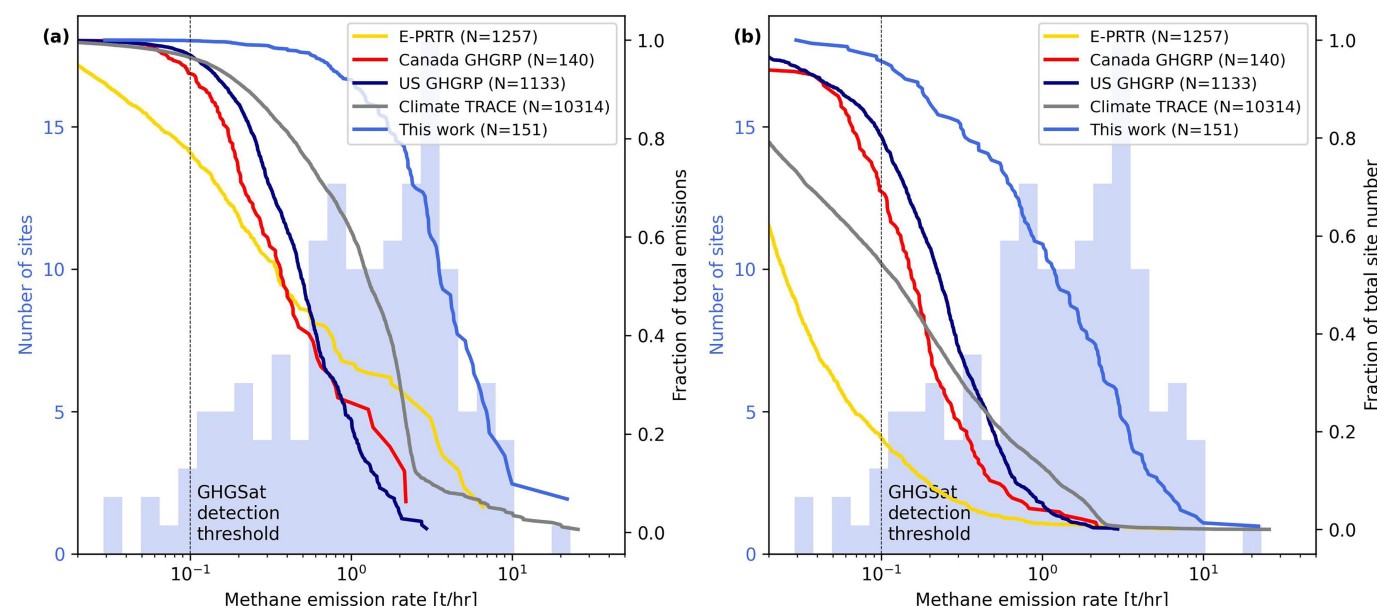

**Extended Data Fig. 7 | Comparison of cumulative distribution functions between this work and bottom-up datasets.** Distribution of emission rates for the 151 sites included in this study (left y-axes) compared to the cumulative distribution functions (right y-axes) for total emissions (a) and total number of sites (b) computed for all the sites included in the bottom-up facility-scale datasets considered in this work (see Supplementary Note 7) and for our sample of sites. The sites studied in this work and the GHGSat detection threshold cover the emission rate range that contributes most to total solid waste methane emissions. This highlights the potential of high-resolution methane imaging satellites to monitor solid waste methane emissions globally.

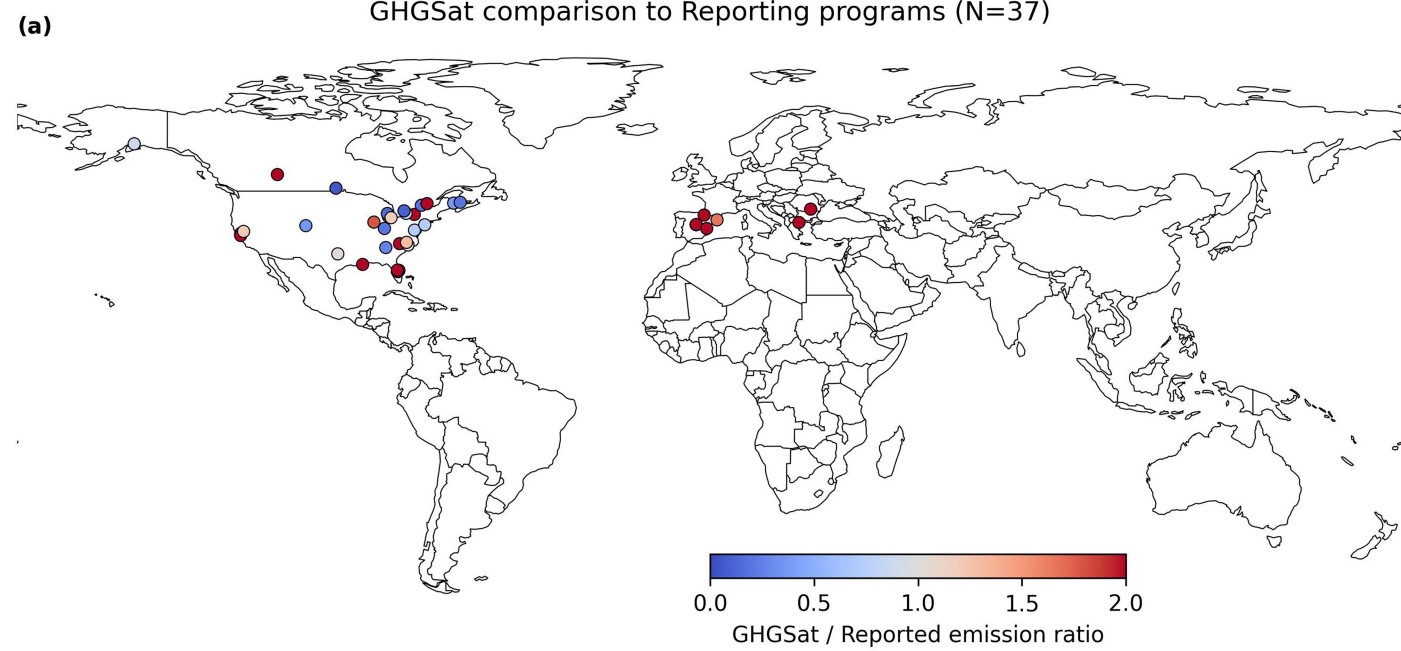

**(a)** GHGSat comparison to Reporting programs (N=37)

GHGSat / Reported emission ratio

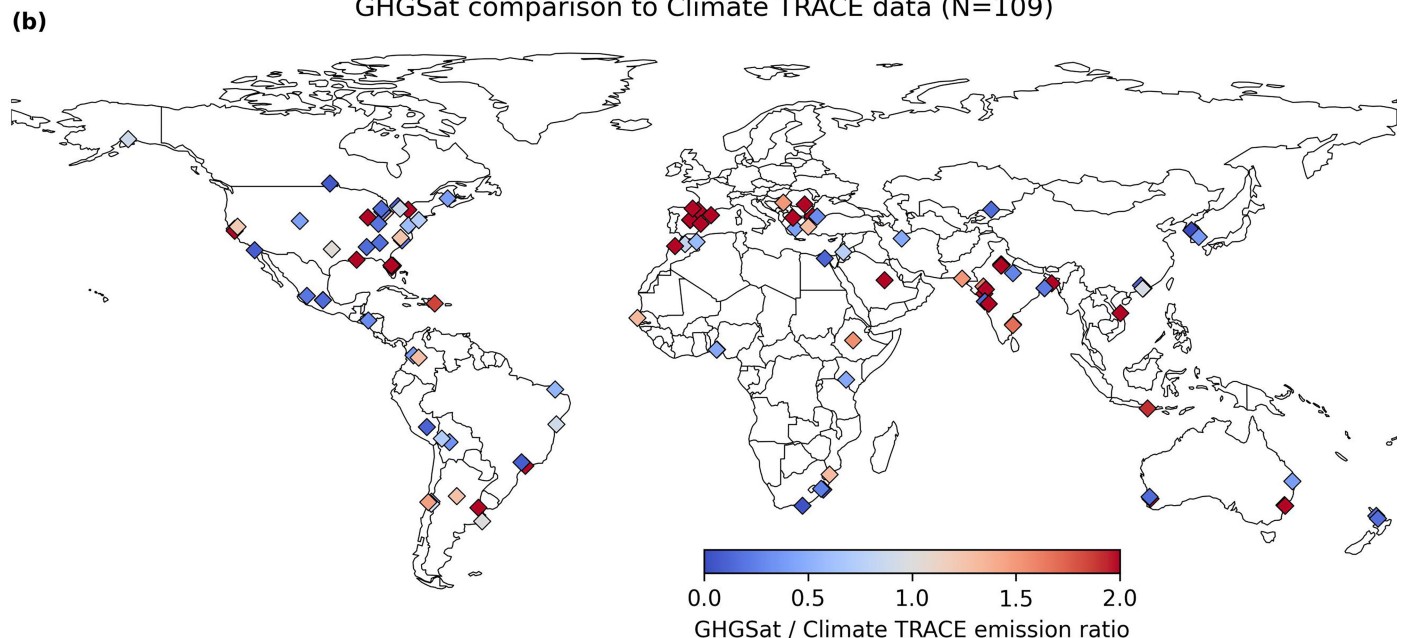

**(b)** GHGSat comparison to Climate TRACE data (N=109)

GHGSat / Climate TRACE emission ratio

**Extended Data Fig. 8 | Spatial distribution of waste disposal sites for which bottom-up estimates are available.** Spatial distribution of site-wise comparison between GHGSat methane emission rates and reported (a) or calculated (b, Climate TRACE) emissions. These are the sites included in Fig. 3.

# Article

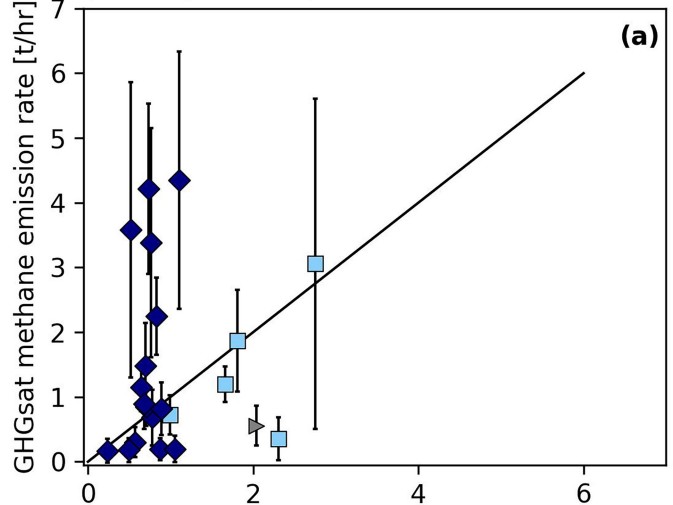

Reported to GHGRP
GHGSat - GHGRP = 0.42 ± 1.44 t/hr
GHGSat mean/GHGRP mean = 1.40
(N=22, r=0.08)

Modelled emissions
Difference = -0.46 ± 0.79 t/hr (N=5, r=0.56)
Captured gas method
Difference = 0.82 ± 1.42 t/hr (N=16, r=0.26)
2021-22 Method change
Difference = -1.49 t/hr (N=1)

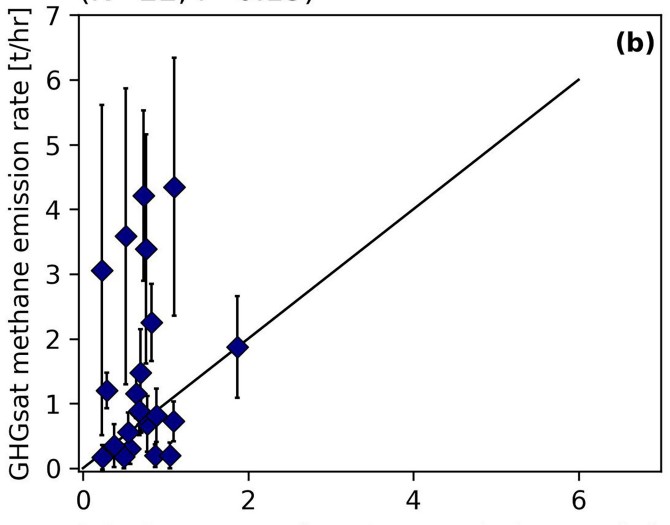

Captured gas method
GHGSat - GHGRP = 0.75 ± 1.35 t/hr
GHGSat mean/GHGRP mean = 2.03
(N=22, r=0.13)

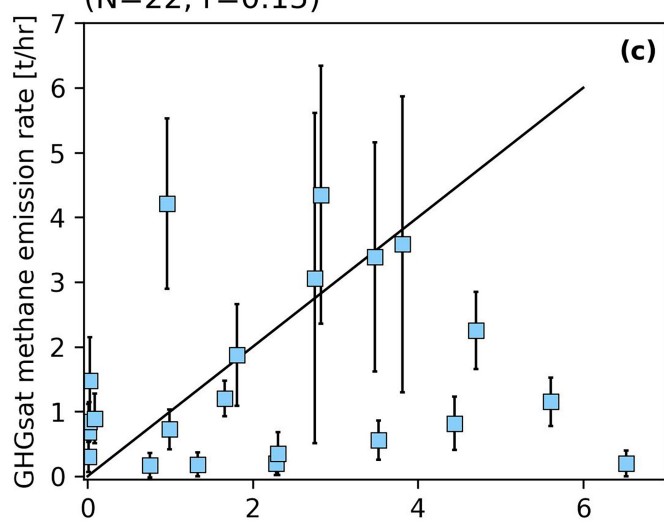

Modelled emissions
GHGSat - GHGRP = -0.79 ± 2.14 t/hr
GHGSat mean/GHGRP mean = 0.65
(N=22, r=0.15)

**Extended Data Fig. 9 | Impact of US GHGRP reporting method on the comparison against GHGSat-based emission rates.** GHGSat-based emission rates compared to annual US GHGRP reported emissions rates averaged over the corresponding GHGSat observation years, and obtained from the official GHGRP dataset (a), GHGRP results of the gas-capture efficiency method for all landfills (b), and GHGRP results of the waste-decay modelling for all landfills (c).

Reported data are provided as annual totals and have been converted to hourly rates assuming constant emissions. Black lines show the 1:1 line. Waste decay modelling provides higher emission estimates compared to GHGSat-based emission rates, while the gas-capture efficiency method provides lower estimates. Regardless of the method, reported estimates do not correlate with GHGSat-based emission rates.

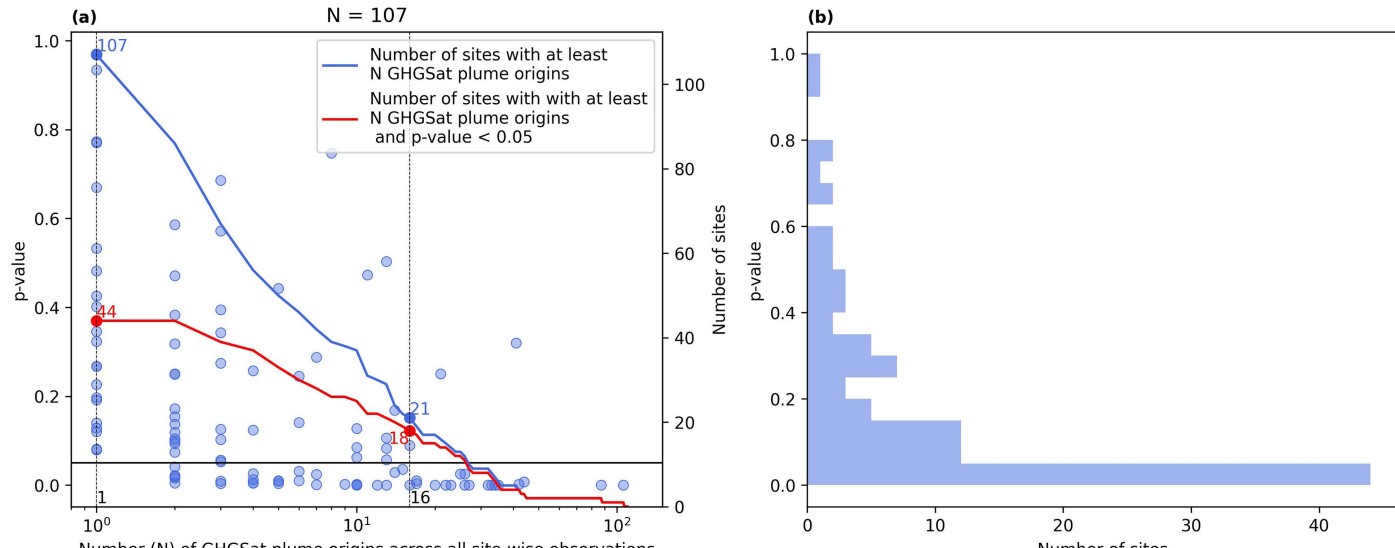

**Extended Data Fig. 10 | Summary of GHGSat plume source comparison to Sentinel-2 detected surface activity.** Proximity significance p-value for the 107 sites that passed all Sentinel-2 detected surface activity filtering criteria against the number of plume sources identified per site (a, left y-axis), and total number of sites that show at least a given number of plume sources (a, right y-axis, blue line) and that also show a p-value < 0.05 (a, right y-axis, red line). Distribution of proximity significance p-value values (b). We find that 44 sites show statistically significant proximity between Sentinel-2 detected surface activity and GHGSat plume sources, with an increasing fraction of sites showing such proximity for an increasing number of GHGSat observations.

# Reporting Summary

## Statistics

For all statistical analyses, confirm that the following items are present in the figure legend, table legend, main text, or Methods section.

| n/a | Confirmed | |
|---|---|---|
| ☐ | ☒ | The exact sample size (*n*) for each experimental group/condition, given as a discrete number and unit of measurement |
| ☒ | ☐ | A statement on whether measurements were taken from distinct samples or whether the same sample was measured repeatedly |
| ☐ | ☒ | The statistical test(s) used AND whether they are one- or two-sided<br>*Only common tests should be described solely by name; describe more complex techniques in the Methods section.* |
| ☒ | ☐ | A description of all covariates tested |
| ☒ | ☐ | A description of any assumptions or corrections, such as tests of normality and adjustment for multiple comparisons |
| ☐ | ☒ | A full description of the statistical parameters including central tendency (e.g. means) or other basic estimates (e.g. regression coefficient) AND variation (e.g. standard deviation) or associated estimates of uncertainty (e.g. confidence intervals) |
| ☐ | ☒ | For null hypothesis testing, the test statistic (e.g. *F*, *t*, *r*) with confidence intervals, effect sizes, degrees of freedom and *P* value noted<br>*Give P values as exact values whenever suitable.* |
| ☒ | ☐ | For Bayesian analysis, information on the choice of priors and Markov chain Monte Carlo settings |
| ☒ | ☐ | For hierarchical and complex designs, identification of the appropriate level for tests and full reporting of outcomes |
| ☐ | ☒ | Estimates of effect sizes (e.g. Cohen's *d*, Pearson's *r*), indicating how they were calculated |

*Our web collection on statistics for biologists contains articles on many of the points above.*

## Software and code

Policy information about availability of computer code

| Data collection | We use the well-established L2 (concentration) fields from ESA (Lorente et al., 2021) and GHGSat (Jervis et al., 2021) as starting point for the study<br><br>Lorente, A. et al.: Methane retrieved from TROPOMI: improvement of the data product and validation of the first 2 years of measurements, Atmos. Meas. Tech., 14, 665–684, https://doi.org/10.5194/amt-14-665-2021, 2021.<br><br>Jervis, D. et al.: The GHGSat-D imaging spectrometer, Atmos. Meas. Tech., 14, 2127–2140, https://doi.org/10.5194/amt-14-2127-2021, 2021 |
|---|---|
| Data analysis | Custom code was used, available on Code Ocean: https://codeocean.com/capsule/2078268/tree |

For manuscripts utilizing custom algorithms or software that are central to the research but not yet described in published literature, software must be made available to editors and reviewers. We strongly encourage code deposition in a community repository (e.g. GitHub). See the Nature Portfolio guidelines for submitting code & software for further information.

## Data

Policy information about availability of data

All manuscripts must include a data availability statement. This statement should provide the following information, where applicable:
- Accession codes, unique identifiers, or web links for publicly available datasets
- A description of any restrictions on data availability
- For clinical datasets or third party data, please ensure that the statement adheres to our policy

The Sentinel-5P TROPOMI data and Sentinel-2 data are available at the Copernicus Data Hub (https://dataspace.copernicus.eu). GEOS FP wind data can be downloaded from https://gmao.gsfc.nasa.gov/GMAO_products/. ERA5 and GEOS-CF meteorological data were sampled using Google Earth Engine. The GHGSat-detected methane plumes are available via Zenodo at: https://doi.org/10.5281/zenodo.16641834. Tables summarizing the results at site level for GHGSat, and at urban area level for TROPOMI are already included in the Supplementary Notes.

## Research involving human participants, their data, or biological material

Policy information about studies with human participants or human data. See also policy information about sex, gender (identity/presentation), and sexual orientation and race, ethnicity and racism.

| | |
|---|---|
| Reporting on sex and gender | Does not apply |
| Reporting on race, ethnicity, or other socially relevant groupings | Does not apply |
| Population characteristics | Does not apply |
| Recruitment | Does not apply |
| Ethics oversight | Does not apply |

Note that full information on the approval of the study protocol must also be provided in the manuscript.

# Field-specific reporting

Please select the one below that is the best fit for your research. If you are not sure, read the appropriate sections before making your selection.

☐ Life sciences  ☐ Behavioural & social sciences  ☒ Ecological, evolutionary & environmental sciences

For a reference copy of the document with all sections, see nature.com/documents/nr-reporting-summary-flat.pdf

# Ecological, evolutionary & environmental sciences study design

All studies must disclose on these points even when the disclosure is negative.

| | |
|---|---|
| Study description | Satellite survey of landfill methane emissions and associated urban methane emissions |
| Research sample | 151 waste disposal sites located across all inhabited continents for which successful GHGSat observations of methane plumes were available. Associated TROPOMI analysis of data over those areas |
| Sampling strategy | TROPOMI uses a continuous global sampling strategy while GHGSat performs targeted observations |
| Data collection | Satellite observation |
| Timing and spatial scale | Global scale observation set over 2021 and 2022 |
| Data exclusions | For TROPOMI we only use cloud-free observations to detect urban plumes using an automated approach as described in Schuit et al. (2023). For GHGSat, only clear-sky observations suitable for emission quantification were used<br><br>Schuit, B. J. et al.: Automated detection and monitoring of methane super-emitters using satellite data, Atmos. Chem. Phys., 23, 9071–9098, https://doi.org/10.5194/acp-23-9071-2023, 2023 |
| Reproducibility | The Code Ocean repository allows to reproduce the analysis |
| Randomization | Does not apply |
| Blinding | Does not apply |

Did the study involve field work? ☐ Yes ☒ No

# Reporting for specific materials, systems and methods

We require information from authors about some types of materials, experimental systems and methods used in many studies. Here, indicate whether each material, system or method listed is relevant to your study. If you are not sure if a list item applies to your research, read the appropriate section before selecting a response.

## Materials & experimental systems

| n/a | Involved in the study |
|-----|----------------------|
| ☒ ☐ | Antibodies |
| ☒ ☐ | Eukaryotic cell lines |
| ☒ ☐ | Palaeontology and archaeology |
| ☒ ☐ | Animals and other organisms |
| ☒ ☐ | Clinical data |
| ☒ ☐ | Dual use research of concern |
| ☒ ☐ | Plants |

## Methods

| n/a | Involved in the study |
|-----|----------------------|
| ☒ ☐ | ChIP-seq |
| ☒ ☐ | Flow cytometry |
| ☒ ☐ | MRI-based neuroimaging |

## Plants

**Seed stocks**

*Report on the source of all seed stocks or other plant material used. If applicable, state the seed stock centre and catalogue number. If plant specimens were collected from the field, describe the collection location, date and sampling procedures.*

**Novel plant genotypes**

*Describe the methods by which all novel plant genotypes were produced. This includes those generated by transgenic approaches, gene editing, chemical/radiation-based mutagenesis and hybridization. For transgenic lines, describe the transformation method, the number of independent lines analyzed and the generation upon which experiments were performed. For gene-edited lines, describe the editor used, the endogenous sequence targeted for editing, the targeting guide RNA sequence (if applicable) and how the editor was applied.*

**Authentication**

*Describe any authentication procedures for each seed stock used or novel genotype generated. Describe any experiments used to assess the effect of a mutation and, where applicable, how potential secondary effects (e.g. second site T-DNA insertions, mosiacism, off-target gene editing) were examined.*

