## [Peer Review File · Nature]

Actionable waste methane mitigation insights from a global satellite survey

Corresponding Author: Dr Matthieu Dogniaux

Version 0:

Reviewer comments:

Referee #1

(Remarks to the Author)

While satellite data offers promising insights into site-specific methane emissions globally, we face several challenges in fully utilizing these data.

First, the reliability of the data is a concern. This includes the accuracy of translating meteorological parameters into actionable information and effectively filtering out noise.

Second, how can we generate continuous temporal profiles from satellite data? This is crucial for assessing the feasibility of employing satellites for long-term monitoring and global data sharing.

Third, how do we integrate satellite data with ground-based observations? Based on the current manuscript, I find reconciling the two sources challenging.

Looking at Fig. 2, I noticed that the site and emission sources don't perfectly overlap, which suggests that wind might play a role in the discrepancy. Have you considered the wind direction at the time of the scan? If the wind is blowing in the opposite direction, there could be emissions from sources beyond the landfill that need to be excluded. Although the authors mention this in lines 226–228, there are still recognized facilities contributing to the emissions. Were there instances where emissions occurred from unidentified sources near the landfills? This raises questions about the uncertainties inherent in satellite data. Given its limited application in this field, I am hesitant to fully trust the data without further validation. However, applications of satellite data from other fields might help build that trust.

Additionally, where can we locate the centers of these emissions? The map shows several hotspots of concentrated emissions, some of which are outside the landfill boundary. I speculate this could be due to vertical variations in the methane plume. Satellite scans capture only 2D information, but assessing emission volumes requires 3D data. How can we address this limitation to obtain accurate results?

The manuscript also presents limited data from Africa, Australia, northern North America, and northern China. Is there a reason for this?

Moreover, how can we validate data collected during limited detection periods? Methane emissions from landfills are highly dependent on temperature and moisture, which vary seasonally. This calls for time-series profiles to better understand the emissions. How do you plan to address this challenge?

Lastly, I understand there is a less-than-linear correlation between GHGSat observations and reported methane emissions/Climate TRACE methane emissions. What about the correlation between reported methane emissions and Climate TRACE data for the same sites? Is it possible that the dataset is too small to establish meaningful correlations between satellite and ground-based observations? Beyond this, satellite observations should be linked to specific times and landfill stages to strengthen correlations with on-land data.

Comments on the waste management aspect:

As we navigate a complex landscape of models and discrete tests on methane emissions from global landfills, satellite remote sensing technology offers a new perspective that could provide a clearer and more comprehensive view. This manuscript represents a potential breakthrough in remote sensing applications for solid waste management, demonstrating

a high-tech, cross-disciplinary innovation. The results, while preliminary in terms of both completeness and temporal coverage, signal a paradigm shift. This approach has the potential to move landfill monitoring from isolated, sporadic measurements to large-scale, continuous surveillance, much like the role meteorological satellites play in weather forecasting.

However, for this technology to gain widespread acceptance, it must demonstrate its credibility and cost-effectiveness. As of now, significant barriers remain, particularly regarding the high costs associated with satellite scanning. This financial hurdle is especially critical for low- and middle-income countries, where the majority of the world's landfills and dumpsites are located. Given that climate change is a global issue, this scenario presents an opportunity for regional cooperation, with potential for knowledge transfer and data-sharing that could enhance the adoption of satellite monitoring technologies. Yet, we must acknowledge that the validation of satellite data is still in its infancy. The reliability of these measurements requires time, with further testing, validation, and refinement of both the scanning equipment and inversion methods. Only through extensive use and iterative improvements will we be able to fully assess the validity and accuracy of this technique. Returning to the core focus of the paper, satellite-based monitoring of methane emissions could reveal large-scale, dynamic changes over time, offering new possibilities for real-time tracking of various landfill attributes. For instance, this technology could potentially detect accidental chemical leaks, allowing for swift intervention before they escalate into major environmental incidents.

Looking beyond the specifics of satellite monitoring, an essential question emerges: how should we balance old and new technologies in environmental science? Despite the promise of new methods, traditional techniques will remain crucial for the foreseeable future. The coexistence of multiple technologies is not a contradiction, but a testament to the diversity of tools available to address complex challenges in waste management. This diversity ensures that technological innovation continues to advance in the sector while maintaining a foundation of proven methodologies.

(Remarks on code availability)

Referee #2

(Remarks to the Author)
General comments

This manuscript examines methane emissions from landfills across the globe using satellite remote sensing. Two satellites and instruments are used, the TROPOMI instrument onboard Sentinel 5P, which maps methane globally, and GHGSat. Significant methane emissions from landfills are observed globally and point to targets for methane emission mitigation. As such, this study presents new findings and the scope of the study is a significant advancement compared to existing published studies also using satellites to detect methane emissions from landfills.

In the study, the authors examine the emissions for two classes of landfills, "managed" versus "dumping sites" with the authors concluding that there is no significant difference in emission rates between these two types. This is a very important conclusion, which must be fully supported by the data. However, at present there are some remaining questions about this conclusion, which need to be addressed before the study can be accepted for publication.

Specific comments

L20-21: The statement that "managed fills and dumping sites show similar levels of emissions" is a very important one and needs to be fully supported by observations. However, I have some reservations about this statement:

- 1) This sentence does not reflect the fact landfills with low emission rates are below the detection limit of GHGSat, and the authors state themselves that only 151 landfills were observed out of the >10,000 landfills included in e.g. Climate TRACE. So there is a sampling bias, only landfills with large emissions are detected, and potentially for a given landfill, if the emissions are variable low emission times will be missed if these are below the detection threshold.
- 2) I would like to see more ground-truthing for the classification of landfills as "managed" or "dumping sites". For instance, is the presence of structures in satellite imagery sufficient to determine the management practice? How visible are emission abatement strategies such as gas recovery or the presence of active soil layers to encourage CH₄ oxidation?
- 3) In Supplement 6, the authors state that the distributions of managed versus dumping-site landfill emissions are different if the emissions are normalised by total landfill area, with the dumping-sites having significantly higher emissions. The authors then go on to compare the distributions normalising by only the active area, i.e., excluding the area of managed landfills that is covered or inactive, whereas for dumping-sites there is no closed or inactive area. This, in my opinion is an unfair comparison, as it is precisely the cover of a managed landfill that mitigates the emissions. To properly compare landfill emissions, it necessary to normalise by some parameter, such as amount of organic matter added to the landfill, or if this is difficult to obtain, the landfill total area.

L34: In fact, the amount of waste produced is more strongly linked to population (e.g. Chen et al. Environ. Res. Lett. 2020). Furthermore, the amount of organic waste, which is the most relevant for CH₄ emissions, should stabilise once food demand is met and not increase further with economic development, but rather only with population. I suggest the authors revise this sentence accordingly.

L39: The warming is relative to the pre-industrial temperature, which is defined in the IPCC AR6 as the mean for the period 1850-1900. I suggest the authors state that this is the warming relative to this time period.

L50: Specify what is “them” in this sentence – is it “the emissions”?

L74: In this sentence presumably “2021” refers to the year but it should be clearly stated, e.g. “A third of the methane emission plumes detected in the year 2021 by TROPOMI...”

L80: This sentence is not clear, does the 82% of all plume detections refer to the total number of detections across all 46 urban areas? Please revise to make this clear.

L81: It's not clear what is being referred to here, do the authors refer to the 130 urban areas where plumes were detected with GHGSat.

Figure 1: How was the classification of the landfills as “managed” or “dumping sites” determined? I see this is mentioned L144-146 but I think it should be mentioned earlier, i.e., when Figure 1 is referred to.

L119: The authors state they examined “emission variability against meteorology, including surface pressure”, but they should state what other meteorological variables they examined and not just “meteorology”.

L119: Concerning the lack of correlation with meteorological parameters, did the authors consider the possibility that this may be because just could not be detected in the column mixing ratio observations? For instance, how sensitive is GHGSat to the lower troposphere and near surface, where the signal would be the largest?

L119-122: I suggest the authors combine the sentence on L119-120 with the sentence starting “Although surface pressure has been reported...” since this logically follows. Whereas, the second part of the sentence on L122, starting “our findings based on...” should be a new sentence.

L143-146: How reliable is the visual classification of landfills? Was any ground-truthing performed to verify if the visual classification was correct? Also, it is possible to obtain information on the management practices at the landfills, for example, how many of these have the capacity to recover gas, and how many have active soil covers to encourage oxidation of methane?

L396-400: For TROPOMI, the IME-based emission rates lead to a 7 to 47% overestimation compared to atmospheric inversion estimates. However, I do not see an estimate for the uncertainty in the IME based emission rates for GHGSat? How important are uncertainties in the emissions estimated from GHGSat retrievals – can these be assumed to be similar to those obtained for TROPOMI? How does the uncertainty impact the conclusion that there is no difference in the emission distribution between “managed” and “dumping site” landfills?

(Remarks on code availability)

Referee #3

(Remarks to the Author)

Estimates of methane emissions from landfill are presented using satellite observations from GHGSat, along with urban area emissions from TROPOMI. Emissions from 151 landfill sites are estimated. The paper concludes that similar emissions are found for managed or open landfill, and that reported/modelled emissions estimates do not agree well with the estimates from the satellite.

I find the work to be extremely impressive. The technology is very exciting, and the investigation underpinning this article is exhaustive and meticulous. However, I have three general concerns about the framing of the results in the text, which should be addressed.

Firstly, I wonder how robust the comparison with reported or modelled (“bottom-up”) estimates can be, given the very small sample size in the top-down estimates (median number of samples is 5 per site), and because this sample is likely to be biased, due to the relatively high detection limit of the satellite. The discussion focuses on the difficulty in modelling landfill emissions, but couldn't the representation errors in the satellite plume detection also be a major factor in the difference between the top-down and bottom-up (e.g., the text in lines 174 and 175 discusses under- or over-estimates in the bottom-up, but doesn't discuss potential top-down issues)? One of the most interesting aspects of the paper is the remarkable correlation of the plume origin and the activity area within some landfills (an outstanding technical achievement by the team). However, doesn't this potentially underscore how variable these emissions could be? And therefore, how difficult or misleading it could be to compare annual reported emissions to a small number of snapshots (which are potentially biased high)?

Secondly, the fact that no statistical difference could be detected between managed and open landfills is an intriguing, and perhaps concerning, result. Several parts of the text suggest that a reason for focusing on landfill is that it is “low-hanging fruit” for mitigation, if management is improved (e.g., L43 – 45, L89 – 90, L243 – 245). However, the results of this paper would suggest that such efforts do not necessarily result in lower emissions? Perhaps I'm misunderstanding, but in any

case, I think the reader needs to be led through these apparently contradictory statements and findings.

Thirdly, I didn't understand the reason for the inclusion of the TROPOMI plume analysis in this paper. These plumes originate from integrated urban sources, and it wasn't clear to me how they tie in with the source-specific GHGSat landfill plumes. By the time I got to the discussion of the GHGSat results (L100 onwards), I was starting to get confused about the different numbers of identified plumes, urban areas, sites, etc. from the two different instruments. I suspect a reader unfamiliar with these datasets could be even more confused. Could the TROPOMI work be cut? I'm not sure what it adds.

In summary, I want to emphasize how impressive I found this work. However, to me, it is impressive from a technological/methodological perspective. Whether it should be published in a general interest journal such as Nature likely depends on how the authors address the above comments. I think a general reader may be very interested in the technology and methodology, but I think that the conclusions about landfill emissions are less well-defined, or potentially a little misleading.

Specific comments

L13 and 37: Slightly nit-picking, but methane emissions being the second most important contributor to climate change seems to be a somewhat ambiguous framing. I assume this means in comparison to total anthropogenic CO₂ emissions, but a reader could potentially interpret this as being compared to some emissions sector.

L36: I suggest providing the methane lifetime here, for the non-expert audience.

L43: I don't know what "treatment with energy recovery" means?

L59: What is the difference between an emission hotspot and super-emitting point source?

L80: I found this a little confusing. Is this saying that 14 areas EACH had ≥ 21 detected plumes?

L85: it's not immediately clear what the "19 areas" refers to here (and I still don't really understand why it's important).

L89: Waste disposal sites being the "most concentrated and mitigatable sources" seems rather a strong statement. Is this universally true? What is the reference? I don't think you need this motivating statement in any case.

L111: I'm clearly missing something here, but I was expecting the number of observations to be equal to the number with at least one plume plus the number of null detections. But 1447 minus 1085 doesn't equal 449...

164: I suggest using "estimates", rather than "data" (which, to me, suggests an observation). You may want to clarify what is meant by "modelled" (e.g., "process model-based", or similar).

L167 – 169: Couldn't this be due to the sampling bias in your dataset? Also, I don't understand what bias is referred to at the start of this sentence.

L218: I wonder why 16 was chosen as a threshold here (21 sites with 16+ plumes)? Earlier, (L104) talks about there being 23 sites with 20+ observations. How are these two statistics consistent?

L224: How will this fine-grained information help to focus mitigation efforts?

L226 – 228: I don't really understand why these other sources are pertinent to the discussion. Cut these lines?

L243 – 245: As in my general comments, this concluding sentence seems to be contradicted by the results on open vs managed landfill. Can you clarify?

Methods

L377: I suggest "its imaging capabilities enable the detection of anthropogenic methane..."

L381 – 383: The IME equation is provided later. If you're going to give the equation, it would make sense to include it at the first mention.

L414: What is the reference for the LES simulations?

L415: Introduce a_i here, rather than in the following sentence.

L426: By assuming zero uncertainty on the null results, aren't the site-average emissions uncertainties going to be underestimated, if they are included in the mean?

Supplement:

P2, L3: "compared", rather than "compare"

P18: I don't understand this sentence: "This points towards landfill management and emission mitigations measures being more efficient for closed modules that show no activity than for active ones that are in operation." How can a landfill be operating efficiently if there is no activity?

(Remarks on code availability)

I have taken a brief look at the code. It looks to be well documented (although would benefit from a README).

Version 1:

Reviewer comments:

Referee #1

(Remarks to the Author)

The manuscript presents several novel and significant findings, highlighting both discrepancies and consistencies between bottom-up and top-down observations. The use of satellite-based remote sensing offers fresh insights into global solid waste management, and the revised version provides a clearer presentation of key results, well supported by Supplements. I appreciate the authors' efforts in addressing my previous concerns and am pleased to recommend this version for minor revisions:

Line 61: Please replace "about 1500" with the exact number for clarity and precision.

Lines 72–73: I wonder if the GHGSat detection threshold is 13.3 mg/m²/day, which is derived from 100 kg/hr divided by (12 × 15 km²). If so, please state this explicitly.

Lines 121–122: The authors mention that the sampled sites fall on the upper end of the global waste disposal site emission rate distribution. This raises a concern regarding the representativeness of these sites for assessing global landfill and dumpsite greenhouse gas (GHG) emissions. Could the authors clarify how these data can be generalized to represent global emission patterns, given the potential bias toward high-emission sites?

Lines 122–124: The manuscript refers to different frequencies of satellite observations. Since GHG emissions from landfill sites are inherently time-dependent, it is unclear how these varying observation frequencies are interpreted in the context of emission magnitude. Could the authors elaborate on how such temporal discrepancies are addressed when extrapolating or projecting periodical GHG emissions? Additionally, how are inter-site observation variabilities reconciled to ensure consistency and robustness in the global-scale assessment?

(Remarks on code availability)

Referee #2

(Remarks to the Author)

This manuscript examines methane emissions from landfills across the globe using satellite remote sensing. Two satellites and instruments are used, the TROPOMI instrument onboard Sentinel 5P, which maps methane globally, and GHGSat, which is used to determine point source emissions based on plume images.

The authors have revised the manuscript based on the previous review and to a large extent addressed the comments from that review. I have now mostly minor comments, and only a couple of more major comments.

Major comments:

L21-26: "...also observe that managed landfills and dumping sites show similar levels of total emissions"

Although the authors have modified this statement based on the first review comments, I think it is still misleading. The authors state themselves that when the sites are normalized by area, dumping sites have larger emissions compared to managed landfills. In order to compare landfill emissions it is necessary to normalize by something that reflects the size of the landfill, since all else being the same bigger landfills would be expected to have bigger emissions. Therefore, I think the authors should add here (i.e. in the abstract/summary) that when normalized by landfill area the dump site emissions are larger than those of managed landfills.

Furthermore, I would like to see further investigation into the distribution of the emissions from dump sites versus managed landfills. The authors say they use a Kolmogorov-Smirnov (KS) test to check if the emission distributions differ. I'm not a statistician, but I think the KS test is only really suitable for continuous distributions and does not perform well where the distributions have long tails, which applies to these data. Therefore, I think the authors, should examine other methods for comparing the distributions.

L20: "Within this dataset..."

Although in response to the first review, the authors now add the qualification that the results only apply to this dataset, I think it is necessary to spell-out that "this dataset" represents only a small fraction of all landfill emissions and only landfills with very large emissions (detection limit of GHGSat being 100 kg/hr).

Minor comments:

L21: "...we find that our satellite-based estimates generally show no correlation with reported or modelled emission estimates"

I suggest the authors state here that this is at the facility and country scales.

L48: It is not clear what "These emission estimates" are, presumably those from inventories, but it is not stated anywhere.

L90-91: I am not sure what "Sentinel 2 detected surface activity" actually means, what kind of activity is detected.

L142: Please change "any significant link between them" to "any significant link with them" as "between" would mean between the meteorological variables, whereas what is meant is between the emissions and meteorological variables.

L141-142: I think it is a bit too speculative to conclude that "This finding suggests that operational practices could be driving emission variability for the sites we observed" based on the fact that no dependence of methane emission with meteorological parameters could be found. Are the differences in emissions between observations of the same site significant compared to the uncertainty of the emission estimates? And could the lack of dependence on meteorological parameters be simply because these variations are too small to be detected by GHGSat, and/or that there are too few observations of each site in different meteorological conditions?

(Remarks on code availability)

Referee #3

(Remarks to the Author)

The authors have addressed many of the reviewers' concerns and clarified several aspects of the paper. I think the paper still needs some work before it can be publishable. Furthermore, I think some of the findings need to be further softened, to avoid giving the wrong impression to readers.

All reviewers had concerns about the (necessarily) relatively small sample size, and sampling bias, and the extent to which this could be compared to bottom-up products and inventories, which are time-averaged. Some new text has been added to try to address this, but I don't find all of the new text to be helpful in this regard. I elaborate below.

The authors have also addressed what seems to be a misleadingly worded statement, that managed landfills were as emissive as open dump sites. They have clarified that, normalised by area, the emissions from open dump sites are much larger. But in that case, I'm not sure what wider lessons we are supposed to learn from this finding; doesn't it suggest that the two populations that have been measured just happen to have consistent total emissions, but for a large difference in landfill sizes? Unless I'm misunderstanding, I think the wording on this still needs to be softened substantially, as I'm concerned that readers could still draw the wrong conclusions from the text. See specific comments below.

Specific comments relating to these two comments:

- L20: When I read this line, the implication seems to be that the reports are models are wrong. I'm not sure that the new addition of "within this dataset" gives the reader enough information to judge how representative the observations are. I think that a further qualifying statement is required to emphasise that this difference may be because of sampling size and sampling bias in this work.

- L24-25: Unless I'm misunderstanding, as articulated above, I don't see a good justification for including this statement in the abstract. It seems to be a statistical coincidence, and I suggest cutting it.

- L25: The new part of this sentence that has been added is a little confusing: "the area where waste is being added often aligning with the detected emission sources within a facility". This should be re-worded in more plain English, but it is also an opportunity to re-state what is said in the main text: that it's mainly the open part of the landfill that is emissive, but the covered parts are not detected. It's a nice finding.

- L160: This needs a qualifying statement. Here, the new framing in the abstract would be appropriate ("within this dataset...").

- L171 – 172: I think this framing is also a little misleading. I don't think you can claim to have "detected" an emission rate of "2.9 million tons per year", because this requires (I assume) up-scaling to an annual total, and all the associated assumptions and sampling issues. I think this needs softening substantially.

- L211 – 225: Given the reviewers response, and my concerns about up-scaling the point measurements, I think this paragraph should be deleted. You are extrapolating to national scales an extrapolation (in time) of a small number of samples. Surely the errors involved in such an exercise will be so large as to preclude any reliable conclusions?

- L268: I'd emphasise the potential role of representation issues here, as it implies it's just the inventories that are wrong.
- L275: This probably doesn't need addressing in the text, but this statement does beg the question as to what satellite data would add, if we had continuous in situ monitoring... I guess it would be a way to test how representative the satellite data are (and therefore how useful for sampling unmonitored facilities)
- Figure S8.3: This is a strange new statistical test. I'm not sure what value it adds, and I had to read it several times to try to figure out what the authors were trying to show. I strongly suggest cutting it.

Other comments:

- I wonder if the authors have now gone a little too far in trying to say that this approach has been validated. It seems that the validation set is very small (1 or 2 releases). For me, this is fine, and we clearly need more work on validating these datasets. On L70, I'd like to see it mentioned that the validation dataset is small at present.

(Remarks on code availability)

Version 2:

Reviewer comments:

Referee #1

(Remarks to the Author)

All of my comments have been properly addressed. I have no further comments on the technical aspects of the paper. I think the paper is ready for publication in Nature.

(Remarks on code availability)

The code relevant to the manuscript is usable for the community, including the README file and all necessary Python code for reproducing the graphics. The code can be run.

Referee #2

(Remarks to the Author)

This manuscript examines methane emissions from landfills across the globe using satellite remote sensing and highlights the potential of this relatively new technology for monitoring waste emissions. It is comprehensive in scope, examining 151 waste sites globally with 1447 individual satellite observations.

The authors have substantially modified the manuscript and addressed my concerns from the last round of review. Therefore, I can recommend this paper for publication with no further revisions needed.

(Remarks on code availability)

Referee #3

(Remarks to the Author)

I think the authors have satisfactorily addressed my comments, and I think the paper can be accepted.

(Remarks on code availability)

I have not had time to review the code.

We thank all referees for their feedback and comments. We reply to all their points in **red** within their text, and reproduce excerpts of the revised manuscript highlighting **in bold** the implemented changes in the manuscript (with revised manuscript line numbers or Supplements page).

Referees' comments:

Referee #1 (Remarks to the Author):

While satellite data offers promising insights into site-specific methane emissions globally, we face several challenges in fully utilizing these data.

First, the reliability of the data is a concern. This includes the accuracy of translating meteorological parameters into actionable information and effectively filtering out noise.

We agree with the reviewer on satellite data coming with both large promise and some challenges. We think our manuscript provides a big step forward in using satellite data to provide actionable data on landfill emissions. Regarding the accuracy of GHGSat plume-based emission rate estimates, this question intersects with comments from referees #2 and #3. We include here a common answer to all referees.

<Beginning of common answer to referees #1 #2 #3 on accuracy>

Controlled releases are the gold standard validation reference for emission quantifications, including those made using satellite data. These consist in releasing methane at known emission rates on the ground at the same time of satellite overpasses. Satellite based estimated emission rates can then be compared with the metered rate. GHGSat has been extensively validated using both internal and public controlled releases. So far there have been two of those controlled release campaigns reported in the scientific literature (Sherwin et al., 2023, 2024). GHGSat participated in both those single-blind controlled release campaigns where the true emission rates are not known to the satellite data providers and the comparisons are done by third party, in this case the team from Stanford University (Sherwin et al., 2023, 2024). Figure R123.1 shows the results for the GHGSat-based emission estimates taken from Sherwin et al. (2023), demonstrating great agreement with the metered emission rates. Only one GHGSat observation is included in Sherwin et al. (2024), it agrees with metered emission within its uncertainty range (24% positive bias on a 0.4 t/hr metered emission rate).

Figure R123.1. Quantification performance by GHGSat, with 1-sigma X and Y error bars. The black dashed line denotes the 1:1 line. Fitted slope and uncentered R^2 shown for an ordinary least squares regression with the intercept fixed at zero (Figure copied from and caption adapted from Sherwin et al., 2023). Quantifications were made without knowledge of the real emission rate or wind speed. Please note the figure has three data points.

All controlled releases have been performed for point sources (at the scale of the satellites) up to now. These may not fully reflect the more spatially spread-out sources that landfills show (such as active surfaces) when the emission rates are close to the detection limit of the instrument. Performing controlled releases for such localized extended sources is a research effort that is currently under investigation (Dave Risk’s Flux Lab group at Saint Francis Xavier university) and we do account for the larger source area in the calibration of our emission quantification (see Method section in the manuscript).

Besides controlled releases, a different strategy to validate satellite-based emission estimates has been employed by Cusworth et al. (2024) for landfills. They showed that airborne mass-balance emission estimates based on in-situ concentration measurements (that measure the net emission flux from a landfill, including the sum of the diffuse area sources and the point sources) match within error bars with estimates based on simultaneous airborne methane plume imaging (similar to satellite-based plume imaging), with an instrument that can detect plumes down to 10 kg/hr. Besides, an in-depth study of two landfills near Madrid that included both similar airborne observations and GHGSat satellite observations showed that GHGSat satellite-based estimates satisfyingly match the total of airborne-detected plumes for same day observations (ESA study: Maasackers et al., 2023). Combined, these results point towards strongly emitting (areas in) landfills being detectable from space with emission estimates equivalent to total landfill emissions (within uncertainties).

We included the following changes in the manuscript:

In the main text lines 69-71	Calibrated mass-balance approaches are employed to translate these instantaneous snapshots into emission rates (see Methods), validated by single-blind controlled release²⁵
--

In the Methods section Lines 463-478	The calibration of this mass-balance approach against LES of known synthetic emission rates ensures that the estimated rates correctly account for the different advective transport conditions explored within the set of LES. Beyond this calibration on simulations, numerous real-life validation efforts have been organized, including controlled-releases experiments which are the validation gold standard. Notably, GHGSat participated and showed excellent agreement with metered emission rates in two single-blind controlled release campaigns, where the true emission rates (and wind speeds) are not known to the satellite data
---	---

	providers and the comparisons are done by a third party (in this case a research group from Stanford University)^{25,55}. Beyond controlled releases, landfill emission rates obtained through aerial methane imagery with an instrument that can detect plumes down to 10 kg/hr have been validated against traditional aerial mass-balance results^{18,41}. Besides, an in-depth study of two landfills near Madrid that included both similar airborne observations and GHGSat satellite observations showed that GHGSat satellite-based estimates match the total of airborne-detected plumes for same day observations within uncertainties⁵⁶. Combined, these results show that GHGSat satellite-based observations can provide accurate estimates of methane emissions from waste disposal sites.
--	--

<End of common answer to referees #1 #2 #3 on accuracy>

Second, how can we generate continuous temporal profiles from satellite data? This is crucial for assessing the feasibility of employing satellites for long-term monitoring and global data sharing.

By essence, sun-synchronous satellites in low-Earth orbit provide only instantaneous snapshots, making it impossible to obtain fully continuous temporal profiles (at least with the satellite instruments currently in orbit). However, they can cover the whole planet at a high revisit frequency (depending on orbit and observation geometries, targeting strategy, etc.) with globally consistent observations. They are thus extremely useful for global scale surveys (such as this work) and are complementary to traditional monitoring methods that can be deployed on the ground to provide a continuous temporal resolution. Geostationary satellites could offer even higher up to near-continuous temporal coverage. However, there are no geostationary satellites currently in orbit providing observations with a methane sensitivity comparable to TROPOMI and/or GHGSat.

However, beyond continuous temporal resolution, satellites can offer revisits of the same sites to explore emission variability at temporal scales ranging from a few dozens of minutes (same day observations) to yearly. This way, ample estimates can be obtained to compare to either surface observations or model-based emission estimates. In this study, 23/151 sites have been observed at least 20 times, and the median observation number is 5 (see Main text). This temporal resolution has for example been used to discuss emission rate variability against meteorological variables (Supplement S5) or to discuss the origin of the observed plumes (see Main text, Figure 1 – Figure 2 of the preprint –, Supplements S13 and S14).

We adjust the main text at several locations to better reflect what satellites can offer (coverage) and how this compares to the capabilities of on-ground methods.

In the introduction

In the main text lines 55-60	Considering all these uncertainties, independent observations of methane emitted from waste disposal sites are critical and can be obtained through various on-ground and/or aerial measurement methods¹⁸ that are deployable at site level and that can provide emission estimates at high
--

	continuous temporal resolution. Complementary to these site-specific approaches, satellites offer extensive global coverage, providing consistent observation sets across a large number of sites.
--	---

Describing satellite observations

In the main text lines 69-71	Calibrated mass-balance approaches are employed to translate these instantaneous snapshots into emission rates (see Methods), validated by single-blind controlled release ²⁵
---

In the conclusion

In the main text lines 272-275	Ideally, such studies would involve partners operating waste disposal sites, bottom-up modelers, aerial and satellite-based methane observations augmented by complementary on-ground observations that can provide continuous measurements, including at night.
---

Third, how do we integrate satellite data with ground-based observations? Based on the current manuscript, I find reconciling the two sources challenging.

Following the previous reply, we see satellite-based observations as complementary to ground-based observations. They offer a wider coverage across sites while ground-based observations can provide a finer continuous temporal resolution and provide additional spatial resolution (for example using walk-over surveys for which the satellite observations can provide useful guidance and total emissions). The reconciliation of satellite and sub-orbital data is possible, illustrated for example by Cusworth et al. (2024) showing that aerial mass-balance flights relying on in-situ measurements of methane concentration satisfactorily match methane plume imaging results.

Overall, the reconciliation of satellite- and ground-based data calls for dedicated measurement campaigns that combine all these different observations to provide a comprehensive picture of methane emissions on various types of landfills. We recommend in the conclusions that such dedicated campaigns are organized, notably to help bridge the gap between top-down and bottom-up understandings of landfill methane emissions that our results show (for the 151 sites included in our sample).

The modifications from the previous reply contribute to reflect this question in the revised manuscript as well.

Looking at Fig. 2, I noticed that the site and emission sources don't perfectly overlap, which suggests that wind might play a role in the discrepancy. Have you considered the wind direction at the time of the scan? If the wind is blowing in the opposite direction, there could be emissions from sources beyond the landfill that need to be excluded. Although the authors mention this in lines 226–228, there are still recognized facilities contributing to the emissions. Were there instances where emissions occurred from unidentified sources near the landfills? This raises questions about the uncertainties inherent in satellite data. Given its limited application in this field, I am hesitant to fully trust the data without further validation. However, applications of satellite data from other fields might help build that trust.

The wind direction is taken into account to automatically estimate the source of the plume as the most upwind highly enhanced pixel in the plume mask (columns named “lat,lon” in /data/Plume_detection_CSVs/GHGSat_detected_plumes.csv provided in the Code Ocean reproducibility capsule). Elevated methane enhancements that constitute the plume are transported downwind from the source through advective atmospheric transport, often extending beyond the facility boundaries.

We have manually verified all these estimated sources, manually selecting multiple sources where appropriate (See Methods: Comparison of landfill surface activity results and GHGSat methane plume origins, column “manually_pinned_sources” in /data/Plume_detection_CSVs/GHGSat_detected_plumes.csv provided in the Code Ocean reproducibility capsule) for cases that show one plume mask containing two plumes that arise from within the same facility. Figure R1.1 illustrates such an example for a landfill located near Casablanca, in Morocco.

Figure R1.1. Methane emission plume observed on 2022-07-25 at a landfill near Casablanca (Morocco). The magenta dots denote manually verified (and pinpoint where appropriate) plume origin(s), and both vector arrow illustrate the wind direction from two meteorological products (ERA5 and GEOS-CF).

Because the wind direction allows us to determine where a given plume originates from and the observation provides methane data at 25-m resolution, we can be certain that the observed facility is responsible for the detected emission plume. The data therefore ensure that other sources upwind or near the landfill do not contribute to the emission plumes that are quantified here. As can be observed for the GHGSat plumes showcased in Figure 1 (Figure 2 in the preprint), the strong plume enhancements dissipate into background methane levels within a few kilometers. Hence, the possible influence of all sources located further away is corrected when removing the background methane concentration.

We include an updated description of the data we use, and of what defines a plume in the revised manuscript:

Main text lines 66-71	They provide spatial images of atmospheric methane concentrations that enable the detection of anthropogenic emission plumes. These consist of strong enhancements in methane concentration that extend downwind from localized emission sources, as illustrated in Figure 1. Calibrated mass-balance approaches are employed to translate these instantaneous snapshots into emission rates (see Methods), validated by single-blind controlled release²⁵.
---

We include the following additional explanations in the Methods section to explain how we can determine the origin of the detected plumes and thus attribute the emissions.

In Methods lines 576-580	The wind direction allows to estimate the plume origin as the most upwind highly enhanced pixel included in the plume mask. In addition, we manually verify this result and pinpoint the approximate source(s) of all GHGSat plumes, allowing to select multiple sources for overlapping plumes originating from the disposal site where appropriate. We use these source locations to compare to the Sentinel-2 based surface activity analysis.
--

Besides, we also modify Figure 1 (Figure 2 in the preprint) to include a wind vector illustrating the wind direction at the time of the observation, and to include the plume origins for GHGSat detected plumes.

In Main text, Figure 1 caption lines 92-96	Black crosses show site locations, white dots the GHGSat plume origins and thick black contours highlight landfill site boundaries. White arrows (panels a, b, c, d, e, f, g, h, j, k, l and m) illustrate the wind direction sampled from the ERA5 reanalysis³⁵, with the wind speed provided inset. Plumes overall follow the reanalysis wind direction, with some exceptions at low wind speeds.
--	---

The archive GHGSat observation dataset analyzed in this work has been carefully filtered to only include emission plumes detected at or very near waste disposal sites. Because a gas processing and a wastewater treatment plant are located near a landfill close to Madrid and another close to Shanghai, respectively, two plumes coming from these other sites were included in the dataset (sites 52 and 42, resp.) for illustration purposes to highlight the possibility of mitigable emissions sources nearby/related to the landfill. Figures included in Supplement S15 included in the preprint clearly align along the wind direction and arise from outside of the landfill masks. No plume origin were reported for these two plumes because we excluded them from the analysis.

GHGSat observes spatially contiguous 12 x 15 km² images of methane concentration at ~25 m resolution, and automatically identifies and selects spatially consistent methane enhancements that originate from the vicinity of the(ir) target(s) as emission plumes. All resulting plume masks are manually verified by a trained GHGSat operator. In practice, our dataset shows that the overwhelming majority of the plumes considered have origin(s)

located within landfill boundaries, except for a few that are located in the immediate proximity of the boundary, outside of it, but in all likelihood arising from the waste disposal site.

We better describe the illustration purpose of these two plumes originating from facilities neighboring waste disposal sites in the revised manuscript:

In Main text, lines 251-256	This spatial information can for example help site operators focus mitigation efforts more effectively. Our dataset also includes two example plumes originating from adjacent facilities: a biogas plant near the Las Dehesas landfill near Madrid and from a wastewater treatment plant near Shanghai (both filtered from the analysis, see Supplements S15). They illustrate the mitigation potential that satellite observation can detect in facilities related to and neighboring waste disposal sites.
---

Additionally, where can we locate the centers of these emissions? The map shows several hotspots of concentrated emissions, some of which are outside the landfill boundary. I speculate this could be due to vertical variations in the methane plume. Satellite scans capture only 2D information, but assessing emission volumes requires 3D data. How can we address this limitation to obtain accurate results?

These questions on accuracy and source location are closely related to the ones included in the previous paragraph. We detail above how emission sources are automatically identified using wind direction (and manually verified), and we explain as well that enhancements are transported downwind of the source, often beyond its boundaries, through advective atmospheric transport. Figure 1 (Figure 2 in the preprint) now includes wind direction vectors and estimated emission sources for all illustrated GHGSat plumes.

Regarding the “2D information” part of the question: GHGSat retrieves total column of methane, or methane column density. Because this column is a vertically integrated (concentration) content, the full volume (and hence extent of the methane plume) is taken into account and mass-balance based estimates are thus possible.

References provided in the Methods sections of the article describe these quantities and the physics of the retrieval in full detail. However, to clarify this aspect, we adjust the revised manuscript in the following way:

In Methods lines 407-409	Total columns (vertically integrated concentrations) of methane with near vertically uniform sensitivity down to the surface are retrieved from these observations using a full-physics approach
---

In Methods lines 441-443	These instruments estimate the total column (vertically integrated concentration) of methane at $\sim 25 \times 25 \text{ m}^2$ resolution over targeted $12 \times 15 \text{ km}^2$ domains
---

The manuscript also presents limited data from Africa, Australia, northern North America, and northern China. Is there a reason for this?

The archive of waste disposal site observations analyzed in our study was gathered opportunistically, in parallel to regular (commercial) GHGSat activities. As GHGSat observations only cover areas of 15 km x 12 km, simply not all landfill sites have been observed. 51% of the opportunistic observations have been guided by super-emitting urban areas uncovered by the TROPOMI observations. Some super-emitting urban areas may have been missed by TROPOMI because of limited satellite coverage due to poor observation conditions or because of TROPOMI high 8 t/hr detection limit. Besides TROPOMI-guided observations, the other half of the observations were acquired by exploratory initiative in GHGSat planning strategy based on other datasets, trading-off with other interesting methane-emitting targets such as oil and gas (e.g. Alberta in Canada, Permian basin in the US, Algeria, Libya or Nigeria in Africa) or coal mines (Shanxi mountains in Northern China, East coast of Australia, South Africa). These trade-offs can explain coverage disparities.

However, it is to be noted that this work (N=151) satisfactorily covers the large-scale patterns showed by the distribution of the waste disposal sites identified by Climate TRACE, as illustrated in Figure R1.2. We observe more on the East coasts of both North and South Americas than on their West coasts, we cover Southern Europe, North Africa, Nigeria and South Africa, South and South East Asia as well as South China that shows more landfills than in the North. Regarding Australia, we cover both coasts as Climate TRACE, also taking into account where the population is located in this continent-sized country.

Figure R1.2. Distributions of the sites observed in this work (top) and included in Climate TRACE dataset (bottom).

Moreover, how can we validate data collected during limited detection periods? Methane emissions from landfills are highly dependent on temperature and moisture, which vary seasonally. This calls for time-series profiles to better understand the emissions. How do you plan to address this challenge?

We addressed the question of validating single snapshot satellite-based emission estimates in the answers above.

Regarding the question of validating satellite-based results over longer time ranges, comparing total sums of emissions across sites between satellite and reported data can provide an indication of possible large systematic biases included in the satellite-based estimates. This question intersects with another comment by referee #3, we thus include below a common answer to both referees.

<Beginning of common answer to referees #1 #3 on accuracy across time>

If the few snapshots per site are significantly biased towards emissions anomalies (because of diurnal variability, or management practices, etc., short time scale variability), total summed emissions across all sites should be significantly different between GHGSat and reported or modelled bottom-up results. In Tables R13.1 and R13.2, we complete Tables S8.1 and S8.2 to include sums across all sites of each category. We can notice that while we find large site-to-site discrepancies, the total emission sums across all sites are not significantly different (considering the ~100% uncertainty for reported totals in the US, depending on the bottom-up method) between top-down and bottom-up. Finally, at global scale we report that conservatively population-averaged GHGSat results agree with UNFCCC estimates within their error bar (see Main text in preprint lines 205-206 or revised manuscript lines 222-225). Even if, per site, the small number of available points may bias emission rates towards a specific season, etc. (variability over a longer time range), we can note that, overall, our dataset shows no correlation with meteorology (see Supplements S5). We also find that only using sites for which we have a higher number of observations (above 5) does not improve the correlation against reported or modelled data (see Figure R3.1 related to a later comment, reproducing Figure 3 for sites that have at least 5 observations). All this evidence suggests that there is no significantly large variability-based emission bias when relying on the satellite data included in our dataset. Finally, the availability of methane-sensitive satellite data at facility-scale is growing, which will allow to study this question further in the future.

Reporting scope	Number of sites	GHGSat reported difference	– GHGSat totals	Reported totals
USA	22	0.43 ± 1.45 t/hr	32.6 ± 4.8 t/hr	23.1 t/hr (reported to GHGRP) 49.9 t/hr (using emission models)
Canada	8	-0.49 ± 0.93 t/hr	3.2 ± 0.6 t/hr	7.1 t/hr

EU	7	1.75 ± 1.22 t/hr	14.5 ± 1.5 t/hr	2.3 t/hr
All together	37	0.48 ± 1.49 t/hr	50.3 ± 5.1 t/hr	32.6 t/hr (with data reported to GHGRP) 59.3 t/hr (using reported emission models for US)

Table R13.1. Methane emission rate statistics comparing GHGSat-based rates and data included in facility-scale reporting program databases.

Waste disposal site types	Number of sites	GHGSat – Climate TRACE difference	GHGSat totals	Climate TRACE totals
Managed landfills	77	-0.63 ± 4.93 t/hr	170 ± 8 t/hr	219 t/hr
Dumping sites	32	-0.35 ± 2.93 t/hr	63 ± 6 t/hr	74 t/hr
All together	109	-0.55 ± 4.44 t/hr	234 ± 10 t/hr	294 t/hr

Table R13.2. Methane emission rate statistics comparing GHGSat-based rates and data calculated by Climate TRACE.

We include these elements in the supplements of the revised manuscript:

Supplement S8 pages 26-28	We included this analysis in Supplements S8.
--

<End of common answer to referees #1 #3 on accuracy across time>

Finally, we addressed the question related to the necessity of temporal profiling in the answers to previous comments above: our conclusions support the complementary use of satellites (that excel at coverage) and on-ground measurements (that can provide continuous temporal profiling) to better understand top-down estimations of landfill methane emissions and bridge the gap with bottom-up approaches. We also improved the description in the introduction of how satellites are complementary to traditional measurement methods on the ground (see above).

Lastly, I understand there is a less-than-linear correlation between GHGSat observations and reported methane emissions/Climate TRACE methane emissions. What about the correlation between reported methane emissions and Climate TRACE data for the same sites? Is it possible that the dataset is too small to establish meaningful correlations between satellite and ground-based observations? Beyond this, satellite observations should be linked to specific times and landfill stages to strengthen correlations with on-land data.

Figure R1.3 shows how Climate TRACE data compares to reported bottom-up data. The correlation is excellent because Climate TRACE uses these reported datasets to train their data-driven emission models (Climate TRACE, 2023). Climate TRACE does not use ground-based observations to produce their modelled emission results.

Figure R1.3. Comparison for Climate TRACE modelled data against reported methane emissions.

Regarding the size of the dataset, we ensured that the set of sites included in the study are as representative as possible of different locations around the globe, acknowledging that their emissions must be high enough to produce a plume above GHGSat detection threshold at least once, which means that our sample is located on the upper end of the emission rate distribution.

We precise this important aspect in the main text:

Main text line 120-122	Only sites for which at least one methane emission plume has been detected by GHGSat are included, meaning that our sample is on the upper end of the global waste disposal site emission rate distribution.
---

The conclusions we draw from our dataset do not presume anything with regard to results that could be obtained with more sites.

We adjusted our abstract and conclusions to better reflect that we do not draw conclusions beyond the extent of our dataset.

Abstract line 20-21	Within this dataset, we find that our satellite-based estimates generally show no correlation with reported or modeled emission estimates.
---

Main text line 266-268	Across the 151 surveyed sites, we find that bottom-up and top-down satellite-based solid waste emission estimates cannot currently be reconciled at facility and country scales.
---

Finally, we agree with the referee that the interpretation of satellite data would be eased with waste disposal site operators providing publicly available operational practice data, with temporal resolution, if possible, into harmonized datasets. Getting increased temporal resolution in bottom-up dataset could indeed help improve the agreement between satellite-based estimates and bottom-up results, as we suggest in the conclusion of the manuscript.

Comments on the waste management aspect:

As we navigate a complex landscape of models and discrete tests on methane emissions from global landfills, satellite remote sensing technology offers a new perspective that could provide a clearer and more comprehensive view. This manuscript represents a potential breakthrough in remote sensing applications for solid waste management, demonstrating a high-tech, cross-disciplinary innovation. The results, while preliminary in terms of both completeness and temporal coverage, signal a paradigm shift. This approach has the potential to move landfill monitoring from isolated, sporadic measurements to large-scale, continuous surveillance, much like the role meteorological satellites play in weather forecasting.

However, for this technology to gain widespread acceptance, it must demonstrate its credibility and cost-effectiveness. As of now, significant barriers remain, particularly regarding the high costs associated with satellite scanning. This financial hurdle is especially critical for low- and middle-income countries, where the majority of the world's landfills and dumpsites are located. Given that climate change is a global issue, this scenario presents an opportunity for regional cooperation, with potential for knowledge transfer and data-sharing that could enhance the adoption of satellite monitoring technologies.

We agree with the referee that data availability and cooperation between satellite data providers and waste disposal site operators from all countries, including from low- and middle-income countries, is an important aspect to take into account in order to increase the adoption of satellite data where relevant. We think this manuscript can also play a role in that process.

GHGSat pioneered providing high-resolution satellite monitoring of anthropogenic emissions and has developed by selling these services to paying customers. However, when relevant and possible, GHGSat participates in research efforts that contribute to better understanding anthropogenic methane emissions (e.g. Varon et al., 2019, Maasackers et al., 2022, Schuit et al., 2023), providing public access to all the observations that are instrumental for the results. In the same way, we will provide all the plume observations detected by GHGSat that are included in this work (see our Data availability statement).

Beyond these collaborations, GHGSat methane observations are becoming more widely accessible with GHGSat recently joining the European Space Agency's (ESA) Third Party Mission (TPM) program, the National Aeronautics and Space Administration's (NASA) Commercial Smallsat Data Acquisition (CSDA) Program and the U.K. Satellite Applications Catapult. These enable scientists to access archived GHGSat observations and submit targets to be observed. Besides, GHGSat also recently became a Copernicus Contributing Mission through the Copernicus Atmosphere Monitoring Service (CAMS):

- https://www.esa.int/Applications/Observing_the_Earth/GHGSat_joins_ESA_s_Third_Party_Mission_Programme

- <https://www.ghgsat.com/en/newsroom/ghgsat-approved-for-nasa-commercial-smallsat-data-acquisition-csda-program/>
- <https://sa.catapult.org.uk/news/ghgsat-and-satellite-applications-catapult-accelerating-climate-innovation-in-the-uk/>
- <https://www.ghgsat.com/en/newsroom/ghgsat-joins-esa-copernicus-contributing-mission/>

Regarding collaboration with operators and Global South countries, SRON and GHGSat are currently involved in a collaboration with the Global Methane Hub (see our Acknowledgements statement) named “Targeting Waste Emissions Observed from Space (TWOS)”. It supports emission mitigation efforts across different landfills located in the Global South by providing monthly monitoring of methane emissions from space to local authorities and waste disposal site operators that are also involved in the project:

- <https://earth.sron.nl/project/targeting-waste-emissions-observed-from-space/>
- <https://www.ghgsat.com/en/newsroom/new-projects-use-space-data-to-cut-landfill-emissions/>

The aim of this project is to provide data to support emission mitigation at landfills in the Global South. This is also important as additional facility-scale satellite data will become available in the public domain in the future.

Besides GHGSat, the availability of satellite-based facility-scale observations of methane emissions is currently growing, with new instruments being progressively put on orbit. We can mention here NASA Earth Surface Mineral Dust Source Investigation (EMIT) flying since 2022 on the International Space Station (ISS) that was not originally designed to observe methane, but can nonetheless image emission plumes in ideal conditions from ~1 t/hr (Thorpe et al., 2023, Ayasse et al, 2024). In addition, the Carbon Mapper initiative has recently launched its first satellite Tanager-1, that is expected to provide openly-accessible facility level methane observations with a detection threshold comparable to GHGSat’s.

We include in the revised manuscript a reference to this expanding suite of satellite as a positive outlook for improved satellite data availability

Main text line 263-266	The availability of such high-resolution methane-sensitive satellite observations is currently growing, with the expanding GHGSat constellation and new initiatives like Carbon Mapper’s Tanager-1 satellite, as well as public hyperspectral satellite missions³.
--

Yet, we must acknowledge that the validation of satellite data is still in its infancy. The reliability of these measurements requires time, with further testing, validation, and refinement of both the scanning equipment and inversion methods. Only through extensive use and iterative improvements will we be able to fully assess the validity and accuracy of this technique.

In the answers provided above we show that different strategies have been and are currently being developed and implemented to validate the accuracy of satellite plume-based emission rates. These efforts show positive results and are more than adequate to support the findings of the current manuscript.

Returning to the core focus of the paper, satellite-based monitoring of methane emissions could reveal large-scale, dynamic changes over time, offering new possibilities for real-time tracking of various landfill attributes. For instance, this technology could potentially detect accidental chemical leaks, allowing for swift intervention before they escalate into major environmental incidents.

Looking beyond the specifics of satellite monitoring, an essential question emerges: how should we balance old and new technologies in environmental science? Despite the promise of new methods, traditional techniques will remain crucial for the foreseeable future. The coexistence of multiple technologies is not a contradiction, but a testament to the diversity of tools available to address complex challenges in waste management. This diversity ensures that technological innovation continues to advance in the sector while maintaining a foundation of proven methodologies.

We fully agree with the reviewer. This is also consistent with the previous discussion/points intersect with previous ones regarding the difference between satellite-based observations, and other technologies that can provide continuous temporal profiles. We see satellite-based observations as a complement to traditional techniques that can be deployed on the ground, as well as to emission modeling, not as a replacement: satellites offer coverage across many sites while traditional techniques can provide fine temporal resolution and modeling can link emissions to underlying processes. The conclusions of our Main text in the preprint already argues for in-depths studies combining all these complementary approaches.

The revised manuscript now strengthens this message:

In the introduction

In the main text lines 55-60	Considering all these uncertainties, independent observations of methane emitted from waste disposal sites are critical and can be obtained through various on-ground and/or aerial measurement methods¹⁸ that are deployable at site level and that can provide emission estimates at high continuous temporal resolution. Complementarily to these site-specific approaches, satellites offer extensive global coverage, providing consistent observation sets across a large number of sites.
--

Describing satellite observations

In the main text lines 69-71	Calibrated mass-balance approaches are employed to translate these instantaneous snapshots into emission rates (see Methods), validated by single-blind controlled release ²⁵
---

In the conclusion

In the main text lines 272-275	Ideally, such studies would involve partners operating waste disposal sites, bottom-up modelers, aerial and satellite-based methane observations augmented by complementary on-ground observations that can provide continuous measurements, including at night.
---

Referee #2 (Remarks to the Author):

General comments

This manuscript examines methane emissions from landfills across the globe using satellite remote sensing. Two satellites and instruments are used, the TROPOMI instrument onboard Sentinel 5P, which maps methane globally, and GHGSat. Significant methane emissions from landfills are observed globally and point to targets for methane emission mitigation. As such, this study presents new findings and the scope of the study is a significant advancement compared to existing published studies also using satellites to detect methane emissions from landfills.

In the study, the authors examine the emissions for two classes of landfills, “managed” versus “dumping sites” with the authors concluding that there is no significant difference in emission rates between these two types. This is a very important conclusion, which must be fully supported by the data. However, at present there are some remaining questions about this conclusion, which need to be addressed before the study can be accepted for publication.

Specific comments

L20-21: The statement that “managed fills and dumping sites show similar levels of emissions” is a very important one and needs to be fully supported by observations. However, I have some reservations about this statement:

1) This sentence does not reflect the fact landfills with low emission rates are below the detection limit of GHGSat, and the authors state themselves that only 151 landfills were observed out of the >10,000 landfills included in e.g. Climate TRACE. So there is a sampling bias, only landfills with large emissions are detected, and potentially for a given landfill, if the emissions are variable low emission times will be missed if these are below the detection threshold.

Our study only includes sites for which at least one plume has been detected in GHGSat observations. We acknowledge that this means that our sample is located on the upper end of the emission rate distribution.

We precise this important aspect in the main text:

Main text line 120-122	Only sites for which at least one methane emission plume has been detected by GHGSat are included, meaning that our sample is on the upper end of the global waste disposal site emission rate distribution.
---

The conclusions we draw from our dataset do not presume anything with regard to results that could be obtained with more sites, especially sites emitting methane below GHGSat detection threshold. We adjusted our abstract and conclusions to better reflect that our conclusions relate to our dataset.

Abstract line 20-21	Within this dataset , we find that our satellite-based estimates generally show no correlation with reported or modeled emission estimates.
--

Main text line 266-268	Across the 151 surveyed sites , we find that bottom-up and top-down satellite-based solid waste emission estimates cannot currently be reconciled at facility and country scales.
--

Beyond the question of the studied sample, please note that we include all detections and non-detections (considered as 0 t/hr) in our average emission calculation approach. This means that the averages that we report do (conservatively) account for temporally-variable emissions that can be below the GHGSat detection threshold when observed by the satellites.

2) I would like to see more ground-truthing for the classification of landfills as “managed” or “dumping sites”. For instance, is the presence of structures in satellite imagery sufficient to determine the management practice? How visible are emission abatement strategies such as gas recovery or the presence of active soil layers to encourage CH₄ oxidation?

In this work, we resorted to devise our own classification of waste disposal sites as we could not find a dataset providing a classification type for all the sites that we observed. As an example, Nanda et al. (2021) distinguishes:

- open dump landfills: waste is dumped arbitrarily in an open environment with air contact
- semi-controlled landfills: waste is compacted and organized, covered with top-soil but gas emissions and leachate are not managed
- sanitary landfills: includes semi-controlled landfill features with additional gas emission and leachate management

As suggested by the referee, accurate knowledge of management practices such as the presence of gas capture systems or the nature of top-soils is difficult to infer from timeseries of satellite imagery included in Google Earth. Following Nanda et al. (2021), this makes the distinction between semi-controlled and sanitary landfills arduous.

Consequently, we opted to sort waste disposal sites between “managed landfills” (semi-controlled or sanitary landfills following Nanda et al. 2021) that show organized structures to bury waste in Google Earth imagery (such as isolated modules that evolve with time, covers being visible in some parts, soil covering location previously showing waste to the open air, etc.) and “dumping sites” that do not exhibit such features. This distinction is easier to determine using Google Earth imagery timeseries.

As an independent verification of the classification type of the sites studied, we also compared to the Climate TRACE site classification. The Climate TRACE dataset includes a waste disposal site type built from a variety of sources ranging from low (e.g. Global Plastic Watch) to high confidence (e.g. EPA, Waste Atlas, etc.). Across the 109 waste disposal sites of our dataset intersecting Climate TRACE, three site types are reported by Climate TRACE: “nan” (N=21), “Sanitary Landfills” (N=54) and “Dumpsites” (N=34). Figure R2.1 presents the confusion matrix between Climate TRACE waste disposal site types and ours, that were determined in a completely independent way.

Figure R2.1. Waste disposal site type confusion matrix between the Climate TRACE dataset and our own site type classification.

Our classification of waste disposal site types agrees for 78/88 (89%) of the sites for which Climate TRACE reported either “Dumpsites” or “Sanitary Landfills”. The largest cluster of disagreements is sites that we report as “managed landfills” whereas Climate TRACE identifies them as “Dumpsites” (N=8). We double-checked our classification for these sites confirming our original classification based on Google Earth imagery timeseries. This discrepancy illustrates that our “managed landfill” label includes more managed sites than just the managed landfills that comprise elaborate leachate and gas emission management systems (the Nanda et al. (2021) definition).

Finally, we use the Climate TRACE site types, which are independent of our classification, to discuss the robustness of our conclusions regarding differences in emission distributions. Figures R2.2 and R2.3 include total (filled colors) and full-area-normalized (lines) distributions of waste disposal site methane emissions using our classification and Climate TRACE’s, respectively. We notice from the two-sided KS tests results for all cases that our conclusions do not change significantly between using our classification on all 151 sites, our classification for the 88 overlapping Climate TRACE sites, or Climate TRACE’s classification for the 88 overlapping sites.

Figure R2.2. Distributions of total (full colors) and full-area-normalized (lines) methane emission rates for “managed landfills” (orange) and “dumping sites” (purple) following our classification.

Figure R2.3. Distributions of total (full colors) and full-area-normalized (lines) methane emission rates for “managed landfills” (orange) and “dumping sites” (purple) following our (top), and Climate TRACE’s (bottom) classifications, for sites that have a non-null Climate TRACE label.

Changes in the Supplements:

Supplements S6 pages 19-20	This analysis using our both our own classification of site types and Climate TRACE’s has been included in Supplements S6 of the revised manuscript.
--

3) In Supplement 6, the authors state that the distributions of managed versus dumping-site landfill emissions are different if the emissions are normalised by total landfill area, with the dumping-sites having significantly higher emissions. The authors then go on to compare the distributions normalising by only the active area, i.e., excluding the area of managed landfills that is covered or inactive, whereas for dumping-sites there is no closed or inactive area. This, in my opinion is an unfair comparison, as it is precisely the cover of a managed landfill that mitigates the emissions. To properly compare landfill emissions, it necessary to normalise by

some parameter, such as amount of organic matter added to the landfill, or if this is difficult to obtain, the landfill total area.

This comment is intersecting with one provided by referee #3 (their second general comment). We are reproducing here a common answer to both comments.

<Beginning of common answer to referees #2 #3 on difference of distribution between landfills and dumping sites>

The emission rate distribution comparison between managed landfills and dumping sites is one of the striking results that our dataset provides. We agree with both referees #2 and #3 that the conclusions drawn from this comparison must be phrased very precisely.

On the one hand, total waste disposal site emissions are relevant to discuss climate-change impacts of landfill methane emissions. Our data strikingly show that managed landfills and dumping site do not have significantly different emission rates distributions. On the other hand, emission rates normalized by the full area of waste disposal sites are relevant to assess the efficiency of potential mitigation measures put in place. Applying this normalization yields area-normalized emission rates being significantly lower for managed landfills compared to dumping sites. This reflects the expected effect of definitively covering and closing some parts of managed landfills (closed modules are included in our landfill masks), thus confirming the efficiency of this mitigation strategy, and clearing the apparent contradiction of managed landfills emitting at similar levels compared to dumping sites.

We interpret the impact of normalizing emissions by the total site area for managed landfills as a small fraction of the site area being responsible for the detected emissions. Following conclusions from our Sentinel-2 based surface activity correlating with plume origins, we identify the open active side of managed landfills (or work face) as this small area responsible for most of the emissions. These two results put the spotlight on the predominant role open active work faces play in managed landfills emissions. Similar conclusions on the predominant role of the work face have been drawn from extensive aerial survey datasets collected in the United States (Cusworth et al., 2023, Scarpelli et al., 2024). Our dataset confirms these findings with a global perspective, discussing emission distribution differences across management and economic development levels.

In a nutshell, the conclusions obtained from our dataset mean that managed landfill active surfaces are not significantly different from dumping sites regarding total emission rates (total emission distribution comparison) until they are mitigated by closing the active module for good (total area-normalized emission distribution comparison).

We include the following changes in the revised manuscript to reflect these ideas, explain the apparent contradiction and avoid misleading readers.

In the abstract: with this new formulation, we seek to directly link “similar levels of emissions” directly with “where waste is being added”

Abstract lines 21-26	This reveals major uncertainties in the current understanding of methane emissions from waste-disposal sites, warranting further investigations to
--

	reconcile bottom-up and top-down approaches. We also observe that managed landfills and dumping sites show similar levels of total emissions, with the area where waste is being added often aligning with the detected emission sources within a facility.
--	--

In the main text:

Main text line 161-164	Managed landfills and dumping sites do not show statistically significant different detected emission rate distributions. However, when normalized by the total site area, managed landfills show significantly lower area-normalized emission rates compared to dumping sites, thus showing expected effects of emission mitigation through closing and covering modules of the landfill (see Supplement S6).
---

In the main text:

Main text line 236-248	When considering only the 21 sites for which at least 16 plume origins can be identified in GHGSat observations, we find statistically-significant proximity for 18 (86%) of them (See Supplements S14). Revisiting our dataset showing that total site area-normalized emission rates are significantly lower for managed landfills compared to dumping sites (see Supplements S6 and above), we conclude that the small fraction of open active areas in managed landfill accounts for almost all of the emissions detected by GHGSat. This highlights the predominant role open modules play in managed landfill emissions. This result is consistent with reports of methane emissions being observed originating from landfill work faces in on-ground and satellite-based studies for a limited number of sites^{5,48}, and with an extensive aerial survey covering the United States that showed the prevalence of work faces in total landfill emissions^{41,49}. Our dataset shows the active surface is the dominant emission source across management and economic development levels.
--

In the Supplements:

Supplements S6, page 19	Considering the low p-value (0.01) in the two-sided two-sample Kolmogorov-Smirnov evaluation for emission per area using total site area as reference, we conclude that emission per area distributions are not comparable between managed landfills and dumping sites, with dumping sites showing significantly higher emissions per area. This may be explained by the fact that managed landfills include closed inactive modules that generally show no emissions above the GHGSat detection threshold but still add to the total site area whereas, by definition, dumping sites do not show these closed inactive modules. This reflects the expected effect of definitively covering and closing some parts of managed landfills, thus confirming the efficiency of this mitigation strategy. However, managing a landfill does not mean that emissions are completely mitigated: we observe emissions arising from the open active modules from managed landfills (See Supplements S13, S14). These emissions of the active module are not fully mitigated despite the landfill being managed.
---

< End of common answer to referees #2 #3 on difference of distribution between landfills and dumping sites >

L34: In fact, the amount of waste produced is more strongly linked to population (e.g. Chen et al. Environ. Res. Lett. 2020). Furthermore, the amount of organic waste, which is the most relevant for CH₄ emissions, should stabilise once food demand is met and not increase further with economic development, but rather only with population. I suggest the authors revise this sentence accordingly.

Thank you for this correction, we included it in the revised manuscript.

Main text line 33-34	Global waste production has nearly tripled since 1965, reaching 2 billion tons per year in 2016 and, with growing population ⁶ , is expected to further increase by 70% by 2050 ⁷ .
--

L39: The warming is relative to the pre-industrial temperature, which is defined in the IPCC AR6 as the mean for the period 1850-1900. I suggest the authors state that this is the warming relative to this time period.

We included this suggestion in the revised manuscript.

Main text line 36-40	Methane is a short-lived (~9-year lifetime ⁸) but potent greenhouse gas and its anthropogenic emissions are the second most important contributor to human-induced climate change after anthropogenic carbon dioxide emissions, accounting for ~30% of current positive warming relative to pre-industrial temperatures (1850-1900 average) ⁹ .
---

L50: Specify what is “them” in this sentence – is it “the emissions”?

In this sentence “them” designates “the models”, we corrected the sentence in the revised manuscript:

Main text line 51-52	The parameters (e.g. methane generation potential of the waste) that drive these models are also uncertain and specific to each facility ^{15,16} .
--

L74: In this sentence presumably “2021” refers to the year but it should be clearly stated, e.g. “A third of the methane emission plumes detected in the year 2021 by TROPOMI...”

We included a clarification in the revised manuscript.

Main text line 98-99	A third of methane emission plumes detected in TROPOMI data from the year 2021 are related to urban areas ⁴ .
---

L80: This sentence is not clear, does the 82% of all plume detections refer to the total number of detections across all 46 urban areas? Please revise to make this clear.

Following a comment from referee #3, we removed most of this paragraph from the Main text to include it in the Supplements. However, we do clarify the revised text based on this question, now included in Supplement S2.

Supplement S2 page 3	A small fraction of urban areas account for the majority of detected plumes: 14 urban areas show at least 21 detected plumes with a total encompassing 82% of all TROPOMI plumes detected for the 46 urban areas.
--

L81: It's not clear what is being referred to here, do the authors refer to the 130 urban areas where plumes were detected with GHGSat.

Indeed, we are referring to the remainder of urban areas (N=84) that do not show plumes in TROPOMI data but that were targeted by GHGSat. We revised this sentence as well to include a clarification (moved to Supplements S2 in the revised manuscript).

Supplement S2 page 3	Not detecting urban scale emission plumes in TROPOMI data does not mean that there are no emissions (see next Supplement S3). We find that 62 urban areas do not show detected plumes because of coverage-related challenges (e.g. persistent cloudiness or sharp elevation gradients), 19 urban areas are not expected to have total emissions exceeding the ~8 t/hr TROPOMI plume detection threshold² based on emission inventories³ and GHGSat estimates, and finally 3 urban areas are surrounded by artefact-causing surface albedo features, which complicate the detection of methane plumes.
--

Figure 1: How was the classification of the landfills as “managed” or “dumping sites” determined? I see this is mentioned L144-146 but I think it should be mentioned earlier, i.e., when Figure 1 is referred to.

As we first refer to Figure 2 (Figure 1 in the preprint) to discuss urban areas for which TROPOMI data shows methane plume, explaining how the classification between “managed landfill” and “dumping sites” has been determined would confuse the message of the paragraph where this figure is first referred to. We propose to add a sentence explaining how the classification has been determined in the caption of Figure 2 (Figure 1 in the preprint), in addition to the one in the text.

Figure 2 (Figure 1 in the preprint) caption line 114-115	This site classification has been manually determined using satellite and aerial imagery from Google Earth (see main text).
--	--

L119: The authors state they examined “emission variability against meteorology, including surface pressure”, but they should state what other meteorological variables they examined and not just “meteorology”.

We included this feedback in the revised manuscript.

See main text line 139-142	We compare site-wise emission variability against meteorological variables (10m wind speed, 2m temperature, surface pressure, surface pressure change, accumulated precipitation over two weeks) and hemisphere-corrected “day in the year” but do not find any significant link between them (see Supplements S5)
---

L119: Concerning the lack of correlation with meteorological parameters, did the authors consider the possibility that this may be because just could not be detected in the column mixing ratio observations? For instance, how sensitive is GHGSat to the lower troposphere and near surface, where the signal would be the largest?

GHGSat observes reflected backscattered sunlight in the shortwave infrared near 1.65 μm . The radiative transfer at these wavelengths provides near-unit sensitivity from the surface to the top of the troposphere (Varon et al., 2018). This means that variations of the methane concentration at these atmospheric levels are accounted for in the estimate total column enhancements.

We include this information in the revised manuscript:

See Methods line 407-409	Total columns (vertically integrated concentrations) of methane with near vertically uniform sensitivity down to the surface are retrieved from these observations using a full-physics approach...
--

See Methods line 441-444	These instruments estimate the total column (vertically integrated concentration) of methane at $\sim 25 \times 25 \text{ m}^2$ resolution over targeted $12 \times 15 \text{ km}^2$ domains ^{S2} from backscattered sunlight measurements in the shortwave infrared near 1.65 μm , that provide near-surface sensitivity.
---

Regarding plausible explanations to our results not showing a correlation with surface pressure or surface pressure change, we hypothesize that operational practices could drive most of the variability in the emissions from these landfills. Another possible explanation is that the single-observation uncertainty (median $\sim 45\%$, see Supplements S4) does not allow us to detect a meteorological driving (around a baseline emission rate) that is much less intense than $\sim 45\%$ (and despite our large number of observation, the number per specific meteorological condition is still limited, making it difficult to average out the noise). We also include this other hypothesis in the revised manuscript.

See Main text line 134-136	The plumes’ detected methane emission rates show a 2.4 t/hr median with 5 th and 95 th percentiles of 0.5 t/hr and 15.4 t/hr, respectively. The relative uncertainty of these emission rates shows a median of $\sim 45\%$ (see Supplements S4).
--

See Main text line 146-148	Another possibility is that the meteorological driving results in faint emission changes that cannot be captured within single observation uncertainty.
--

L119-122: I suggest the authors combine the sentence on L119-120 with the sentence starting “Although surface pressure has been reported...” since this logically follows. Whereas, the

second part of the sentence on L122, starting “our findings based on...” should be a new sentence.

We follow this suggestion in the revised manuscript.

See Main text line 39-145	We compare site-wise emission variability against meteorological variables (10m wind speed, 2m temperature, surface pressure, surface pressure change, accumulated precipitation over two weeks) and hemisphere-corrected “day in the year” but do not find any significant link between them (see Supplements S5), although surface pressure change has been reported to drive landfill methane emissions in on-site studies³⁸⁻⁴⁰. Our findings based on satellite observations of high-emitting active sites are consistent with recent airborne-based results ⁴¹ .
--

L143-146: How reliable is the visual classification of landfills? Was any ground-truthing performed to verify if the visual classification was correct? Also, it is possible to obtain information on the management practices at the landfills, for example, how many of these have the capacity to recover gas, and how many have active soil covers to encourage oxidation of methane?

We discussed the reliability of the landfill visual classification answering another comment above. Overall, we find 89% agreement with the independently determined Climate TRACE site classification.

For example, the US EPA collects quite some information on reporting facilities, so it is indeed possible to get additional information for some sites:

e.g. (last access 2025-03-14)

<https://ghgdata.epa.gov/ghgp/service/html/2021?id=1008215&et=undefined>

However, as similar data are not available across all the sites we include in this study, we decided not pursue further in this direction, focusing on analyses that use the full extent of the dataset.

L396-400: For TROPOMI, the IME-based emission rates lead to a 7 to 47% overestimation compared to atmospheric inversion estimates. However, I do not see an estimate for the uncertainty in the IME based emission rates for GHGSat?

To precise the phrasing employed in the manuscript for the description of methane emission rates, we name:

- Uncertainty (or precision) the random error affecting our estimates. These are for instance used to characterize the error bars in our plots
- Accuracy, the average absolute error between an estimation and an external reference.

Here, Supplement S1 of the preprint discusses for TROPOMI the accuracy of using plume-based emission rates obtained through a mass-balance approach for days where a plume is visible compared to atmospheric inversion results that have ingested all the satellite data,

including days where no plume is visible. This discussion relates to the accuracy of emission calculation approaches, but is not related to uncertainty (random error).

For single urban scale plumes observed by TROPOMI, we apply the mass-balance IME method (Varon et al., 2018) in an ensemble approach to determine the emission rate and its uncertainty, all described in Schuit et al. (2023). For aggregated results, we apply a 2-sigma filter to the set of TROPOMI plumes detected for a given urban area and report the mean and standard deviation of this filtered set, as described in the Methods section.

This discussion related to the processing approaches for TROPOMI data and their relative accuracy does not translate for GHGSat, where all the available observations are processed with the IME method, and a long-term atmospheric inversion is not feasible and would not add any information as the instrument is only sensitive to methane emission plumes.

How important are uncertainties in the emissions estimated from GHGSat retrievals – can these be assumed to be similar to those obtained for TROPOMI?

Uncertainty calculation for TROPOMI-based and GHGSat-based plume emission rate quantifications are different, but include the influence of similar error sources. Regarding single-plume GHGSat observation uncertainties (or precision, random error), these are calculated by summing quadratic contributions of (1) wind speed error; (2) methane column retrieval error (1.4 - 2.9%, details are included in the Methods section); and (3) IME calibration error (Varon et al., 2019), as described in the Methods section. Uncertainty results for GHGSat are presented in Supplement S4 of the preprint for single-plumes (Figure S4.3 of the preprint, median uncertainty is 45%) and site-aggregated results (Figure S4.4 of the preprint, median uncertainty is 45%).

Regarding the accuracy of GHGSat plume-based emission rate estimates, this question intersects with comments from referees #1 and #3. We include here a common answer to all referees.

<Beginning of common answer to referees #1 #2 #3 on accuracy>

Controlled releases are the gold standard validation reference for emission quantifications, including those made using satellite data. These consist in releasing methane at known emission rates on the ground at the same time of satellite overpasses. Satellite based estimated emission rates can then be compared with the metered rate. GHGSat has been extensively validated using both internal and public controlled releases. So far there have been two of those controlled release campaigns reported in the scientific literature (Sherwin et al., 2023, 2024). GHGSat participated in both those single-blind controlled release campaigns where the true emission rates are not known to the satellite data providers and the comparisons are done by third party, in this case the team from Stanford University (Sherwin et al., 2023, 2024). Figure R123.1 shows the results for the GHGSat-based emission estimates taken from Sherwin et al. (2023), demonstrating great agreement with the metered emission rates. Only one GHGSat observation is included in Sherwin et al. (2024), it agrees with metered emission within its uncertainty range (24% positive bias on a 0.4 t/hr metered emission rate).

Figure R123.1. Quantification performance by GHGSat, with 1-sigma X and Y error bars. The black dashed line denotes the 1:1 line. Fitted slope and uncentered R^2 shown for an ordinary least squares regression with the intercept fixed at zero (Figure copied from and caption adapted from Sherwin et al., 2023). Quantifications were made without knowledge of the real emission rate or wind speed. Please note the figure has three data points.

All controlled releases have been performed for point sources (at the scale of the satellites) up to now. These may not fully reflect the more spatially spread-out sources that landfills show (such as active surfaces) when the emission rates are close to the detection limit of the instrument. Performing controlled releases for such localized extended sources is a research effort that is currently under investigation (Dave Risk’s Flux Lab group at Saint Francis Xavier university) and we do account for the larger source area in the calibration of our emission quantification (see Method section in the manuscript).

Besides controlled releases, a different strategy to validate satellite-based emission estimates has been employed by Cusworth et al. (2024) for landfills. They showed that airborne mass-balance emission estimates based on in-situ concentration measurements (that measure the net emission flux from a landfill, including the sum of the diffuse area sources and the point sources) match within error bars with estimates based on simultaneous airborne methane plume imaging (similar to satellite-based plume imaging), with an instrument that can detect plumes down to 10 kg/hr. Besides, an in-depth study of two landfills near Madrid that included both similar airborne observations and GHGSat satellite observations showed that GHGSat satellite-based estimates satisfyingly match the total of airborne-detected plumes for same day observations (ESA study: Maasackers et al., 2023). Combined, these results point towards strongly emitting (areas in) landfills being detectable from space with emission estimates equivalent to total landfill emissions (within uncertainties).

We included the following changes in the manuscript:

In the main text lines 69-71	Calibrated mass-balance approaches are employed to translate these instantaneous snapshots into emission rates (see Methods), validated by single-blind controlled release²⁵
--

In the Methods section Lines 463-478	The calibration of this mass-balance approach against LES of known synthetic emission rates ensures that the estimated rates correctly account for the different advective transport conditions explored within the set of LES. Beyond this calibration on simulations, numerous real-life validation efforts have been organized, including controlled-releases experiments which are the validation gold standard. Notably, GHGSat participated and showed excellent agreement with metered emission rates in two single-blind controlled release campaigns, where the true emission rates (and wind speeds) are not known to the satellite data
---	---

providers and the comparisons are done by a third party (in this case a research group from Stanford University)^{25,55}. Beyond controlled releases, landfill emission rates obtained through aerial methane imagery with an instrument that can detect plumes down to 10 kg/hr have been validated against traditional aerial mass-balance results^{18,41}. Besides, an in-depth study of two landfills near Madrid that included both similar airborne observations and GHGSat satellite observations showed that GHGSat satellite-based estimates match the total of airborne-detected plumes for same day observations within uncertainties⁵⁶. Combined, these results show that GHGSat satellite-based observations can provide accurate estimates of methane emissions from waste disposal sites.

<End of common answer to referees #1 #2 #3 on accuracy>

How does the uncertainty impact the conclusion that there is no difference in the emission distribution between “managed” and “dumping site” landfills?

This is an important comment on which we improve in the revised manuscript. We now generated 1000 randomly drawn emission sets for the managed landfills and the dumping sites separately, following normal distributions centered on site-wise average emission rates using site-wise emission uncertainty as standard deviation. For all 1000 randomly drawn sets of emission rates, we compared distributions for managed landfills and dumping sites for total emissions per site and emissions normalized by the total site area. Figures R2.4 shows the distributions of obtained p-values using our own site classification and Climate TRACE’s for the 88 overlapping sites.

Figure R2.4. Distributions of p-values obtained for a two-sided K-S test comparing total emission rate distributions (blue) and total-area normalized emission rate distributions (red) between managed landfills and dumping sites, using our devised site classification (top) and Climate TRACE’s (bottom). Low p-values mean that distribution between managed landfills and dumping sites are significantly different, while p-values larger than 0.01 mean that they are not significantly different.

In both cases, p-values are overall very different from 0.01 (or lower) when comparing total emission rate distributions between managed landfills and dumping sites, and overall close to very low values when comparing total-area normalized emission rate distributions between managed landfills and dumping sites. This means that our conclusions on differences in total or area-normalized emission rate distributions between managed landfills and dumping sites, for the sites that we observed, hold when accounting for uncertainties in site classification and emission rate uncertainties.

Supplements
S6 pages 21-
22

This analysis using our own classification of site types has been included in Supplements S6 of the revised manuscript.

Referee #3 (Remarks to the Author):

Estimates of methane emissions from landfill are presented using satellite observations from GHGSat, along with urban area emissions from TROPOMI. Emissions from 151 landfill sites are estimated. The paper concludes that similar emissions are found for managed or open landfill, and that reported/modelled emissions estimates do not agree well with the estimates from the satellite.

I find the work to be extremely impressive. The technology is very exciting, and the investigation underpinning this article is exhaustive and meticulous. However, I have three general concerns about the framing of the results in the text, which should be addressed.

Firstly, I wonder how robust the comparison with reported or modelled (“bottom-up”) estimates can be, given the very small sample size in the top-down estimates (median number of samples is 5 per site), and because this sample is likely to be biased, due to the relatively high detection limit of the satellite. The discussion focuses on the difficulty in modelling landfill emissions, but couldn’t the representation errors in the satellite plume detection also be a major factor in the difference between the top-down and bottom-up (e.g., the text in lines 174 and 175 discusses under- or over-estimates in the bottom-up, but doesn’t discuss potential top-down issues)?

We include additional description and discussion elements on the advantages and drawbacks of using satellite observation in the revised manuscript main text, methods, and in the supplements. We list them below:

We now describe how satellite-based observations are complementary to legacy ground-based methods:

Main text lines 55-60	Considering all these uncertainties, independent observations of methane emitted from waste disposal sites are critical and can be obtained through various on-ground and/or aerial measurement methods¹⁸ that are deployable at site level and that can provide emission estimates at high continuous temporal resolution. Complementarily to these site-specific approaches, satellites offer extensive global coverage, providing consistent observation sets across a large number of sites.
--

We now include a description of methane plumes, including the fact that these observations are instantaneous, and these instantaneous emission rates are calibrated and validated:

Main text lines 66-71	They provide spatial images of atmospheric methane concentrations that enable the detection of anthropogenic emission plumes. These consist of strong enhancements in methane concentration that extend downwind from localized emission sources, as illustrated in Figure 1. Calibrated mass-balance approaches are employed to translate these instantaneous snapshots into emission rates (see Methods), validated by single-blind controlled release²⁵.
---

We now acknowledge that the GHGSat detection threshold only allows us to observe waste disposal sites emitting at rates located at the upper end of the emission distribution:

Main text line 120-122	Only sites for which at least one methane emission plume has been detected by GHGSat are included, meaning that our sample is on the upper end of the global waste disposal site emission rate distribution.
---

We now provide the median of single-plume emission rate relative uncertainties for GHGSat, previously just included in Supplement S4:

See Main text line 134-136	The plumes' detected methane emission rates show a 2.4 t/hr median with 5 th and 95 th percentiles of 0.5 t/hr and 15.4 t/hr, respectively. The relative uncertainty of these emission rates shows a median of ~45% (see Supplements S4).
--

Finally, Supplement S4 includes the detailed descriptions of the single-plume and site-wise average emission rate uncertainties.

Regarding the accuracy of GHGSat-based emission rates, we now include a description of the state-of-the-art regarding the validation of plume-based emission rates for high-resolution satellite instruments. This point intersects with comments from referees #1 and #2. We include here a common answer to all referees.

<Beginning of common answer to referees #1 #2 #3 on accuracy>

Controlled releases are the gold standard validation reference for emission quantifications, including those made using satellite data. These consist in releasing methane at known emission rates on the ground at the same time of satellite overpasses. Satellite based estimated emission rates can then be compared with the metered rate. GHGSat has been extensively validated using both internal and public controlled releases. So far there have been two of those controlled release campaigns reported in the scientific literature (Sherwin et al., 2023, 2024). GHGSat participated in both those single-blind controlled release campaigns where the true emission rates are not known to the satellite data providers and the comparisons are done by third party, in this case the team from Stanford University (Sherwin et al., 2023, 2024). Figure R123.1 shows the results for the GHGSat-based emission estimates taken from Sherwin et al. (2023), demonstrating great agreement with the metered emission rates. Only one GHGSat observation is included in Sherwin et al. (2024), it agrees with metered emission within its uncertainty range (24% positive bias on a 0.4 t/hr metered emission rate).

Figure R123.1. Quantification performance by GHGSat, with 1-sigma X and Y error bars. The black dashed line denotes the 1:1 line. Fitted slope and uncentered R^2 shown for an ordinary least squares regression with the intercept fixed at zero (Figure copied from and caption adapted from Sherwin et al., 2023). Quantifications were made without knowledge of the real emission rate or wind speed. Please note the figure has three data points.

All controlled releases have been performed for point sources (at the scale of the satellites) up to now. These may not fully reflect the more spatially spread-out sources that landfills show (such as active surfaces) when the emission rates are close to the detection limit of the instrument. Performing controlled releases for such localized extended sources is a research effort that is currently under investigation (Dave Risk’s Flux Lab group at Saint Francis Xavier university) and we do account for the larger source area in the calibration of our emission quantification (see Method section in the manuscript).

Besides controlled releases, a different strategy to validate satellite-based emission estimates has been employed by Cusworth et al. (2024) for landfills. They showed that airborne mass-balance emission estimates based on in-situ concentration measurements (that measure the net emission flux from a landfill, including the sum of the diffuse area sources and the point sources) match within error bars with estimates based on simultaneous airborne methane plume imaging (similar to satellite-based plume imaging), with an instrument that can detect plumes down to 10 kg/hr. Besides, an in-depth study of two landfills near Madrid that included both similar airborne observations and GHGSat satellite observations showed that GHGSat satellite-based estimates satisfyingly match the total of airborne-detected plumes for same day observations (ESA study: Maasackers et al., 2023). Combined, these results point towards strongly emitting (areas in) landfills being detectable from space with emission estimates equivalent to total landfill emissions (within uncertainties).

We included the following changes in the manuscript:

In the main text lines 69-71	Calibrated mass-balance approaches are employed to translate these instantaneous snapshots into emission rates (see Methods), validated by single-blind controlled release²⁵
--

In the Methods section Lines 463-478	The calibration of this mass-balance approach against LES of known synthetic emission rates ensures that the estimated rates correctly account for the different advective transport conditions explored within the set of LES. Beyond this calibration on simulations, numerous real-life validation efforts have been organized, including controlled-releases experiments which are the validation gold standard. Notably, GHGSat participated and showed excellent agreement with metered emission rates in two single-blind controlled release campaigns, where the true emission rates (and wind speeds) are not known to the satellite data
---	---

	providers and the comparisons are done by a third party (in this case a research group from Stanford University)^{25,55}. Beyond controlled releases, landfill emission rates obtained through aerial methane imagery with an instrument that can detect plumes down to 10 kg/hr have been validated against traditional aerial mass-balance results^{18,41}. Besides, an in-depth study of two landfills near Madrid that included both similar airborne observations and GHGSat satellite observations showed that GHGSat satellite-based estimates match the total of airborne-detected plumes for same day observations within uncertainties⁵⁶. Combined, these results show that GHGSat satellite-based observations can provide accurate estimates of methane emissions from waste disposal sites.
--	--

<End of common answer to referees #1 #2 #3 on accuracy>

Besides, in Figure 3 that compares satellite-based estimates with reported or modelled bottom-up estimates, we use site-wise averages of top-down results. These averages include both detection and non-detection of methane emission plumes, where non-detections are conservatively treated as emission rates of 0 t/hr instead of an unknown rate between 0 t/hr and the satellite detection limit (see Main text of the preprint). For example, we observe only two plumes (2.1 t/hr and 1.1 t/hr) in 30 observations at Icheon in South Korea for site #87, which includes 28 times 0 t/hr in the emission time series which is averaged. This approach that includes both detection and non-detection can thus only underestimate average emission rates, implicitly accounting for the impact of the detection threshold of GHGSat instruments by including non-detections as 0 t/hr. As we include sites for which at least one plume has been detected, it means that our study results are not representative of sites which emission rates are always below GHGSat detection threshold.

We stress this last aspect in the Main text of the revised manuscript:

Abstract line 20-21	Within this dataset , we find that our satellite-based estimates generally show no correlation with reported or modeled emission estimates.
Main text line 120-122	Only sites for which at least one methane emission plume has been detected by GHGSat are included, meaning that our sample is on the upper end of the global waste disposal site emission rate distribution.
Main text line 266-268	Across the 151 surveyed sites , we find that bottom-up and top-down satellite-based solid waste emission estimates cannot currently be reconciled at facility and country scales.

To discuss Figure 3 results through a more data driven approach, we include here a version of the Figure only including sites with at least 5 GHGSat observations (the median number of observations). We can notice that correlations remain very low in Figure R3.1.

Figure R3.1. Version of Figure 3 including only sites with at least 5 GHGSat observations. The conclusions remain unchanged. European sites (yellow) show high correlation if just considered on their own, but the change of correlation sign when excluding the highest emitters illustrates that it is mostly an artefact of the low number of points.

Finally, the question of whether the site-wise average uncertainties could explain the lack of correlation between top-down and bottom-up results can be explored by perturbing top-down emission estimates within their uncertainty range. The most favorable statistically relevant situation that we can simulate involves placing top-down results as close as possible to the 1:1 line within their reported uncertainty. Figure R3.2 illustrates this fictitious situation and provides the subsequent recomputed correlation, which remains weak at best. We can conclude from this that the results we present are robust to our uncertainty estimates.

Figure R3.2. Version of Figure 3 in very favorable top-down to bottom-up comparison conditions. Correlations remain weak at best. In this Figure, sites in the US are not split between different bottom-up calculation methods, unlike in the Main text.

We include the following additional discussion elements in Supplement S8

Supplement S8 page 27-28	Figure S8.2 reproduces Figure 3 only including sites that show at least 5 observations. The resulting correlations remain poor. Finally, Figure S8.3 illustrates for Figure 3 a fictitious situation where all sites show an agreement between top-down and bottom as well as their uncertainty range (1-sigma) can allow. We can notice that correlations still remain low. Consequently, the results shown in Figure 3 are robust to the number of available observations per site, and to the prescribed uncertainty ranges. Figure S8.2 = Figure R3.1 Figure S8.3 = Figure R3.2
---

One of the most interesting aspects of the paper is the remarkable correlation of the plume origin and the activity area within some landfills (an outstanding technical achievement by the team). However, doesn't this potentially underscore how variable these emissions could be? And therefore, how difficult or misleading it could be to compare annual reported emissions to a small number of snapshots (which are potentially biased high)?

This comment relates to accuracy across time (few satellite snapshots against yearly totals) and intersects with a comment from referee #1, so we reproduce a common answer to both here.

<Beginning of common answer to referees #1 #3 on accuracy across time>

If the few snapshots per site are significantly biased towards emissions anomalies (because of diurnal variability, or management practices, etc., short time scale variability), total summed emissions across all sites should be significantly different between GHGSat and reported or modelled bottom-up results. In Tables R13.1 and R13.2, we complete Tables S8.1 and S8.2 to include sums across all sites of each category. We can notice that while we find large site-to-site discrepancies, the total emission sums across all sites are not significantly different (considering the ~100% uncertainty for reported totals in the US, depending on the bottom-up method) between top-down and bottom-up. Finally, at global scale we report that conservatively population-averaged GHGSat results agree with UNFCCC estimates within their error bar (see Main text in preprint lines 205-206 or revised manuscript lines 222-225). Even if, per site, the small number of available points may bias emission rates towards a specific season, etc. (variability over a longer time range), we can note that, overall, our dataset shows no correlation with meteorology (see Supplements S5). We also find that only using sites for which we have a higher number of observations (above 5) does not improve the correlation against reported or modelled data (see Figure R3.1 related to a later comment, reproducing Figure 3 for sites that have at least 5 observations). All this evidence suggests that there is no significantly large variability-based emission bias when relying on the satellite data included in our dataset. Finally, the availability of methane-sensitive satellite data at facility-scale is growing, which will allow to study this question further in the future.

Reporting scope	Number of sites	GHGSat reported difference	–	GHGSat totals	Reported totals
-----------------	-----------------	----------------------------	---	---------------	-----------------

USA	22	0.43 ± 1.45 t/hr	32.6 ± 4.8 t/hr	23.1 t/hr (reported to GHGRP) 49.9 t/hr (using emission models)
Canada	8	-0.49 ± 0.93 t/hr	3.2 ± 0.6 t/hr	7.1 t/hr
EU	7	1.75 ± 1.22 t/hr	14.5 ± 1.5 t/hr	2.3 t/hr
All together	37	0.48 ± 1.49 t/hr	50.3 ± 5.1 t/hr	32.6 t/hr (with data reported to GHGRP) 59.3 t/hr (using reported emission models for US)

Table R13.1. Methane emission rate statistics comparing GHGSat-based rates and data included in facility-scale reporting program databases.

Waste disposal site types	Number of sites	GHGSat – Climate TRACE difference	GHGSat totals	Climate TRACE totals
Managed landfills	77	-0.63 ± 4.93 t/hr	170 ± 8 t/hr	219 t/hr
Dumping sites	32	-0.35 ± 2.93 t/hr	63 ± 6 t/hr	74 t/hr
All together	109	-0.55 ± 4.44 t/hr	234 ± 10 t/hr	294 t/hr

Table R13.2. Methane emission rate statistics comparing GHGSat-based rates and data calculated by Climate TRACE.

We include these elements in the supplements of the revised manuscript:

Supplement S8 pages 26-28	We included this analysis in Supplements S8.
--

<End of common answer to referees #1 #3 on accuracy across time>

Secondly, the fact that no statistical difference could be detected between managed and open landfills is an intriguing, and perhaps concerning, result. Several parts of the text suggest that a reason for focusing on landfill is that it is “low-hanging fruit” for mitigation, if management is improved (e.g., L43 – 45, L89 – 90, L243 – 245). However, the results of this paper would suggest that such efforts do not necessarily result in lower emissions? Perhaps I’m misunderstanding, but in any case, I think the reader needs to be led through these apparently contradictory statements and findings.

This comment is intersecting with one provided by referee #2. We are reproducing here a common answer to both comments.

<Beginning of common answer to referees #2 #3 on difference of distribution between landfills and dumping sites>

The emission rate distribution comparison between managed landfills and dumping sites is one of the striking results that our dataset provides. We agree with both referees #2 and #3 that the conclusions drawn from this comparison must be phrased very precisely.

On the one hand, total waste disposal site emissions are relevant to discuss climate-change impacts of landfill methane emissions. Our data strikingly show that managed landfills and dumping site do not have significantly different emission rates distributions. On the other hand, emission rates normalized by the full area of waste disposal sites are relevant to assess the efficiency of potential mitigation measures put in place. Applying this normalization yields area-normalized emission rates being significantly lower for managed landfills compared to dumping sites. This reflects the expected effect of definitively covering and closing some parts of managed landfills (closed modules are included in our landfill masks), thus confirming the efficiency of this mitigation strategy, and clearing the apparent contradiction of managed landfills emitting at similar levels compared to dumping sites.

We interpret the impact of normalizing emissions by the total site area for managed landfills as a small fraction of the site area being responsible for the detected emissions. Following conclusions from our Sentinel-2 based surface activity correlating with plume origins, we identify the open active side of managed landfills (or work face) as this small area responsible for most of the emissions. These two results put the spotlight on the predominant role open active work faces play in managed landfills emissions. Similar conclusions on the predominant role of the work face have been drawn from extensive aerial survey datasets collected in the United States (Cusworth et al., 2023, Scarpelli et al., 2024). Our dataset confirms these findings with a global perspective, discussing emission distribution differences across management and economic development levels.

In a nutshell, the conclusions obtained from our dataset mean that managed landfill active surfaces are not significantly different from dumping sites regarding total emission rates (total emission distribution comparison) until they are mitigated by closing the active module for good (total area-normalized emission distribution comparison).

We include the following changes in the revised manuscript to reflect these ideas, explain the apparent contradiction and avoid misleading readers.

In the abstract: with this new formulation, we seek to directly link “similar levels of emissions” directly with “where waste is being added”

Abstract lines 21-26	This reveals major uncertainties in the current understanding of methane emissions from waste-disposal sites, warranting further investigations to reconcile bottom-up and top-down approaches. We also observe that managed landfills and dumping sites show similar levels of total emissions, with the area where waste is being added often aligning with the detected emission sources within a facility.
---

In the main text:

Main text line 161-164	Managed landfills and dumping sites do not show statistically significant different detected emission rate distributions. However, when normalized by the total site area, managed landfills show significantly lower area-
--

	normalized emission rates compared to dumping sites, thus showing expected effects of emission mitigation through closing and covering modules of the landfill (see Supplement S6).
--	---

In the main text:

Main text line 236-248	When considering only the 21 sites for which at least 16 plume origins can be identified in GHGSat observations, we find statistically-significant proximity for 18 (86%) of them (See Supplements S14). Revisiting our dataset showing that total site area-normalized emission rates are significantly lower for managed landfills compared to dumping sites (see Supplements S6 and above), we conclude that the small fraction of open active areas in managed landfill accounts for almost all of the emissions detected by GHGSat. This highlights the predominant role open modules play in managed landfill emissions. This result is consistent with reports of methane emissions being observed originating from landfill work faces in on-ground and satellite-based studies for a limited number of sites^{5,48} , and with an extensive aerial survey covering the United States that showed the prevalence of work faces in total landfill emissions ^{41,49} . Our dataset shows the active surface is the dominant emission source across management and economic development levels.
--

In the Supplements:

Supplements S6, page 19	Considering the low p-value (0.01) in the two-sided two-sample Kolmogorov-Smirnov evaluation for emission per area using total site area as reference, we conclude that emission per area distributions are not comparable between managed landfills and dumping sites, with dumping sites showing significantly higher emissions per area. This may be explained by the fact that managed landfills include closed inactive modules that generally show no emissions above the GHGSat detection threshold but still add to the total site area whereas, by definition, dumping sites do not show these closed inactive modules. This reflects the expected effect of definitively covering and closing some parts of managed landfills, thus confirming the efficiency of this mitigation strategy. However, managing a landfill does not mean that emissions are completely mitigated: we observe emissions arising from the open active modules from managed landfills (See Supplements S13, S14). These emissions of the active module are not fully mitigated despite the landfill being managed.
---

< End of common answer to referees #2 #3 on difference of distribution between landfills and dumping sites >

Thirdly, I didn't understand the reason for the inclusion of the TROPOMI plume analysis in this paper. These plumes originate from integrated urban sources, and it wasn't clear to me how they tie in with the source-specific GHGSat landfill plumes. By the time I got to the discussion of the GHGSat results (L100 onwards), I was starting to get confused about the different numbers of identified plumes, urban areas, sites, etc. from the two different instruments. I

suspect a reader unfamiliar with these datasets could be even more confused. Could the TROPOMI work be cut? I'm not sure what it adds.

TROPOMI provides observations that have been instrumental in revealing methane super-emitters from space, and are being used to guide (so-called “tip-and-cue” strategy) a significant fraction of high-resolution observations aiming to identify the industrial sites (landfills in this case) responsible for urban-scale emission hotspots (Maasackers et al., 2022; Schuit et al., 2023). We included the TROPOMI plume detection analysis in this work to provide an urban-scale context for methane emissions in the 130 investigated urban areas.

Considering this comment and other referee’s comments calling for additional explanations regarding high-resolution-satellite-based emission monitoring and to improve the flow of the manuscript, we have moved most of the TROPOMI-based urban-scale context analysis to the Supplements, only keeping its overall conclusions in the revised manuscript.

Here is the simplified TROPOMI-related paragraph in the main text :

Main text lines 98-107	A third of methane emission plumes detected in TROPOMI data from the year 2021 are related to urban areas ⁴ . In 2021 and 2022, we detect 897 plumes with TROPOMI across 46 urban areas among the 130 covered by GHGSat (see Supplements S1 and S2). These detections, that depend on observational coverage and emission magnitude (see Supplement S3) , are located on six different continents, with the majority coming from Asia (Figure 2). TROPOMI plumes illustrate the mitigation potential concentrated in urban areas, which harbor a range of sources including wastewater treatment, natural gas distribution, and incomplete combustion ³⁷ . Waste disposal sites, however, are some of the most concentrated and mitigable sources in urban areas and are therefore the facilities that we focus on in our GHGSat analysis.
---

Here are the updated details in the Supplements

Supplement S2 page 3	This supplement provides additional information on TROPOMI plume detections for the urban areas targeted by GHGSat. The median number of plume detections over two years is 6 (Figure S2.1). A small fraction of urban areas account for the majority of detected plumes: 14 urban areas show at least 21 detected plumes with a total encompassing 82% of all TROPOMI plumes detected for the 46 urban areas. These cities combine regular coverage with large emissions, leading to a high number of detected emission plumes. The cities are (sorted per country, themselves in alphabetical order):  • Argentina: Buenos Aires • Bangladesh: Dhaka • India: Delhi, Ahmedabad, Lucknow, Kanpur, Hyderabad, Mumbai, Kolkata • Iran: Tehran • Morocco: Casablanca • Pakistan: Lahore, Karachi • Spain: Madrid
--

	Not detecting urban scale emission plumes in TROPOMI data does not mean that there are no emissions (see next Supplement S3). We find that 62 urban areas do not show detected plumes because of coverage-related challenges (e.g. persistent cloudiness or sharp elevation gradients), 19 urban areas are not expected to have total emissions exceeding the ~8 t/hr TROPOMI plume detection threshold² based on emission inventories³ and GHGSat estimates, and finally 3 urban areas are surrounded by artefact-causing surface albedo features, which complicate the detection of methane plumes.
--	---

We also swapped the order of presentation of satellites in the introduction, first presenting GHGSat and then its link to TROPOMI:

Main text lines 71-80	Our study focuses on measurements from GHGSat’s high-resolution (~25 × 25 m²) methane imaging satellites, which capture targeted 12 × 15 km² scenes and detect facility-scale plumes at emission rates as low as 100 kg/hr. They can be attributed to individual sources across oil and gas facilities (onshore and offshore), coal mines, and waste disposal sites^{5,26–28}. Many of these individual sources were first coarsely spatially identified^{5,29–33} with the TROPospheric Monitoring Instrument (TROPOMI) on board the Sentinel-5 Precursor satellite³⁴. It maps the atmospheric concentration of methane with daily global coverage and a resolution down to 7 × 5.5 km², enabling the detection⁴ of methane plumes which can be followed-on, in a so-called ‘tip-and-cue’ strategy, by targeted GHGSat observations to identify the exact sources. This approach has been demonstrated for four urban areas with strongly-emitting waste disposal sites⁵. Here, we present a global GHGSat-based survey of methane emissions from waste disposal sites across 130 urban areas in 47 countries during 2021–2022.
--

In summary, I want to emphasize how impressive I found this work. However, to me, it is impressive from a technological/methodological perspective. Whether it should be published in a general interest journal such as Nature likely depends on how the authors address the above comments. I think a general reader may be very interested in the technology and methodology, but I think that the conclusions about landfill emissions are less well-defined, or potentially a little misleading.

Specific comments

L13 and 37: Slightly nit-picking, but methane emissions being the second most important contributor to climate change seems to be a somewhat ambiguous framing. I assume this means in comparison to total anthropogenic CO₂ emissions, but a reader could potentially interpret this as being compared to some emissions sector.

We understand how this could be wrongfully interpreted. The revised manuscript follows the referee’s advice, with a simplified abstract and more details in the introduction.

Abstract lines 13-14	Methane is a potent but short-lived greenhouse gas and rapid reductions of its anthropogenic emissions could help decrease near-term warming ¹ .
--

Main text line 36-40	Methane is a short-lived (~9-year lifetime⁸) but potent greenhouse gas and its anthropogenic emissions are the second most important contributor to human-induced climate change after anthropogenic carbon dioxide emissions , accounting for ~30% of current positive warming relative to pre-industrial temperatures (1850-1900 average)⁹ .
---

L36: I suggest providing the methane lifetime here, for the non-expert audience.

We included the methane lifetime in the revised manuscript (see previous reported change).

L43: I don’t know what “treatment with energy recovery” means?

We listed here all mitigation approaches included in Table 2 by Höglund-Isaksson et al. (2020) for municipal solid waste. Examining the details included in Supplement 6 of Höglund-Isaksson et al. (2020), “treatment with energy recovery” seems to be here an umbrella-phrase to capture how they recommend to handle the fraction of waste that cannot be separated, reused, recycled or treated in an anaerobic digester, citing Höglund-Isaksson et al. (2020) in their Supplement 6: “In the MFR scenario is assumed that all waste that is not possible to separate, reuse, recycle or treat in an anaerobic digester, is combusted in a well-managed incinerator with energy recovered and utilized”

We adapted the revised manuscript to clarify this sentence and better reflect the main mitigation strategies Höglund-Isaksson et al. (2020) include in their Supplement 6.

Main text lines 43-46	However, some mitigation options are available e.g. banning organic waste in landfills, source separation, reuse, recycling or treatment with an anaerobic digester¹¹ . If these are implemented to their fullest potential, 2050 methane emissions from solid waste could be as low as 11 million tons per year ¹¹ .
--

L59: What is the difference between an emission hotspot and super-emitting point source?

An emission hotspot may contain several super-emitting point sources, e.g. the urban area around Madrid appears as an emission hotspot in TROPOMI data, and zooming with high resolution instruments we notably find two super-emitting landfills.

We adapt the text to precise that hotspots are larger scale and gather one or several super-emitting sources.

Main text lines 63-65	Satellite remote sensing of atmospheric methane can play an active role in methane emission mitigation by locating emission hotspots and identifying the super-emitting sources they contain¹⁹ .
--

L80: I found this a little confusing. Is this saying that 14 areas EACH had ≥ 21 detected plumes?

Indeed, among the 46 urban areas for which we detect plumes in TROPOMI data, 14 urban areas each show plume counts higher or equal to 21, with a total number of plumes amounting to 82% of all the plumes detected across the 46 urban areas that show plumes. This means that a few urban areas account for the majority of detections. These areas are located where TROPOMI shows a high coverage and have expected emissions that are above the TROPOMI plume detection threshold of about 8 t/hr, see discussion in Supplement S3.

Following a preceding question, the details of TROPOMI-based analysis has been moved to Supplement S2. We now expand on the explanations to clarify the sentence.

Supplement S2 page 3	A small fraction of urban areas account for the majority of detected plumes: 14 urban areas show at least 21 detected plumes with a total encompassing 82% of all TROPOMI plumes detected for the 46 urban areas.
--

L85: it's not immediately clear what the "19 areas" refers to here (and I still don't really understand why it's important).

These 19 urban areas do not show methane emission plumes in TROPOMI data primarily because of expected emissions (from bottom-up inventories) being lower than the ~ 8 t/hr detection threshold of the TROPOMI instrument.

We think that it is important to explain why we could not detect urban scale plumes for these areas in TROPOMI data, because not detecting urban-scale plumes does not necessarily mean that there are no emissions. Following an earlier comment, this discussion has been moved to Supplement S2.

Supplement S2 page 3	Not detecting urban scale emission plumes in TROPOMI data does not mean that there are no emissions (see next Supplement S3). We find that 62 urban areas do not show detected plumes because of coverage-related challenges (e.g. persistent cloudiness or sharp elevation gradients), 19 urban areas are not expected to have total emissions exceeding the ~ 8 t/hr TROPOMI plume detection threshold² based on emission inventories³ and GHGSat estimates, and finally 3 urban areas are surrounded by artefact-causing surface albedo features, which complicate the detection of methane plumes.
---

L89: Waste disposal sites being the "most concentrated and mitigatable sources" seems rather a strong statement. Is this universally true? What is the reference? I don't think you need this motivating statement in any case.

Relying on TROPOMI data to identify super-emitting sources in urban emission hotspots by using wind-rotated averages of methane column observations (see method in Maasackers et al., 2022), we have mostly identified landfills, thus motivating this statement. It is indeed very general and we follow the referee's advice to nuance it in the revised manuscript but keep

part of the sentence to highlight the difference with (for example) more diffuse gas leakage from gas distribution or use.

Main text lines 105-107	Waste disposal sites, however, are some of the most concentrated and mitigable sources in urban areas and are therefore the facilities that we focus on in our GHGSat analysis.
--

L111: I'm clearly missing something here, but I was expecting the number of observations to be equal to the number with at least one plume plus the number of null detections. But 1447 minus 1085 doesn't equal 449...

Thank you very much for catching this inconsistency. The GHGSat observation dataset that we study covers three types of objects:

- observations: which are 12x15 km² images acquired by GHGSat satellites
- sites: targeted anthropogenic emitters on the ground (solid waste disposal sites in this case)
- plumes: cluster of enhanced methane concentration pixels identified in an image, attributed to a site

In the dataset that we study, the simplest case is that one observation contains one site, for which one plume is detected. But we also find instances where several plumes are observed for a given site in a single observation (e.g. for site #4 near Buenos Aires, Argentina), or that several sites are contained within one observation (e.g. for sites #51 and #52, two landfills that are a few kilometers apart near Madrid), each possibly showing an emission plume.

The raw file that contains all the plume detections and non-detections considered in this study has been shared on the Code Ocean capsule:

`/data/Plume_detection_CSVs/GHGSat_detected_plumes.csv`

It contains 1534 lines:

- 1085 positive detection lines
- 449 non-detection lines (15 of which are wrongful duplicates of positive detections for site #118, an artefact of subsite-merging when gathering the dataset that has no impact on the results as all (non-)detections are summed for each site within a given observation)

These data have been acquired through 1354 unique observations which, crossed with the 151 targeted sites, yields 1447 unique *site observations*, which is the number that we chose to report. However, for plumes, we reported the numbers of positive and negative plume detections included in the raw csv file, hence the non-additivity to match the number of observations (the inconsistency between the numbers and the sentence must have been introduced along draft versions).

We adjust the numbers to match the formulation of the preprint:

- There are 1013 site observations that contain at least one plume (1085 plumes in total)
- There are 434 site observations that show no plume (449 non-detection lines in the file, minus 15 non-impacting duplicate artefacts).

Here are the changes included in the revised manuscript:

Main text lines 128-132	Out of the 1447 observations, 1013 show at least one emission plume above GHGSat’s detection threshold (1085 plumes in total, examples are shown in Figure 1; quantified as described in Methods). We conservatively consider the emission rate of the 434 site-level null detections to be zero even though we may miss (possibly diffuse) emissions that are lower than the GHGSat detection threshold.
--

We also correct the file /data/Plume_detection_CSVs/GHGSat_detected_plumes.csv in the Code Ocean capsule to remove these 15 artefact lines of non-detections.

164: I suggest using “estimates”, rather than “data” (which, to me, suggests an observation). You may want to clarify what is meant by “modelled” (e.g., “process model-based”, or similar).

We follow the referee’s suggestions in the revised manuscript:

Main text lines 176-179	Figure 3 compares facility-level GHGSat-detected methane emission rates against national site-level reporting programs ^{43–45} (based on process-based models or gas capture efficiency assumptions), and against emissions obtained from data-driven models developed by the non-profit Climate TRACE coalition ⁴² .
---

Main text lines 181-183	Overall, we find no correlation between satellite-based and reported or modeled estimates ($r = 0.04$ for reported emissions, and $r = 0.18$ for Climate TRACE)
---

L167 – 169: Couldn’t this be due to the sampling bias in your dataset? Also, I don’t understand what bias is referred to at the start of this sentence.

Here, we refer to the difference between GHGSat-based and bottom-up reported or modelled estimates provided in Supplement S8. It is the same “bias” as the one mentioned line 165 of the preprint. We adjust the revised manuscript to clarify.

Main text lines 186-188	Although no overall bias is found between reported and GHGSat-based estimates , emissions from 14 (out of 37) landfills are at least twice as large compared to what is reported to national programs.
---

The purpose of this statement is solely to highlight that facility-to-facility disagreements between bottom-up and GHGSat-based estimates can be quite large. We cannot exclude that the sampling of our sites has an impact on these number, hence we do not draw conclusions beyond our data (see adjustments in the revised manuscript made to address similar comments).

L218: I wonder why 16 was chosen as a threshold here (21 sites with 16+ plumes)? Earlier, (L104) talks about there being 23 sites with 20+ observations. How are these two statistics consistent?

These two statistics were chosen independently:

- the goal of the first one (23 sites with 20+ observations) is to describe the large spread of observation number in our data, with a median of 5 observations per site but 23 that show 20+ observations (these 23 sites gather 707 observations in total, 49% of the total observation number). Here, an observation can include several plumes.
- Figure S14.1 discusses the impact of observation number on the p-value result describing whether Sentinel-2 detected activity and GHGSat-detected sources are significantly close. We decided to report the number of plumes from which more than 80% of the sites show this statistically-significant proximity, which is 16. Also, here, the number of plumes per site is considered, and only sites for which we have good Sentinel-2 data and activity detection results are considered (107/151).

The apparent inconsistency between these two statistics stems from the fact that the first one considers “observations” and all sites, whereas the second is related to plumes (sometimes several per observations) and to a subset of sites (107/151).

L224: How will this fine-grained information help to focus mitigation efforts?

Engaging with various stakeholders and operators through different projects, we found that being able to provide information on where emissions come from within a given facility has a strong actionable value (e.g. the correlation between plume origins and surface activity). Indeed, operators can reflect on the mitigation strategies implemented at these locations, or at the operations that took place at the moment of the satellite overpass.

We clarify this point in the revised manuscript.

Main text lines 251-252	This spatial information can for example help site operators focus mitigation efforts more effectively.
---

L226 – 228: I don’t really understand why these other sources are pertinent to the discussion. Cut these lines?

Two observations within our dataset identify landfill-neighboring facilities as sources of methane plumes. One is a wastewater treatment plant near Shanghai (China), the other is a gas processing plant linked to the landfill near Madrid (Spain). The latter is an especially interesting case because it exemplifies that gas capturing as an emission mitigation strategy is only efficient if the entire capturing chain is reliable. Being able to identify these specific cases also illustrates, again, the source attribution capabilities of high-resolution satellite observation, and the emission mitigation potential it can hold.

We rephrase this last part of the paragraph to better underline why these two examples of plumes are interesting:

Main text lines 252-256	Our dataset also includes two example plumes originating from adjacent facilities: a biogas plant near the Las Dehesas landfill near Madrid and from a wastewater treatment plant near Shanghai (both filtered from the analysis, see Supplements S15). They illustrate the mitigation potential
---

	that satellite observation can detect in facilities related to and neighboring waste disposal sites.
--	---

L243 – 245: As in my general comments, this concluding sentence seems to be contradicted by the results on open vs managed landfill. Can you clarify?

We replied and proposed adjustments in the abstract, main text and supplements below the general comment. We reproduce here the changes introduced in the revised manuscript, from the common answer to this point intersecting comments from referee #2 and #3 (see above).

In the abstract: with this new formulation, we seek to directly link “similar levels of emissions” directly with “where waste is being added”

Abstract lines 21-26	This reveals major uncertainties in the current understanding of methane emissions from waste-disposal sites, warranting further investigations to reconcile bottom-up and top-down approaches. We also observe that managed landfills and dumping sites show similar levels of total emissions, with the area where waste is being added often aligning with the detected emission sources within a facility.
---

In the main text:

Main text line 161-164	Managed landfills and dumping sites do not show statistically significant different detected emission rate distributions. However, when normalized by the total site area, managed landfills show significantly lower area-normalized emission rates compared to dumping sites, thus showing expected effects of emission mitigation through closing and covering modules of the landfill (see Supplement S6).
---

In the main text:

Main text line 236-248	When considering only the 21 sites for which at least 16 plume origins can be identified in GHGSat observations, we find statistically-significant proximity for 18 (86%) of them (See Supplements S14). Revisiting our dataset showing that total site area-normalized emission rates are significantly lower for managed landfills compared to dumping sites (see Supplements S6 and above), we conclude that the small fraction of open active areas in managed landfill accounts for almost all of the emissions detected by GHGSat. This highlights the predominant role open modules play in managed landfill emissions. This result is consistent with reports of methane emissions being observed originating from landfill work faces in on-ground and satellite-based studies for a limited number of sites^{5,48}, and with an extensive aerial survey covering the United States that showed the prevalence of work faces in total landfill emissions^{41,49}. Our dataset shows the active surface is the dominant emission source across management and economic development levels.
--

In the Supplements:

Supplements S6, page 19	Considering the low p-value (0.01) in the two-sided two-sample Kolmogorov-Smirnov evaluation for emission per area using total site area
--

	as reference, we conclude that emission per area distributions are not comparable between managed landfills and dumping sites, with dumping sites showing significantly higher emissions per area. This may be explained by the fact that managed landfills include closed inactive modules that generally show no emissions above the GHGSat detection threshold but still add to the total site area whereas, by definition, dumping sites do not show these closed inactive modules. This reflects the expected effect of definitively covering and closing some parts of managed landfills, thus confirming the efficiency of this mitigation strategy. However, managing a landfill does not mean that emissions are completely mitigated: we observe emissions arising from the open active modules from managed landfills (See Supplements S13, S14). These emissions of the active module are not fully mitigated despite the landfill being managed.
--	---

Methods

L377: I suggest “its imaging capabilities enable the detection of anthropogenic methane...”

We included this correction in the revised manuscript.

Methods lines 412-414	In addition to being used in long-term inverse analyses, its imaging capabilities enable the detection of anthropogenic methane emission plumes that arise from the world’s largest emitters²⁹.
--

L381 – 383: The IME equation is provided later. If you’re going to give the equation, it would make sense to include it at the first mention.

For TROPOMI, Schuit et al. (2023) embeds the IME method within an ensemble to compute both the emission rate and its uncertainty. This differs from the approach chosen for GHGSat data, the TROPOMI-related calibration coefficients differ as well (the effective wind speed calibration for the IME method is instrument dependent). Because GHGSat data are more instrumental to the conclusions than TROPOMI-based results that provide total urban emission context, we decided to comprehensively describe the IME method in the section presenting GHGSat-related methods.

L414: What is the reference for the LES simulations?

These LES simulations were performed by Maasackers et al. (2022) in the same study as the one providing the calibration coefficients given right after mentioning the LES (ref #6 in the preprint). So, the reference provided at the beginning of the sentence applies for both the LES and the calibration coefficients.

Methods line 453-454	with U_{eff}, the effective wind speed, calibrated against the 10-m wind speed based on a set of Large Eddy Simulations (LES)⁵
---

L415: Introduce a_i here, rather than in the following sentence.

We included this suggestion in the revised manuscript:

Methods line 453-456	with U_{eff} , the effective wind speed, calibrated against the 10-m wind speed based on a set of Large Eddy Simulations (LES) ⁵ ; $L = \sqrt{\sum_i a_i}$, the plume length computed as the square-root of the plume total area, with a_i the area of the i-th pixel included in the plume; and $\Delta X_{CH_4,i}$, the local enhancement above background of the methane total column for this i-th pixel.
--

L426: By assuming zero uncertainty on the null results, aren't the site-average emissions uncertainties going to be under-estimated, if they are included in the mean?

The non-detections contribute to the uncertainty quantifications by being included in the first step that assesses the sampling-based uncertainty of the site-wise average. Figure R3.3 shows the evolution of the sampling uncertainty (due to the first step bootstrap resampling of observations, red), of the mean emission rate uncertainty (due to the uncertainty of averaged observations, blue) and of the final reported uncertainty (black) against the fraction of non-detection. We note that the sampling uncertainty dominates the final reported uncertainty for large fractions of non-detection.

Figure R3.3. Total reported relative uncertainty (black) and the contributions of sampling (bootstrap resampling in the uncertainty quantification, red) and average observation uncertainty (blue, linked to single-plume reported uncertainties) against the fraction of non-detections. This example assumes 25 available observations and a 45% single-observation uncertainty for positive detections. Changing the number of available observations does not change the conclusions on the impact of the fraction of non-detections.

Figure R3.4 shows the effect on the non-detection frequency based on all the 151 GHGSat-observed sites. For sites with no non-detections or very few, the relative uncertainty overall follows what is expected of independent Gaussian variables through averaging over an

increasing number N of observations (decrease by $1/\sqrt{N}$ of the uncertainty). Sites showing higher non-detection frequencies all exhibit higher uncertainties compared to this trend. This illustrates that our choice of assuming zero uncertainty for non-detect does not result in yielding lower uncertainties for sites with lots of non-detections, and that they actually contribute significantly to the reported final uncertainty by increasing the sampling uncertainty.

Figure R3.4. Reported relative uncertainty as a function of the number of observations and the fraction of non-detections for all 151 GHGSat-observed sites.

Supplement:

P2, L3: “compared”, rather than “compare”

We restructured and simplified the sentence in the revised manuscript:

Supplement S1 page 2	For four strongly-emitting cities, Figure S1.1 compares averaged plume-based IME emission estimates and posterior emission rates obtained through atmospheric inversion. The former are based on the automated TROPOMI plume detections over years 2021 and 2022 gathered for this work, and the latter are results for the year 2020 drawn from Maasackers et al. (2022)¹.
--

P18: I don’t understand this sentence: “This points towards landfill management and emission mitigations measures being more efficient for closed modules that show no activity than for active ones that are in operation.” How can a landfill be operating efficiently if there is no activity?

This sentence has been deleted, taking into account the feedback of referee #2 and #3 on this discussion comparing managed landfills and dumping sites emission distributions. See previous replies on this question.

Referee #3 (Remarks on code availability):

I have taken a brief look at the code. It looks to be well documented (although would benefit from a README).

We have updated the Code Ocean capsule to include a README explaining the purpose of each script.

References cited in the replies

Nanda, S., Berruti, F. Municipal solid waste management and landfilling technologies: a review. *Environ Chem Lett* **19**, 1433–1456 (2021). <https://doi.org/10.1007/s10311-020-01100-y>

Climate TRACE. Climate TRACE Emissions Inventory Tracking Real-time Atmospheric Carbon Emissions. (2023).

Varon, D. J., Jacob, D. J., McKeever, J., Jervis, D., Durak, B. O. A., Xia, Y., and Huang, Y.: Quantifying methane point sources from fine-scale satellite observations of atmospheric methane plumes, *Atmos. Meas. Tech.*, **11**, 5673–5686, <https://doi.org/10.5194/amt-11-5673-2018>, 2018.

Schuit, B. J., Maasackers, J. D., Bijl, P., Mahapatra, G., van den Berg, A.-W., Pandey, S., Lorente, A., Borsdorff, T., Houweling, S., Varon, D. J., McKeever, J., Jervis, D., Girard, M., Irakulis-Loitxate, I., Gorroño, J., Guanter, L., Cusworth, D. H., and Aben, I.: Automated detection and monitoring of methane super-emitters using satellite data, *Atmos. Chem. Phys.*, **23**, 9071–9098, <https://doi.org/10.5194/acp-23-9071-2023>, 2023.

Varon, D. J., McKeever, J., Jervis, D., Maasackers, J. D., Pandey, S., Houweling, S., et al. (2019). Satellite discovery of anomalously large methane point sources from oil/gas production. *Geophysical Research Letters*, **46**, 13507–13516. <https://doi.org/10.1029/2019GL083798>

Sherwin, E.D., Rutherford, J.S., Chen, Y. et al. Single-blind validation of space-based point-source detection and quantification of onshore methane emissions. *Sci Rep* **13**, 3836 (2023). <https://doi.org/10.1038/s41598-023-30761-2>

Sherwin, E. D., El Abbadi, S. H., Burdeau, P. M., Zhang, Z., Chen, Z., Rutherford, J. S., Chen, Y., and Brandt, A. R.: Single-blind test of nine methane-sensing satellite systems from three continents, *Atmos. Meas. Tech.*, **17**, 765–782, <https://doi.org/10.5194/amt-17-765-2024>, 2024.

Joannes D. Maasackers et al., Using satellites to uncover large methane emissions from landfills. *Sci. Adv.* **8**, eabn9683 (2022). DOI:10.1126/sciadv.abn9683

Daniel H. Cusworth et al., Quantifying methane emissions from United States landfills. *Science* **383**, 1499-1504 (2024). DOI:10.1126/science.adi7735

Maasakkers, J. D. *et al.* EDAP+ TN on GHGSat Validation.

<https://earth.esa.int/documents/d/earth-online/technical-note-on-ghgsat-validation-pdf> (2023).

Scarpelli, T.R., Cusworth, D.H., Duren, R.M., Kim, J., Heckler, J., Asner, G.P., Thoma, E., Krause, M.J., Heins, D. and Thorneloe, S., 2024. Investigating Major Sources of Methane Emissions at US Landfills. *Environmental Science & Technology*, 58(49), pp.21545-21556.

Andrew K. Thorpe *et al.*, Attribution of individual methane and carbon dioxide emission sources using EMIT observations from space, *Sci. Adv.* 9, eadh2391 (2023). DOI:10.1126/sciadv.adh2391

Ayasse, A.K., Cusworth, D.H., Howell, K., O'Neill, K., Conrad, B.M., Johnson, M.R., Heckler, J., Asner, G.P. and Duren, R., 2024. Probability of Detection and Multi-Sensor Persistence of Methane Emissions from Coincident Airborne and Satellite Observations. *Environmental Science & Technology*, 58(49), pp.21536-21544.

Höglund-Isaksson, L., Gómez-Sanabria, A., Klimont, Z., Rafaj, P. and Schöpp, W., 2020. Technical potentials and costs for reducing global anthropogenic methane emissions in the 2050 timeframe—results from the GAINS model. *Environmental Research Communications*, 2(2), p.025004.

Full copy-pasted revised manuscript references for the manuscript excerpts

1. Ocko, I. B. *et al.* Acting rapidly to deploy readily available methane mitigation measures by sector can immediately slow global warming. *Environ. Res. Lett.* **16**, 054042 (2021).
2. European Commission. EDGAR (Emissions Database for Global Atmospheric Research) Community GHG Database version 8.0. (2018).
3. Jacob, D. J. *et al.* Quantifying methane emissions from the global scale down to point sources using satellite observations of atmospheric methane. *Atmospheric Chem. Phys.* **22**, 9617–9646 (2022).
4. Schuit, B. J. *et al.* Automated detection and monitoring of methane super-emitters using satellite data. *Atmospheric Chem. Phys.* **23**, 9071–9098 (2023).
5. Maasakkers, J. D. *et al.* Using satellites to uncover large methane emissions from landfills. *Sci. Adv.* **8**, eabn9683 (2022).
6. Chen, D. M.-C., Bodirsky, B. L., Krueger, T., Mishra, A. & Popp, A. The world's growing municipal solid waste: trends and impacts. *Environ. Res. Lett.* **15**, 074021 (2020).
7. Kaza, S., Yao, L. C., Bhada-Tata, P. & Woerden, F. V. *What a Waste 2.0*. (The World Bank Group, 2018).
8. Canadell, J. G. *et al.* Global Carbon and other Biogeochemical Cycles and Feedbacks. *Climate Change 2021: The Physical Science Basis. Contribution of Working Group I to the Sixth Assessment Report of the Intergovernmental Panel on Climate Change* 673–816 (2021) doi:10.1017/9781009157896.007.

9. IPCC. Summary for Policymakers. *Climate Change 2021: The Physical Science Basis. Contribution of Working Group I to the Sixth Assessment Report of the Intergovernmental Panel on Climate Change* (2021) doi:10.1017/9781009157896.001.
10. IPCC. Summary for Policymakers. *Global Warming of 1.5°C. An IPCC Special Report on the impacts of global warming of 1.5°C above pre-industrial levels and related global greenhouse gas emission pathways, in the context of strengthening the global response to the threat of climate change, sustainable development, and efforts to eradicate poverty* 1–32 (2018).
11. Höglund-Isaksson, L., Gómez-Sanabria, A., Klimont, Z., Rafaj, P. & Schöpp, W. Technical potentials and costs for reducing global anthropogenic methane emissions in the 2050 timeframe –results from the GAINS model. *Environ. Res. Commun.* **2**, 025004 (2020).
12. Höglund-Isaksson, L. *et al.* *Non-CO₂ Greenhouse Gas Emissions in the EU-28 from 2005 to 2070: GAINS Model Methodology*. <https://pure.iiasa.ac.at/id/eprint/16977/> (2018).
13. United Nations Framework Convention on Climate Change. GHG data from UNFCCC. (2023).
14. Scharff, H. & Jacobs, J. Applying guidance for methane emission estimation for landfills. *Waste Manag.* **26**, 417–429 (2006).
15. Amini, H. R., Reinhart, D. R. & Mackie, K. R. Determination of first-order landfill gas modeling parameters and uncertainties. *Waste Manag.* **32**, 305–316 (2012).
16. Wang, Y. *et al.* Methane emissions from landfills differentially underestimated worldwide. *Nat. Sustain.* 1–12 (2024).
17. Nanda, S. & Berruti, F. Municipal solid waste management and landfilling technologies: a review. *Environ. Chem. Lett.* **19**, 1433–1456 (2021).
18. Mønster, J., Kjeldsen, P. & Scheutz, C. Methodologies for measuring fugitive methane emissions from landfills – A review. *Waste Manag.* **87**, 835–859 (2019).
19. Nisbet, E. G. *et al.* Methane Mitigation: Methods to Reduce Emissions, on the Path to the Paris Agreement. *Rev. Geophys.* **58**, e2019RG000675 (2020).
20. Varon, D. J. *et al.* Quantifying methane point sources from fine-scale satellite observations of atmospheric methane plumes. *Atmospheric Meas. Tech.* **11**, 5673–5686 (2018).
21. Irakulis-Loitxate, I., Guanter, L., Maasackers, J. D., Zavala-Araiza, D. & Aben, I. Satellites Detect Abatable Super-Emissions in One of the World’s Largest Methane Hotspot Regions. *Environ. Sci. Technol.* **56**, 2143–2152 (2022).
22. Guanter, L. *et al.* Mapping methane point emissions with the PRISMA spaceborne imaging spectrometer. *Remote Sens. Environ.* **265**, 112671 (2021).
23. Roger, J. *et al.* High-Resolution Methane Mapping With the EnMAP Satellite Imaging Spectroscopy Mission. *IEEE Trans. Geosci. Remote Sens.* **62**, 1–12 (2024).
24. Thorpe, A. K. *et al.* Attribution of individual methane and carbon dioxide emission sources using EMIT observations from space. *Sci. Adv.* **9**, eadh2391 (2023).
25. Sherwin, E. D. *et al.* Single-blind validation of space-based point-source detection and quantification of onshore methane emissions. *Sci. Rep.* **13**, 3836 (2023).
26. Varon, D. J. *et al.* Satellite Discovery of Anomalously Large Methane Point Sources From Oil/Gas Production. *Geophys. Res. Lett.* **46**, 13507–13516 (2019).
27. Varon, D. J., Jacob, D. J., Jervis, D. & McKeever, J. Quantifying Time-Averaged Methane Emissions from Individual Coal Mine Vents with GHGSat-D Satellite Observations. *Environ. Sci. Technol.* **54**, 10246–10253 (2020).

28. MacLean, J.-P. W. *et al.* Offshore methane detection and quantification from space using sun glint measurements with the GHGSat constellation. *Atmospheric Meas. Tech.* **17**, 863–874 (2024).
29. Pandey, S. *et al.* Satellite observations reveal extreme methane leakage from a natural gas well blowout. *Proc. Natl. Acad. Sci.* **116**, 26376–26381 (2019).
30. Lauvaux, T. *et al.* Global assessment of oil and gas methane ultra-emitters. *Science* **375**, 557–561 (2022).
31. Sadavarte, P. *et al.* Methane Emissions from Superemitting Coal Mines in Australia Quantified Using TROPOMI Satellite Observations. *Environ. Sci. Technol.* **55**, 16573–16580 (2021).
32. Tu, Q. *et al.* Quantifying CH₄ emissions in hard coal mines from TROPOMI and IASI observations using the wind-assigned anomaly method. *Atmospheric Chem. Phys.* **22**, 9747–9765 (2022).
33. Tu, Q. *et al.* Quantification of CH₄ emissions from waste disposal sites near the city of Madrid using ground- and space-based observations of COCCON, TROPOMI and IASI. *Atmospheric Chem. Phys.* **22**, 295–317 (2022).
34. Lorente, A., Borsdorff, T., Martinez-Velarte, M. C. & Landgraf, J. Accounting for surface reflectance spectral features in TROPOMI methane retrievals. *Atmospheric Meas. Tech.* **16**, 1597–1608 (2023).
35. Hersbach, H. *et al.* The ERA5 global reanalysis. *Q. J. R. Meteorol. Soc.* **146**, 1999–2049 (2020).
36. Esri, Maxar, Earthstar Geographics & Community, the G. U. ESRI World Imagery. (2022).
37. Hopkins, F. M. *et al.* Mitigation of methane emissions in cities: How new measurements and partnerships can contribute to emissions reduction strategies. *Earths Future* **4**, 408–425 (2016).
38. Young, A. Volumetric changes in landfill gas flux in response to variations in atmospheric pressure. *Waste Manag. Res.* **8**, 379–385 (1990).
39. Xu, L., Lin, X., Amen, J., Welding, K. & McDermitt, D. Impact of changes in barometric pressure on landfill methane emission. *Glob. Biogeochem. Cycles* **28**, 679–695 (2014).
40. Kissas, K., Ibrom, A., Kjeldsen, P. & Scheutz, C. Annual upscaling of methane emission field measurements from two Danish landfills, using empirical emission models. *Waste Manag.* **150**, 191–201 (2022).
41. Cusworth, D. H. *et al.* Quantifying methane emissions from United States landfills. *Science* **383**, 1499–1504 (2024).
42. Climate TRACE. Climate TRACE Emissions Inventory Tracking Real-time Atmospheric Carbon Emissions. (2023).
43. U.S. Environmental Protection Agency Office of Atmospheric Protection. Greenhouse Gas Reporting Program (GHGRP). (2023).
44. Government of Canada. Greenhouse Gas Reporting Program (GHGRP) - Facility Greenhouse Gas (GHG) Data. (2023).
45. European Environment Agency. European Pollutant Release and Transfer Register (E-PRTR). (2024).
46. Stark, B. M., Tian, K. & Krause, M. J. Investigation of U.S. landfill GHG reporting program methane emission models. *Waste Manag.* **186**, 86–93 (2024).

47. Columbia University, Center for International Earth Science Information Network, CIESIN. Gridded Population of the World, Version 4 (GPWv4): Population Density Adjusted to Match 2015 Revision UN WPP Country Totals, Revision 11. (2018) doi:10.7927/H4F47M65.
48. Kumar, P. *et al.* Detection and long-term quantification of methane emissions from an active landfill. *Atmospheric Meas. Tech.* **17**, 1229–1250 (2024).
49. Scarpelli, T. R. *et al.* Investigating Major Sources of Methane Emissions at US Landfills. *Environ. Sci. Technol.* (2024).
50. Nesser, H. *et al.* High-resolution US methane emissions inferred from an inversion of 2019 TROPOMI satellite data: contributions from individual states, urban areas, and landfills. *Atmospheric Chem. Phys.* **24**, 5069–5091 (2024).
51. Veefkind, J. P. *et al.* TROPOMI on the ESA Sentinel-5 Precursor: A GMES mission for global observations of the atmospheric composition for climate, air quality and ozone layer applications. *Remote Sens. Environ.* **120**, 70–83 (2012).
52. Jervis, D. *et al.* The GHGSat-D imaging spectrometer. *Atmospheric Meas. Tech.* **14**, 2127–2140 (2021).
53. Ramier, A. *et al.* High Resolution Methane Detection with the GHGSat Constellation. in *GLOC 2023 conference proceedings* (2023).
54. Molod, A. *et al.* *The GEOS-5 Atmospheric General Circulation Model: Mean Climate and Development from MERRA to Fortuna*. <https://ntrs.nasa.gov/citations/20120011790> (2012).
55. Sherwin, E. D. *et al.* Single-blind test of nine methane-sensing satellite systems from three continents. *Atmospheric Meas. Tech.* **17**, 765–782 (2024).
56. Maasackers, J. D. *et al.* *EDAP+ TN on GHGSat Validation*. <https://earth.esa.int/documents/d/earth-online/technical-note-on-ghgsat-validation-pdf> (2023).
57. United Nations, D. of E. & Social Affairs, P. D. World Population Prospects 2022 Revision. (2022).
58. United Nations, D. of E. & Social Affairs, P. D. World Population Prospects 2015 Revision. (2015).
59. Livingston, D. C. Colorimetric Analysis of the NTSC Color Television System. *Proc. IRE* **42**, 138–150 (1954).
60. Wang, Z., Bovik, A. C., Sheikh, H. R. & Simoncelli, E. P. Image quality assessment: from error visibility to structural similarity. *IEEE Trans. Image Process.* **13**, 600–612 (2004).
61. Wang, Z. & Bovik, A. C. Mean squared error: Love it or leave it? A new look at Signal Fidelity Measures. *IEEE Signal Process. Mag.* **26**, 98–117 (2009).

Full copy-pasted revised supplement references for the supplement excerpts

1. Maasackers, J. D. *et al.* Using satellites to uncover large methane emissions from landfills. *Sci. Adv.* **8**, eabn9683 (2022).
2. Schuit, B. J. *et al.* Automated detection and monitoring of methane super-emitters using satellite data. *Atmospheric Chem. Phys.* **23**, 9071–9098 (2023).
3. European Commission. EDGAR (Emissions Database for Global Atmospheric Research) Community GHG Database version 8.0. (2018).

4. Piano, S. L., Ferretti, F., Puy, A., Albrecht, D. & Saltelli, A. Variance-based sensitivity analysis: The quest for better estimators and designs between explorativity and economy. *Reliab. Eng. Syst. Saf.* **206**, 107300 (2021).
5. Hersbach, H. *et al.* The ERA5 global reanalysis. *Q. J. R. Meteorol. Soc.* **146**, 1999–2049 (2020).
6. United Nations Framework Convention on Climate Change. GHG data from UNFCCC. (2023).
7. United Nations, D. of E. & Social Affairs, P. D. World Population Prospects 2015 Revision. (2015).
8. United Nations, D. of E. & Social Affairs, P. D. World Population Prospects 2022 Revision. (2022).
9. Columbia University, Center for International Earth Science Information Network, CIESIN. Gridded Population of the World, Version 4 (GPWv4): Population Density Adjusted to Match 2015 Revision UN WPP Country Totals, Revision 11. (2018) doi:10.7927/H4F47M65.
10. Kaza, S., Yao, L. C., Bhada-Tata, P. & Woerden, F. V. *What a Waste 2.0*. (The World Bank Group, 2018).
11. Esri, Maxar, Earthstar Geographics & Community, the G. U. ESRI World Imagery. (2022).

We thank the referees for their feedback and provide below in **red** point-by-point replies to their comments and questions. Excerpts from the revised manuscript and supplements are provided to highlight the changes.

In addition to answering the referees, we have made two minor improvements to the manuscript. For ten plumes, we improved the originally used plume masking to better capture the emissions. Furthermore, we increased the number of samples in our bootstrapping approach to have better converged mean and standard deviation estimates. Neither minor improvement leads to significant changes in the results or conclusions of the manuscript; both are detailed below.

As part of our investigation for the responses, we carefully double-checked all plumes masks used in the study and found we should update a small number of them to more accurately capture the emissions from the landfill. For 10 plumes among the 1085 that our dataset includes, we improved the plume mask and report the impact on emission rates in Table R0.1. All original and revised emission rates have overlapping uncertainty ranges. Site-level averaged estimates generally change very little (see Table R0.2) and all original and revised site-level estimates have overlapping uncertainty ranges.

Besides, while running consistency checks on these changes, we noticed that our bootstrap sampling approach ($N_{\text{bootstrap}} = 1000$) did not always include enough samples to provide site-wise ensemble averages very close to the site-wise input data averages. We therefore increase $N_{\text{bootstrap}}$ to 100000. Figure R0.1 discusses the impact of this change, showing a standard deviation of site-wise ensemble average differences to data averages decreasing from 1.3% for $N_{\text{bootstrap}} = 1000$ to 0.1% for $N_{\text{bootstrap}} = 100000$. Site-wise uncertainty estimates do not change significantly when increasing $N_{\text{bootstrap}}$ compared to $N_{\text{bootstrap}} = 1000$. Consequently, in this revised manuscript, we updated all site-wise results using $N_{\text{bootstrap}} = 100000$.

Beyond the underlying data, these changes have no impact on the high-level results and conclusions of this study.

We have updated the N bootstrap in the revised manuscript, as well as all reported numbers.

In Methods lines 480-482	First, in a bootstrapping approach, we randomly ($N=100000$) resample our set of observations by randomly picking single observations with replacement.
---

Figure R0.1. Impact of the number of samples in the bootstrap ensemble on the difference between site-wise ensemble averages and site-wise data averages (left), and on the difference between site-wise ensemble uncertainty obtained with N_bootstrap = 100000 and N_bootstrap = 1000 (right).

Site ID	Date	Plume ID (in *.tif file name)	Original emission rate [t/hr]	Revised emission rate [t/hr]
4	2021-04-16	5TB2hqH_866	44.7 ± 25.0	42.9 ± 24.1
42	2021-09-01	AVK5QUz_1908	2.7 ± 1.2	2.9 ± 1.3
58	2022-07-03	A_5Gi0-_4384	3.9 ± 1.1	2.9 ± 0.9
64	2021-03-15	ASf1dOp_722	1.6 ± 0.7	1.4 ± 0.6
71	2021-05-08	ATW8MBe_963	10.4 ± 5.8	9.3 ± 5.2
75	2022-03-15	AYLG1-I_3592	17.4 ± 7.3	7.7 ± 3.2
75	2022-11-07	-b4BW-I_7517	10.2 ± 4.8	8.1 ± 3.8
103	2022-02-14	-XuE7M2_3309	11.0 ± 5.9	5.0 ± 2.7
103	2022-12-19	DbiHbM2_8751	5.4 ± 3.8	2.9 ± 2.1
144	2022-05-30	AZYBfor_4124	2.6 ± 1.9	1.4 ± 1.0

Table R0.1. Report on the 10 plumes for which we revised emission rates, thus modifying site-wise averaged results for the involved 8 sites.

Site ID	N obs.	Original emission rate [t/hr]	Revised emission rate [t/hr]	Change [%]
4	67	22.07 ± 1.95	21.99 ± 1.92	-0.34 %
42	20	1.04 ± 0.35	1.06 ± 0.35	1.75 %
58	9	4.26 ± 0.83	4.18 ± 0.84	-1.96 %
64	32	2.54 ± 0.26	2.53 ± 0.26	-0.29 %
71	22	9.06 ± 1.47	9.01 ± 1.49	-0.50 %
75	23	6.91 ± 0.95	6.39 ± 0.75	-7.52 %
103	57	5.87 ± 0.79	5.71 ± 0.78	-2.64 %
144	2	2.63 ± 1.15	2.05 ± 0.96	-21.79 %

Table R0.2. Impact at site level of changing the emission rate for the 10 plumes, combined with increasing N_bootstrap from 1000 to 100000.

Referees' comments:

Referee #1 (Remarks to the Author):

The manuscript presents several novel and significant findings, highlighting both discrepancies and consistencies between bottom-up and top-down observations. The use of satellite-based remote sensing offers fresh insights into global solid waste management, and the revised version provides a clearer presentation of key results, well supported by Supplements. I appreciate the authors' efforts in addressing my previous concerns and am pleased to recommend this version for minor revisions:

We thank the referee for their positive evaluation of the revised manuscript.

Line 61: Please replace "about 1500" with the exact number for clarity and precision.

We have included the suggested change in the revised manuscript:

In the main text lines 61-62	We present here a global-scale survey of methane emissions from waste disposal sites using 1447 high-resolution satellite observations.
--

Lines 72–73: I wonder if the GHGSat detection threshold is 13.3 mg/m²/day, which is derived from 100 kg/hr divided by (12 × 15 km²). If so, please state this explicitly.

The GHGSat detection threshold applies to spatially concentrated sources and has been established based on point source controlled releases with known emission magnitudes. The detection threshold applies to sources with limited areas, ranging from leaking gas pipelines (point sources with extents well below GHGSat ~25 × 25 m² spatial resolution) to small localized sources (e.g. ~250 × 250 m²) like active areas within landfills or dumping sites. Consequently, the GHGSat detection threshold cannot be normalized by the total targeted observation area (12 × 15 km²).

We have precised this aspect in the main text of the revised manuscript:

In the main text Lines 72-75	Our study focuses on measurements from GHGSat's high-resolution (~25 × 25 m ²) methane imaging satellites, which capture targeted 12 × 15 km ² scenes and detect facility-scale plumes arising from localized sources at emission rates as low as 100 kg/hr.
--

We also now provide examples in the Methods section

In Methods Lines 433 - 437	The GHGSat instruments have an empirically measured methane column precision range of 1.4 – 2.9 % ⁵³ , which allows them to observe emission plumes from point (e.g., a gas pipeline leak) or very localized sources (e.g., active faces of landfills) emitting more than ~100 kg/hr (this detection threshold increases with wind speed) ²⁸ .
--

Lines 121–122: The authors mention that the sampled sites fall on the upper end of the global waste disposal site emission rate distribution. This raises a concern regarding the representativeness of these sites for assessing global landfill and dumpsite greenhouse gas

(GHG) emissions. Could the authors clarify how these data can be generalized to represent global emission patterns, given the potential bias toward high-emission sites?

To address the referee’s concern regarding the representativeness of our sample, we give more insight into the emission distributions of the datasets considered. Below, we detail how site emitting above the GHGSat detection threshold are responsible for most of the emissions included in the bottom-up datasets and how that part of the emission distribution is adequately covered in our survey.

Figure R1.1 Precisely illustrates how our sample of sites compares to the expected distributions of facility disposal site-level emission rates included in the bottom-up datasets that we included in this study.

Figure R1.1. Distribution of emission rates for the 151 sites included in this study (left y-axes) compared to the cumulative distribution functions (right y-axes) for total emissions (left panel) and total number of sites (right panel) for all the bottom-up facility-scale datasets included in this study and for our sample of sites.

Unlike E-PRTR and the US and Canadian GHGRP, the Climate TRACE dataset is a global scale dataset that includes both managed landfills and dumping sites. It shows a bend in its total emission and total number of sites cumulative distributions around 2.5 t/hr, where the few sites above this threshold (~1% of total site number) represent ~12.5% of total Climate TRACE emissions (~36.7 Mt/yr). The largest contribution to total emissions comes from sites emitting between 0.1 and 2.5 t/hr. While only 54% of Climate TRACE sites emit above 0.1 t/hr, they contribute 96% of the total Climate TRACE emissions. Regarding datasets containing reported data, sites emitting above 0.1 t/hr gather 97%, 93% and 77% of total reported emission for the US GHGRP (municipal landfills), Canada GHGRP and European E-PRTR, respectively. These high coverage ratios highlight the potential of high-resolution satellite imagery to monitor solid waste methane emission globally (for sites with localized enough emission sources).

While our dataset does only cover a very small fraction of all the sites included in Climate TRACE, it spans nearly three orders of magnitude in emission rates, showing averaged site-wise emission rates ranging between 0.03 t/hr and 21.99 t/hr. This range completely

encompasses the emission range that contributes most significantly to total Climate TRACE emissions. Furthermore, 106 out of the 151 sites included in our sample show emissions below 2.5 t/hr, the inflection point observed in the Climate TRACE cumulative emission distribution. Thus, while our sample is indeed on the upper end of the global emission distribution, most of the waste disposal sites that it includes are representative of the 0.01 t/hr – 2.5 t/hr emission range contributing most to total emissions.

We now refer to this discussion in the main text:

In the main text Lines 172 - 175	This estimated skewness is probably conservative as the 100 kg/hr detection threshold and selective targeting of GHGSat would limit the inclusion of low-emitting sites. This detection threshold enables to cover 54% of the sites (assuming localized enough emission sources) included in the facility-scale waste disposal site emission database compiled by Climate TRACE (Tracking Real-Time Atmospheric Carbon Emissions)⁴², but these 54% of sites amount to 96% of total emissions (see Supplements S4).
--

We also include this discussion in Supplements S4.

In Supplements S4 pages 9 and 13	Finally, Figure S4.5 compares the cumulative emission distribution of our site sample and the GHGSat detection limit to other facility-scale bottom-up datasets, illustrating the potential of high-resolution satellite observations to cover most of the total facility-scale solid waste methane emissions. [...] The Climate TRACE dataset⁴ is a global scale facility-level emission dataset that includes both managed landfills and dumping sites. As illustrated in Figure S4.5, it shows a bend in its total emission and total number of sites cumulative distributions around 2.5 t/hr, where the few sites above this threshold (~1% of total site number) represent ~12.5% of total Climate TRACE emissions (~36.7 Mt/yr, EDGAR v8 includes a total of 38 Mt/yr). The largest contribution to total emissions comes from sites emitting between 0.1 and 2.5 t/hr. Overall, sites emitting above 0.1 t/hr amount to 54% of all Climate TRACE sites while gathering 96% of total Climate TRACE emissions. Similarly, in reported datasets, sites emitting above 0.1 t/hr account for 97%, 93% and 77% of total reported emission for the US GHGRP, Canada GHGRP and European E-PRTR, respectively. These high coverage ratios highlight the potential of high-resolution satellite imagery to monitor solid waste methane emission globally (for sites with localized enough emission sources). Figure S4.5 = Figure R1.1
---

Given the 2.5 t/hr bending point exhibited by the Climate TRACE distribution, we have tested the robustness of our two main quantitative conclusions (namely (1) the managed landfill and dumping site distribution comparison; and (2) the top-down and bottom-up facility scale

comparison) against the range of emissions included in our sample. Figures R1.2 and R1.3 reproduce Figure S6.2 (managed landfill and dumping site distribution comparison) and Figure 3 (top-down and bottom-up facility scale comparison) only including sites with averaged emission rates below 2.5 t/hr. Even relying only on this subset of our sample, that is most representative of the emission range which contributes the most to total emissions, we find that our conclusions remain unchanged.

Figure R1.2. Distributions of total (full colors) and full-area-normalized (lines) methane emission rates for “managed landfills” (orange) and “dumping sites” (purple) following our classification, including only sites with emissions lower than 2.5 t/hr. Managed landfills and dumping site total emission rate distributions are not significantly different, while total-area normalized emission rate distributions are. These conclusions remain unchanged from the analysis of the full data set shown in Figure S6.2.

Figure R1.3. Version of Figure 3 including only sites with averaged emission rates below 2.5 t/hr. The correlation between bottom-up and top-down estimates remain unchanged compared to the analysis of the full data set shown in Figure 3.

We now also include this discussion in Supplements S6 and S8 as well:

In Supplements S6 page 25	Finally, we also assess the sensitivity of our results to the emission rate range of the sampled sites. The 2.5 t/hr inflection point in the cumulative distribution of facility-level emissions in the Climate TRACE dataset (Figure S4.5) indicates that sites emitting below this threshold account for the majority of total emissions. Consequently, we also discuss our comparison of managed landfill and dumping site emissions using only sites emitting below 2.5 t/hr. Figure S6.6 reproduces Figure S6.2 for the 106/151 sites that emit below 2.5 t/hr. We note that our conclusions remain unchanged: total emission distributions are not significantly different between managed landfills and dumping sites, while area-normalized emissions are. Figure S6.6 = Figure R1.2
--

In Supplements S8 page 32	Finally, we also assess the sensitivity of our results to the emission rate range of the sampled sites. Indeed, the bend at 2.5 t/hr in the cumulative distribution of facility-level emission rates exhibited by the Climate TRACE dataset (see Figure S4.5) shows that the sites emitting below 2.5 t/hr contribute most to the total emissions. Consequently, we also discuss our comparison of GHGSat-based emission rates against reported or modelled data using only sites emitting below 2.5 t/hr. Figure S8.4 reproduces Figure 3 for the 106/151 sites that emit below 2.5 t/hr. We note that our conclusions remain unchanged: correlations between top-down and bottom-up datasets remains low. Figure S8.4 = Figure R1.3
---

Lines 122–124: The manuscript refers to different frequencies of satellite observations. Since GHG emissions from landfill sites are inherently time-dependent, it is unclear how these varying observation frequencies are interpreted in the context of emission magnitude. Could the authors elaborate on how such temporal discrepancies are addressed when extrapolating or projecting periodical GHG emissions? Additionally, how are inter-site observation variabilities reconciled to ensure consistency and robustness in the global-scale assessment?

We describe in the Method section how all sites and observations are treated consistently to obtain the results that are instrumental to our conclusions from the raw single observation dataset. For each site, we average all the available single observation emission rates, assuming a conservative 0 t/hr emission rate when no emission plume is detected and assuming all observations are equally representative of the mean emission rate. We also provide a conservative emission rate uncertainty by accounting for both single observation uncertainty

and the sampling uncertainty (see Methods “Estimating site-level GHGSat averages”). As a result, the uncertainty reported for sites with fewer observations or strongly diverging estimates will be relatively higher. Thus, our averaged emission estimates and their related uncertainty together account for the variability of single observations within every given site, across all sites.

Another way to look at this question is to examine whether the site-wise emission variability is driven by (for example) meteorological variations. For the observations included in our dataset, we show in Supplements S5 that we do not find any significant link between meteorological parameter variation and single-observation emission rates. Answering a question from referee #2 (see Figure R2.1 in this document), we assess that most of the site-wise emission variability can be explained by single-observation uncertainty. This remarkable site-wise emission rate consistency greatly alleviates concerns about the possible impact of emission variability on reported averages.

The revised manuscript now better underlines how we account for both single observation uncertainty and variable temporal sampling in our averaged emission rate estimates:

In the main text Lines 150 - 160	The highest three site-averaged detected emission rates are found at the Norte III landfill in Buenos Aires, Argentina (22.0 ± 1.9 t/hr), at a landfill near Hong Kong, China (10.0 ± 2.7 t/hr) and at a landfill near Tehran, Iran (9.4 ± 4.9 t/hr). Averaged emission rates show a median 45% relative uncertainty that accounts for both single observation and sampling uncertainties, calculated consistently across all sites (see Methods).
--

In Methods lines 478 - 482	Given a set of observations for a waste disposal site, we employ a two-step random sampling approach to evaluate the site-level averaged emission rate and its uncertainty, accounting for both single-observation and sampling uncertainties . First, in a bootstrapping approach, we randomly (N=100000) resample our set of observations
--

Referee #2 (Remarks to the Author):

This manuscript examines methane emissions from landfills across the globe using satellite remote sensing. Two satellites and instruments are used, the TROPOMI instrument onboard Sentinel 5P, which maps methane globally, and GHGSat, which is used to determine point source emissions based on plume images.

The authors have revised the manuscript based on the previous review and to a large extent addressed the comments from that review. I have now mostly minor comments, and only a couple of more major comments.

Major comments:

L21-26: "...also observe that managed landfills and dumping sites show similar levels of total emissions"

Although the authors have modified this statement based on the first review comments, I think it is still misleading. The authors state themselves that when the sites are normalized by area, dumping sites have larger emissions compared to managed landfills. In order to compare landfill emissions it is necessary to normalize by something that reflects the size of the landfill, since all else being the same bigger landfills would be expected to have bigger emissions. Therefore, I think the authors should add here (i.e. in the abstract/summary) that when normalized by landfill area the dump site emissions are larger than those of managed landfills.

We thank the reviewer for their positive evaluation of the revised manuscript and understand the concern expressed here. We have modified the abstract to present differences in emissions normalized by facility area, so that this statement cannot mislead readers into thinking that management in landfills does not reduce emissions:

In the Abstract lines 24 - 25	We also observe that managed landfills show lower emissions per area than dumping sites [...]
--

Furthermore, I would like to see further investigation into the distribution of the emissions from dump sites versus managed landfills. The authors say they use a Kolmogorov-Smirnov (KS) test to check if the emission distributions differ. I'm not a statistician, but I think the KS test is only really suitable for continuous distributions and does not perform well where the distributions have long tails, which applies to these data. Therefore, I think the authors, should examine other methods for comparing the distributions.

Like any natural phenomenon, methane emissions can take any positive value (no mathematical and physical reason would exclude at least a single value), hence the distribution of methane emissions is fundamentally continuous. Regarding the tail of the distribution, we follow the referee's advice and now also include the result of a two-sample Anderson-Darling (A-D) test, which is a modification of the K-S test that provides more weight to the tail of the distribution.

We have performed an additional A-D test for all the Figures included in Supplements S6, and the conclusions remain unchanged.

The following changes have been implemented in the supplementary data of the revised manuscript:

In Supplements S6 pages 20 - 25	This supplement provides a comparison of site-wise averaged emission rate distributions for managed landfills and dumping sites. First, Figure S6.1 compares raw averaged emission rates and no-detection frequency between managed landfill and dumping sites. We perform two different statistical tests to assess whether the samples for managed landfills and dumping sites follow similar underlying distributions: a two-sided two-sample Kolmogorov-Smirnov (K-S) and a two-sample Anderson-Darling (A-D) tests. The latter gives more weight to the tail of the distribution than the former. Here, both yield non-significant p-values. This means that the null hypothesis cannot be rejected and thus that GHGSat-based methane emission rate distributions for managed landfills and dumping sites are not significantly different. Figures S6.1, S6.2, S6.4 and S6.5 now also include A-D test p-values.
---

L20: "Within this dataset..."

Although in response to the first review, the authors now add the qualification that the results only apply to this dataset, I think it is necessary to spell-out that "this dataset" represents only a small fraction of all landfill emissions and only land fills with very large emissions (detection limit of GHGSat being 100 kg/hr).

Following this comment and another from referee #3, we now expand on the description of key GHGSat observation characteristics in the abstract to help readers assess what "this dataset" covers:

In the Abstract lines 18 - 20	We present a survey of methane emissions from 151 individual waste disposal sites across six continents using high-resolution satellite observations that can detect localized methane emissions above 100 kg/hr. Within this dataset, we find that [...]
--

Minor comments:

L21: "...we find that our satellite-based estimates generally show no correlation with reported or modelled emission estimates"

I suggest the authors state here that this is at the facility and country scales.

We removed the country scale from this work following a comment from referee #3, and we now add the statement that this result is at facility scale:

In the Abstract lines 20 - 22	Within this dataset, we find that our satellite-based estimates generally show no correlation with reported or modeled emission estimates at facility scale .
--

L48: It is not clear what “These emission estimates” are, presumably those from inventories, but it is not stated anywhere.

These emission estimates refer to the previously cited numbers, which are compiled from bottom-up inventories and future emission scenarios for the future. Since the concept of bottom-up inventories is introduced later in the manuscript, we specify only "solid waste" at this point for clarity.

In the main text lines 49 - 50	Solid waste emission estimates are based on widely used first-order decay models ¹² that are also used in country-level reporting of methane emissions ¹³ and employed at facility scale.
--

L90-91: I am not sure what “Sentinel 2 detected surface activity” actually means, what kind of activity is detected.

Given the method employed to detect it, we name structural changes in Sentinel-2 RGB imagery as “Sentinel-2 detected surface activity”. We now add this precision in the revised manuscript.

In Figure 1 caption lines 91 – 93	The spatiotemporal distributions of all GHGSat plume origins and Sentinel-2 detected surface activity (structural changes between visual Sentinel-2 images, see Methods) for the Casablanca landfill are shown in panels n and o, respectively.
--

L142: Please change “any significant link between them” to “any significant link with them” as “between” would mean between the meteorological variables, whereas what is meant is between the emissions and meteorological variables.

We corrected the revised manuscript:

In Main text lines 141 - 144	We compare site-wise emission variability against meteorological variables (10m wind speed, 2m temperature, surface pressure, surface pressure change, accumulated precipitation over two weeks) and hemisphere-corrected “day in the year” but do not find any significant link with them (see Supplements S5), [...]
---

L141-142: I think it is a bit too speculative to conclude that “This finding suggests that operational practices could be driving emission variability for the sites we observed” based on the fact that no dependence of methane emission with meteorological parameters could be found. Are the differences in emissions between observations of the same site significant compared to the uncertainty of the emission estimates? And could the lack of dependence on meteorological parameters be simply because these variations are too small to be detected by GHGSat, and/or that there are too few observations of each site in different meteorological conditions?

To answer this question in detail, we further evaluated the uncertainty in the individual emission estimates. Figure R2.1 explores the distribution of emission rate differences between all site-wise positive detection pairs against the sum of the uncertainties for each of these site-wise positive detection pairs. Differences greater in absolute value than the sum of both uncertainties are significant.

Figure R2.1. Absolute difference in emission rate (Q_i) compared to single observation uncertainty (Q_{uncert}_i) sum for all site-wise positive detection pairs.

We can note that the overwhelming majority of emission differences for site-wise positive detection pairs are not significant, meaning that the variability could be explained by single observation uncertainty. We agree with the referee that this indeed suggests that possible meteorological driving of emission may not be detectable if its expected impact is below single-observation uncertainty levels. We did mention this possibility in the revised manuscript, right after hypothesising the impact of operational practices (lines 146-148): “Another possibility is that the meteorological driving results in faint emission changes that cannot be captured within single observation uncertainty”. Given the uncertainty of the individual observations, we now revise the manuscript to only mention the hypothesis related to emission variability and single observation uncertainty, and include Figure R2.1 and the related discussion in Supplement S5.

In main text lines 147 - 149	This finding could be explained by meteorological driving producing too small emission changes compared to single observation uncertainty for the sites included in our dataset.
In Supplements S5, pages 18-19	This absence of meteorological driving or its undetectability may be explained by the level of single-observation quantification uncertainty. Figure S5.2 compares the absolute difference in emission rate against the sum of single observation uncertainty for all available site-wise positive

detection pairs. We can observe that most of the emission rate differences are lower than the sum of uncertainties, meaning that most of the variability could be explained by single observation uncertainty.

Figure S5.2 = Figure R2.1

Referee #3 (Remarks to the Author):

The authors have addressed many of the reviewers' concerns and clarified several aspects of the paper. I think the paper still needs some work before it can be publishable. Furthermore, I think some of the findings need to be further softened, to avoid giving the wrong impression to readers.

All reviewers had concerns about the (necessarily) relatively small sample size, and sampling bias, and the extent to which this could be compared to bottom-up products and inventories, which are time-averaged. Some new text has been added to try to address this, but I don't find all of the new text to be helpful in this regard. I elaborate below.

We thank the Referee for their comments and provide point-by-point replies to the more detailed points below.

The authors have also addressed what seems to be a misleadingly worded statement, that managed landfills were as emissive as open dump sites. They have clarified that, normalised by area, the emissions from open dump sites are much larger. But in that case, I'm not sure what wider lessons we are supposed to learn from this finding; doesn't it suggest that the two populations that have been measured just happen to have consistent total emissions, but for a large difference in landfill sizes? Unless I'm misunderstanding, I think the wording on this still needs to be softened substantially, as I'm concerned that readers could still draw the wrong conclusions from the text. See specific comments below.

We understand the concern of the Referee, which overlaps with concerns raised by Referee #2. We have therefore changed the wording in the abstract to focus on the finding that normalized by area, the emissions from the landfills are lower than from the dumping sites. We have included the change in the point-by-point responses below.

Specific comments relating to these two comments:

- L20: When I read this line, the implication seems to be that the reports are models are wrong. I'm not sure that the new addition of "within this dataset" gives the reader enough information to judge how representative the observations are. I think that a further qualifying statement is required to emphasise that this difference may be because of sampling size and sampling bias in this work.

We now provide additional description details for the satellite observations that we use, helping readers to judge the information we get from these observations:

In the Abstract lines 18 - 20	We present a survey of methane emissions from 151 individual waste disposal sites across six continents using high-resolution satellite observations that can detect localized methane emissions above 100 kg/hr. Within this dataset, we find that [...]
--

- L24-25: Unless I'm misunderstanding, as articulated above, I don't see a good justification for including this statement in the abstract. It seems to be a statistical coincidence, and I suggest cutting it.

Following comments from Referee #2, we now present the difference in emission normalized per area

In the Abstract lines 24 - 25	We also observe that managed landfills show lower emissions per area than dumping sites [...]
--

- L25: The new part of this sentence that has been added is a little confusing: “the area where waste is being added often aligning with the detected emission sources within a facility”. This should be re-worded in more plain English, but it is also an opportunity to re-state what is said in the main text: that it’s mainly the open part of the landfill that is emissive, but the covered parts are not detected. It's a nice finding.

We rephrased the sentence in plain English following the advice from the referee:

In the Abstract lines 25 - 27	[...], and that detected emission sources often align with the open non-covered parts of the facility where waste is added.
--

- L160: This needs a qualifying statement. Here, the new framing in the abstract would be appropriate (“within this dataset...”).

We included this qualifying statement, also specifying that this is related to total emissions with the next sentence explaining the difference found when looking at emissions per area:

In the main text lines 163 - 167	Within this dataset , managed landfills and dumping sites do not show statistically significant different total detected emission rate distributions. However, when normalized by the total site area, managed landfills show significantly lower area-normalized emission rates compared to dumping sites, thus showing expected effects of emission mitigation through closing and covering modules of the landfill (see Supplement S6).
--

- L171 – 172: I think this framing is also a little misleading. I don’t think you can claim to have “detected” an emission rate of “2.9 million tons per year”, because this requires (I assume) up-scaling to an annual total, and all the associated assumptions and sampling issues. I think this needs softening substantially.

We agree with the referee that the assumption of constant emissions underlying this upscaling to a yearly total needs to be repeated, along with the fact that we use averages of instantaneous snapshot emission rates, when mentioning the 2.8 million ton total (the emission update for the 10 plumes mentioned at the beginning of this document, and the increase of the bootstrap ensemble size, changed the 2.855 Mt to 2.844 Mt). We adjusted the revised manuscript accordingly.

In the main text Lines 175 - 180	Overall, the 151 waste disposal sites observed here represent a small fraction of the global total number of landfills (over 10,000 are included in the Climate TRACE coalition datasets ⁴²) but, assuming constant emissions, their collective instantaneous emission rate upscales to a yearly total of
--

	2.8 million tons. This corresponds to 7.4% of 2022 global solid waste emissions in version 8 of the Emissions Database for Global Atmospheric Research (EDGAR) inventory ² .
--	--

- L211 – 225: Given the reviewers response, and my concerns about up-scaling the point measurements, I think this paragraph should be deleted. You are extrapolating to national scales an extrapolation (in time) of a small number of samples. Surely the errors involved in such an exercise will be so large as to preclude any reliable conclusions?

The extrapolation at national scales that we presented in the first two manuscript versions relied on a population density map, assuming a conservative range of waste disposal site catchment radii (see ‘Comparison of GHGSat estimates and UNFCCC data’ in Methods, and Supplements S10 and S11 in the first two manuscript versions). As showcased by the comparison of Figures S11.2 and S11.3 (in the Supplements of the first two manuscript versions), the use of this conservative range of radii causes a large increase in the total uncertainties.

We agree with the referee that these uncertainty estimates at country level can be large, especially for cases where the density map shows sharp density changes around the waste disposal sites that we targeted. Considering that these country level results contribute less than others to the overall conclusions of this work and to keep a better focus on the most important results, we agree with the referee and remove the paragraph related to this country scale extrapolation, as well as the related Method sections and Supplements.

- L268: I’d emphasise the potential role of representation issues here, as it implies it’s just the inventories that are wrong.

We now repeat the assumption of constant emissions that is necessary to compare the average of top-down snapshot satellite estimates and bottom-up data provided as yearly totals. Following the previous comment, the extension to the national scale has been removed.

In the main text Lines 256 - 260	Across the 151 surveyed sites, assuming constant emissions to upscale the snapshot averages provided by satellites , we find that bottom-up and top-down satellite-based solid waste emission estimates cannot currently be reconciled at facility scale. This disagreement is consistent with previous facility-scale studies using aerial measurements ⁴¹ and country-scale studies using TROPOMI data ⁴⁹ .
--

- L275: This probably doesn’t need addressing in the text, but this statement does beg the question as to what satellite data would add, if we had continuous in situ monitoring... I guess it would be a way to test how representative the satellite data are (and therefore how useful for sampling unmonitored facilities)

In such studies, satellites and/or airborne methane imaging would e.g. be able to provide comprehensive scans of waste disposal facilities, identifying emission sources and thus help build better links between operational practices and modelling of the emissions. In-situ monitoring (e.g. tracer release emission calculations) could provide total emissions to be

compared to satellite estimates, and/or to provide emission estimates when satellite measurements are not possible e.g. at night. The coordination of all these different and complementary approaches would benefit all of them and, indeed as suggested by the referee, help better assess what satellites can provide for unmonitored waste disposal sites, as satellite observations can be easily extended at global scale. Conversely, site-specific bottom-up modelling improvements (e.g. better inclusion of operational practices in facility-level bottom-up estimates) that may be identified in such studies could also be applied at larger scale as well.

- Figure S8.3: This is a strange new statistical test. I'm not sure what value it adds, and I had to read it several times to try to figure out what the authors were trying to show. I strongly suggest cutting it.

The goal of this test was to show that correlations between top-down and bottom-up remain weak even if best case satellite-based emission estimates are used given their respective uncertainties. This was included to answer a comment related to the robustness of low correlations between top-down and bottom-up data to top-down uncertainties.

We follow the referee's suggestion and remove this test from the Supplements. To continue addressing the question of the low correlation robustness to uncertainties, we now consider an ensemble of 1000 random draws (assuming normal distributions) of site-wise averaged emission rates for which we compute the same correlations as in Figure 3. **Figure R3.1** shows the obtained distribution of Pearson correlation coefficients between GHGSat emission rates and reported or modelled emission rates. Overall, the correlation values included in the ensemble consistently remain weak for all random draws. Thus, our conclusions are robust to site-wise average emission rate uncertainties.

Figure R3.1. Distribution of Pearson correlation coefficient between GHGSat-based emission rate and reported bottom-up values (left) and Climate TRACE modelled emissions (right) for an ensemble of 1000 random draws for GHGSat-based emission rates, following a normal distribution based on site-wise averaged emission rates and their related uncertainties.

We now include this discussion in the Supplements of the revised manuscript, instead of Figure S8.3:

In Supplement S8 Pages 30 – 31

Figure S8.3 helps to assess the robustness of our conclusions to the site-wise averaged emission rate uncertainties. For an ensemble of 1000 random draws following a normal distribution centered on GHGSat-based averaged site-wise emission rates and with a standard deviation matching the averaged emission uncertainties, we compute similar correlations as in Figure 3. Figure S8.3 shows that correlations are consistently low across the random draws for both reported and Climate TRACE modelled bottom-up data, thus showing the robustness of our conclusions to emission rate uncertainties.

Given the sensitivity results provided in Figures S8.2 and S8.3, the results shown in Figure 3 are robust to the number of available observations per site, and to the prescribed uncertainty ranges.

Figure S8.3 = Figure R3.1

Other comments:

- I wonder if the authors have now gone a little too far in trying to say that this approach has been validated. It seems that the validation set is very small (1 or 2 releases). For me, this is fine, and we clearly need more work on validating these datasets. On L70, I'd like to see it mentioned that the validation dataset is small at present.

In the previously revised manuscript, we only reported on peer-reviewed single-blind controlled release experiments (Sherwin et al., 2023, 2024) in which GHGSat instruments took part. Besides these studies, GHGSat performs internal unblinded controlled release experiments, the results of which we report in Figure R3.2 and Table R3.1. Results overall show that GHGSat-based emission rates satisfactorily reproduce metered emissions.

Figure R3.2. Parity plot comparing estimated GHGSat-based emission rates against metered emission rates for controlled releases, for the C1-6 satellite series.

	Results
N	39
Mean bias	21 kg/hr
Bias standard deviation	178 kg/hr
Pearson correlation coefficient	0.9719
Slope	1.0221

Table R3.1. Statistics for GHGSat-based estimates compared to metered controlled release emission rates.

In the revised manuscript, we now mention these internal controlled releases as well. A to-be-peer-reviewed publication is in preparation to present a complete picture of GHGSat validation results, including using the unblinded controlled release data.

In the main text Lines 456 - 460	Notably, GHGSat participated and showed excellent agreement with metered emission rates in internal controlled releases, as well as in two single-blind controlled release campaigns, where the true emission rates (and wind speeds) are not known to the satellite data providers and the comparisons are done by a third party (in this case a research group from Stanford University) ^{25,54}
--

References

Sherwin, E.D., Rutherford, J.S., Chen, Y. *et al.* Single-blind validation of space-based point-source detection and quantification of onshore methane emissions. *Sci Rep* **13**, 3836 (2023). <https://doi.org/10.1038/s41598-023-30761-2>

Sherwin, E. D., El Abbadi, S. H., Burdeau, P. M., Zhang, Z., Chen, Z., Rutherford, J. S., Chen, Y., and Brandt, A. R.: Single-blind test of nine methane-sensing satellite systems from three continents, *Atmos. Meas. Tech.*, 17, 765–782, <https://doi.org/10.5194/amt-17-765-2024>, 2024.